# GAP-DEPENDENT BOUNDS FOR $Q$-LEARNING USING REFERENCE-ADVANTAGE DECOMPOSITION

**Zhong Zheng, Haochen Zhang & Lingzhou Xue**[*]
Department of Statistics
The Pennsylvania State University
State College, PA, 16802, USA
{zvz5337,hqz5340,lzxue}@psu.edu

## ABSTRACT

We study the gap-dependent bounds of two important algorithms for on-policy $Q$-learning for finite-horizon episodic tabular Markov Decision Processes (MDPs): UCB-Advantage (Zhang et al. 2020) and Q-EarlySettled-Advantage (Li et al. 2021). UCB-Advantage and Q-EarlySettled-Advantage improve upon the results based on Hoeffding-type bonuses and achieve the almost optimal $\sqrt{T}$-type regret bound in the worst-case scenario, where $T$ is the total number of steps. However, the benign structures of the MDPs such as a strictly positive suboptimality gap can significantly improve the regret. While gap-dependent regret bounds have been obtained for $Q$-learning with Hoeffding-type bonuses, it remains an open question to establish gap-dependent regret bounds for $Q$-learning using variance estimators in their bonuses and reference-advantage decomposition for variance reduction. We develop a novel error decomposition framework to prove gap-dependent regret bounds of UCB-Advantage and Q-EarlySettled-Advantage that are logarithmic in $T$ and improve upon existing ones for $Q$-learning algorithms. Moreover, we establish the gap-dependent bound for the policy switching cost of UCB-Advantage and improve that under the worst-case MDPs. To our knowledge, this paper presents the first gap-dependent regret analysis for $Q$-learning using variance estimators and reference-advantage decomposition and also provides the first gap-dependent analysis on policy switching cost for $Q$-learning.

## 1 INTRODUCTION

Reinforcement Learning (RL) (Sutton & Barto, 2018) is a subfield of machine learning focused on sequential decision-making. Often modeled as a Markov Decision Process (MDP), RL tries to obtain an optimal policy through sequential interactions with the environment. It finds applications in various fields, such as games (Silver et al., 2016; 2017; 2018; Vinyals et al., 2019), robotics (Kober et al., 2013; Gu et al., 2017), and autonomous driving (Yurtsever et al., 2020).

In this paper, we focus on the on-policy RL tailored for episodic tabular MDPs with inhomogeneous transition kernels. Specifically, the agent interacts with an episodic MDP consisting of $S$ states, $A$ actions, and $H$ steps per episode. The regret information bound for any MDP above and any learning algorithm with $K$ episodes is $O(\sqrt{H^2 SAT})$ where $T = KH$ denotes the total number of steps (Jin et al., 2018). Multiple RL algorithms in the literature (e.g. Zhang et al. (2020); Li et al. (2021); Zhang et al. (2024)) have reached a near-optimal $\sqrt{T}$-type regret that matches the information bound up to logarithmic factors, which acts as a worst-case guarantee.

In practice, RL algorithms often perform better than their worst-case guarantees, as such guarantees can be significantly improved under MDPs with benign structures (Zanette & Brunskill, 2019). This motivates the problem-dependent analysis for algorithms that exploit the benign MDPs (e.g., Wagenmaker et al. (2022a); Zhou et al. (2023); Zhang et al. (2024)). One of the benign structures is based on the dependency on the positive suboptimality gap: for every state, the best action outperforms others by a margin. It is important because nearly all non-degenerate environments with finite

---

[*]Z. Zheng and H. Zhang are co-first authors. L. Xue is the corresponding author.

action sets satisfy some sub-optimality gap conditions (Yang et al., 2021). Recently, Simchowitz & Jamieson (2019) proved the $\log T$-type regret if there exists a strictly positive suboptimality gap. Since then, the gap-dependent regret analysis has been widely studied, for example, Dann et al. (2021); Yang et al. (2021); Xu et al. (2021); Wang et al. (2022); He et al. (2021), etc.

Model-free RL algorithms, the focus of this paper, are also called $Q$-learning algorithms and directly learn the optimal action value function ($Q$-function) and state value function ($V$-function) to optimize the policy. It is widely used in practice due to its easy implementation (Jin et al., 2018) and the lower memory requirement that scales linearly in $S$ while that for model-based algorithms scales quadratically. However, the literature on gap-dependent analysis for $Q$-learning is quite sparse. Yang et al. (2021) studied the gap-dependent regret of the UCB-Hoeffding algorithm (Jin et al., 2018), the first model-free algorithm with a worst-case $\sqrt{T}$-type regret in the literature, and presented the first $\log T$-type regret bound for model-free algorithms:

$$O\left(\frac{H^6 SA \log(SAT)}{\Delta_{\min}}\right). \tag{1}$$

where $\Delta_{\min}$ is defined as the minimum nonzero suboptimality gap for all the state-action-step triples.

Xu et al. (2021) proposed the multi-step bootstrapping algorithm and showed the same dependency on the minimum gap as Yang et al. (2021). Both papers used the simple Hoeffding-type bonuses for explorations in the algorithm design. However, their analysis frameworks based on Hoeffding-type bonuses cannot be directly applied to study two important $Q$-learning algorithms that improve the regrets of Jin et al. (2018) and achieve the almost optimal worst-case regret: UCB-Advantage (Zhang et al., 2020) and Q-EarlySettled-Advantage (Li et al., 2021). In particular, UCB-Advantage and Q-EarlySettled-Advantage use variance estimators in their bonuses and reference-advantage decomposition for variance reduction. It remains an important open question whether such techniques can improve gap-dependent regret:

*Is it possible to establish a potentially improved gap-dependent regret bound for $Q$-learning using variance estimators in the bonuses and reference-advantage decomposition?*

This is a challenging task due to several non-trivial difficulties. In particular, bounding the weighted sum of the errors of the estimated $Q$-functions is necessary to establish the gap-dependent regret bounds for UCB-Advantage and Q-EarlySettled-Advantage, which is very difficult as it involves the estimated reference and advantage functions and the bonuses that include variance estimators for both functions. However, the analysis framework of Xu et al. (2021) for their non-optimism algorithm cannot bound the weighted sum of such errors, and the analysis frameworks in all optimism-based model-free algorithms including Jin et al. (2018); Zhang et al. (2020); Li et al. (2021); Yang et al. (2021) can only bound the weighted sum under the simple Hoeffding-type bonus.

Besides the regret, the policy switching cost is also an important evaluation criterion for on-policy RL, especially in applications with restrictions on policy switching such as compiler optimization (Ashouri et al., 2018), hardware placements (Mirhoseini et al., 2017), database optimization (Krishnan et al., 2018), and material discovery (Nguyen et al., 2019). Under the worst-case MDPs, Bai et al. (2019) modified the algorithms in Jin et al. (2018) to reach a switching cost of $O(H^3 SA \log T)$, and UCB-Advantage (Zhang et al., 2020) reached an improved switching cost of $O(H^2 SA \log T)$ due to the stage design in $Q$-function update, both improving upon the cost of $\Theta(K)$ for regular $Q$-learning algorithms (e.g. Jin et al. (2018)). To our knowledge, none of existing works study gap-dependent switching cost for $Q$-learning algorithms, leaving this as an open question..

**Summary of our contributions.** In this paper, we give an affirmative answer to the open questions above by establishing gap-dependent regret bound for UCB-Advantage (Zhang et al., 2020) and Q-EarlySettled-Advantage (Li et al., 2021) as well as a gap-dependent policy switching cost for UCB-Advantage. For $Q$-learning, this paper provides the first gap-dependent regret analysis with both variance estimators and variance reduction and the first gap-dependent policy switching cost.

Our detailed contributions are summarized as follows.

- **Improved Gap-Dependent Regret.** Denote $\mathbb{Q}^\star \in [0, H^2]$ as the *maximum conditional variance* for the MDP and $\beta \in (0, H]$ as the hyper-parameter to settle the reference function. We prove that

UCB-Advantage guarantees a gap-dependent expected regret of

$$O\left(\frac{\left(\mathbb{Q}^\star + \beta^2 H\right) H^3 SA \log(SAT)}{\Delta_{\min}} + \frac{H^8 S^2 A \log(SAT) \log(T)}{\beta^2}\right),$$ (2)

and Q-EarlySettled-Advantage guarantees a gap-dependent expected regret of

$$O\left(\frac{\left(\mathbb{Q}^\star + \beta^2 H\right) H^3 SA \log(SAT)}{\Delta_{\min}} + \frac{H^7 SA \log^2(SAT)}{\beta}\right).$$ (3)

These results are logarithmic in $T$ and better than the worst-case $\sqrt{T}$-type regret in Zhang et al. (2020); Li et al. (2021). They also have a common gap-dependent term $\tilde{O}\left(\left(\mathbb{Q}^\star + \beta^2 H\right) H^3 SA/\Delta_{\min}\right)$ where $\tilde{O}(\cdot)$ hides logarithmic factors. The other term in either Equation (2) or Equation (3) is gap-free. Our result is also better than Equation (1) for Yang et al. (2021); Xu et al. (2021) in the following ways. (a) Under the worst-case $\mathbb{Q}^\star = \Theta(H^2)$ and setting $\beta = O(1/\sqrt{H})$ as in Zhang et al. (2020) or $\beta = O(1)$ as in Li et al. (2021), $\tilde{O}\left(\left(\mathbb{Q}^\star + \beta^2 H\right) H^3 SA\right)/\Delta_{\min}$ becomes $\tilde{O}(H^5 SA/\Delta_{\min})$, which is better than Equation (1) by a factor of $H$. (b) Under the best variance $\mathbb{Q}^\star = 0$ which will happen when the MDP is deterministic, our regret in Equation (3) can linearly depend on $\tilde{O}(\Delta_{\min}^{-1/3})$, which is intrinsically better than the dependency on $\Delta_{\min}^{-1}$ in Equation (1). (c) Since our gap-free terms also logarithmically depend on $T$, they are smaller than Equation (1) when $\Delta_{\min}$ is sufficiently small.

- **Gap-Dependent Policy Switching Cost.** We can prove that for any $\delta \in (0,1)$, with probability at least $1 - \delta$, the policy switching cost for UCB-Advantage is at most

$$O\left(H|D_{\mathrm{opt}}| \log\left(\frac{T}{H|D_{\mathrm{opt}}|} + 1\right) + H|D_{\mathrm{opt}}^c| \log\left(\frac{H^4 SA^{\frac{1}{2}} \log(\frac{SAT}{\delta})}{\beta\sqrt{|D_{\mathrm{opt}}^c|\Delta_{\min}}}\right)\right).$$ (4)

Here, $D_{\mathrm{opt}}$ is a subset of all state-action-step triples and represents all triples such that the action is optimal. $D_{\mathrm{opt}}^c$ is its complement, and $|\cdot|$ gives the cardinality of the set. Next, we compare Equation (4) with the worst-case cost of $O(H^3 SA \log T)$ in Bai et al. (2019) and $O(H^2 SA \log T)$ in Zhang et al. (2020). Since $|D_{\mathrm{opt}}| < HSA$ for non-degenerate MDPs, our first term in Equation (4) is better than the worst-case cost. Specifically, when each state has a unique optimal action so that $|D_{\mathrm{opt}}| = HS$, it implies the improvement by removing a factor of $A$ compared with $O(H^2 SA \log T)$. This improvement is significant in applications with a large action space (e.g. recommender systems (Covington et al., 2016) and text-based games (Bellemare et al., 2013)). For the second term where $|D_{\mathrm{opt}}^c| < HSA$ in Equation (4), we also improve $\log T$ to $\log\log T$, and the significance of such improvement is pointed out by Qiao et al. (2022); Zhang et al. (2022b).

- **Technical Novelty and Contributions.**

For gap-dependent regret analysis, we develop an error decomposition framework that separates errors in reference estimations, advantage estimations, and reference settling. This helps bound the weighted sums mentioned above. We creatively handle the separated terms in the following way. (a) We relate the empirical errors and the bonus for reference estimations to $\mathbb{Q}^\star$ to avoid using their upper bounds $\Theta(H^2)$. This leverages the variance estimators. (b) When trying to bound the errors in reference and advantage estimations, we tackle the non-martingale difficulty, originating from the settled reference functions that depend on the whole learning process, with our novel surrogate reference functions so that the empirical estimations become martingale sums. To the best of our knowledge, we are the first to construct martingale surrogates in the literature for $Q$-learning using reference-advantage decomposition.

For the gap-dependent policy switching cost, we explore the unbalanced number of visits to states paired with optimal or suboptimal actions, which leads to the two terms in Equation (4).

## 2 PRELIMINARIES

Throughout this paper, for any $C \in \mathbb{N}$, we use $[C]$ to denote the set $\{1, 2, \ldots C\}$. We use $\mathbb{I}[x]$ to denote the indicator function, which equals 1 when the event $x$ is true and 0 otherwise.

**Tabular episodic Markov decision process (MDP).** A tabular episodic MDP is denoted as $\mathcal{M} := (\mathcal{S}, \mathcal{A}, H, \mathbb{P}, r)$, where $\mathcal{S}$ is the set of states with $|\mathcal{S}| = S$, $\mathcal{A}$ is the set of actions with $|\mathcal{A}| = A$, $H$ is the number of steps in each episode, $\mathbb{P} := \{\mathbb{P}_h\}_{h=1}^H$ is the transition kernel so that $\mathbb{P}_h(\cdot \mid s, a)$ characterizes the distribution over the next state given the state action pair $(s, a)$ at step $h$, and $r := \{r_h\}_{h=1}^H$ are the deterministic reward functions with $r_h(s, a) \in [0, 1]$.

In each episode, an initial state $s_1$ is selected arbitrarily by an adversary. Then, at each step $h \in [H]$, an agent observes a state $s_h \in \mathcal{S}$, picks an action $a_h \in \mathcal{A}$, receives the reward $r_h = r_h(s_h, a_h)$ and then transits to the next state $s_{h+1}$. The episode ends when an absorbing state $s_{H+1}$ is reached. Later on, for ease of presentation, when we describe $s, a, h, k$ along with "any, each, all" or "$\forall$", we will omit the sets $\mathcal{S}, \mathcal{A}, [H], [K]$. We denote $\mathbb{P}_{s,a,h} f = \mathbb{E}_{s_{h+1} \sim \mathbb{P}_h(\cdot|s,a)}(f(s_{h+1})|s_h = s, a_h = a)$, $\mathbb{V}_{s,a,h} f = \mathbb{P}_{s,a,h} f^2 - (\mathbb{P}_{s,a,h} f)^2$ and $\mathbb{1}_s f = f(s), \forall(s, a, h)$ for any function $f : \mathcal{S} \to \mathbb{R}$.

**Policies, state value functions, and action value functions.** A policy $\pi$ is a collection of $H$ functions $\left\{\pi_h : \mathcal{S} \to \Delta^{\mathcal{A}}\right\}_{h \in [H]}$, where $\Delta^{\mathcal{A}}$ is the set of probability distributions over $\mathcal{A}$. A policy is deterministic if for any $s \in \mathcal{S}$, $\pi_h(s)$ concentrates all the probability mass on an action $a \in \mathcal{A}$. In this case, we denote $\pi_h(s) = a$. We use $V_h^\pi : \mathcal{S} \to \mathbb{R}$ to denote the state value function at step $h$ under policy $\pi$. Mathematically, $V_h^\pi(s) := \sum_{h'=h}^H \mathbb{E}_{(s_{h'}, a_{h'}) \sim (\mathbb{P}, \pi)} [r_{h'}(s_{h'}, a_{h'}) \mid s_h = s]$. We also use $Q_h^\pi : \mathcal{S} \times \mathcal{A} \to \mathbb{R}$ to denote the state-action value function at step $h$, i.e., $Q_h^\pi(s, a) := r_h(s, a) + \sum_{h'=h+1}^H \mathbb{E}_{(s_{h'}, a_{h'}) \sim (\mathbb{P}, \pi)} [r_{h'}(s_{h'}, a_{h'}) \mid s_h = s, a_h = a]$. Azar et al. (2017) proved that there always exists an optimal policy $\pi^\star$ that achieves the optimal value $V_h^\star(s) = \sup_\pi V_h^\pi(s) = V_h^{\pi^*}(s)$ for all $s \in \mathcal{S}$ and $h \in [H]$. The Bellman equation and the Bellman optimality equation are

$$\begin{cases} V_h^\pi(s) = \mathbb{E}_{a' \sim \pi_h(s)}[Q_h^\pi(s, a')] \\ Q_h^\pi(s, a) := r_h(s, a) + \mathbb{P}_{s,a,h} V_{h+1}^\pi \\ V_{H+1}^\pi(s) = 0, \forall(s, a, h) \end{cases} \text{ and } \begin{cases} V_h^\star(s) = \max_{a' \in \mathcal{A}} Q_h^\star(s, a') \\ Q_h^\star(s, a) := r_h(s, a) + \mathbb{P}_{s,a,h} V_{h+1}^\star \\ V_{H+1}^\star(s) = 0, \forall(s, a, h). \end{cases} \quad (5)$$

For any problem with $K$ episodes, let $\pi^k$ be the policy adopted in the $k$-th episode, and $s_1^k$ be the corresponding initial state. The regret over $T = HK$ steps is $\text{Regret}(T) := \sum_{k=1}^K (V_1^\star - V_1^{\pi^k})(s_1^k)$.

**Suboptimality Gap.** For any given MDP, we can provide the following formal definition.

**Definition 2.1.** *For any $(s, a, h)$, the suboptimality gap is defined as $\Delta_h(s, a) := V_h^\star(s) - Q_h^\star(s, a)$.*

Equation (5) implies that $\Delta_h(s, a) \geq 0, \forall(s, a, h)$. Then it is natural to define the minimum gap, which is the minimum non-zero suboptimality gap with regard to all $(s, a, h)$.

**Definition 2.2.** *We define the **minimum gap** as $\Delta_{\min} := \inf\{\Delta_h(s, a) : \Delta_h(s, a) > 0, \forall(s, a, h)\}$.*

We remark that if $\{\Delta_h(s, a) : \Delta_h(s, a) > 0, \forall(s, a, h)\} = \emptyset$, then all actions are optimal, leading to a degenerate MDP. Therefore, we assume that the set is nonempty and $\Delta_{\min} > 0$. Definitions 2.1 and 2.2 and the non-degeneration are standard in the literature on gap-dependent analysis (e.g. Simchowitz & Jamieson (2019); Xu et al. (2020); Yang et al. (2021); Zhang et al. (2025)).

**Maximum Conditional Variance.** This quantity is formally defined as follows.

**Definition 2.3.** *We define the **maximum conditional variance** as $\mathbb{Q}^\star := \max_{s,a,h}\{\mathbb{V}_{s,a,h}(V_{h+1}^\star)\}$.*

Under our MDP with deterministic reward, Definition 2.3 coincides with that in (Zanette & Brunskill, 2019) which performed variance-dependent regret analysis.

**Policy Switching Cost.** We provide the following definition for any algorithm with $K > 1$ episodes.

**Definition 2.4.** *The policy switching cost for $K$ episodes is defined as $N_{\text{switch}} := \sum_{k=1}^{K-1} \tilde{N}_{\text{switch}}(\pi^{k+1}, \pi^k)$. Here, the $\tilde{N}_{\text{switch}}(\pi, \pi') := \sum_{s \in \mathcal{S}} \sum_{h=1}^H \mathbb{I}[\pi_h(s) \neq \pi_h'(s)]$ represents the local switching cost for any policies $\pi$ and $\pi'$.*

This definition is also used in Bai et al. (2019) and Zhang et al. (2020).

## 3 MAIN RESULTS

This section presents the gap-dependent regret for UCB-Advantage and Q-EarlySettled-Advantage in Subsection 3.1 and the gap-dependent policy switching cost for UCB-Advantage in Subsection 3.3. We highlight a new technical tool for the gap-dependent regret bound in Subsection 3.2.

## 3.1 GAP-DEPENDENT REGRETS

UCB-Advantage (Zhang et al., 2020) is the first model-free algorithm that reaches an almost optimal worst-case regret, which is also reached by Q-EarlySettled-Advantage (Li et al., 2021). Both algorithms are optimism-based, use upper confidence bounds (UCB) for exploration, and employ variance estimators and reference-advantage decomposition. UCB-Advantage settles the reference function at each $(s, h)$ by comparing the number of visits to a threshold that relies on a hyper-parameter $\beta \in (0, H]$. For readers' convenience, we provide UCB-Advantage without any modification in Algorithm 1 of Appendix D.1.

Theorem 3.1 provides the expected regret upper bound of UCB-Advantage.

**Theorem 3.1.** *For UCB-Advantage (Algorithm 1 in Appendix D.1) with $\beta \in (0, H]$, $\mathbb{E}[\mathrm{Regret}(T)]$ is upper bounded by Equation* (2).

Q-EarlySettled-Advantage improved the burn-in cost of Zhang et al. (2020) for reaching the almost-optimal worst-case regret by using both estimated upper and lower confidence bounds for $V_h^\star$ to settle the reference function. In this paper, we slightly modify its reference settling condition. At the end of $k$-th episode, for any $(s, h)$, the algorithm holds $V_h^{k+1}(s), V_h^{\mathrm{LCB},k+1}(s)$, the estimated upper and lower bounds for $V_h^\star(s)$, respectively. When $|V_h^{k+1}(s) - V_h^{\mathrm{LCB},k+1}(s)| \le \beta$ holds for the first time, it settles the reference function value $V_h^{\mathrm{R}}(s)$ as $V_h^{k+1}(s)$. Li et al. (2021) set $\beta = 1$ for worst-case MDPs. Our paper treats $\beta$ as a hyper-parameter within $(0, H]$ to allow better control over the learning process. Algorithms 2 and 3 provide our refined version. For the rest of this paper, we still call it Q-EarlySettled-Advantage without special notice.

Theorem 3.2 provides the expected regret upper bound of Q-EarlySettled-Advantage.

**Theorem 3.2.** *For Q-EarlySettled-Advantage (Algorithms 2 and 3 in Appendix F.1) with $\beta \in (0, H]$, $\mathbb{E}[\mathrm{Regret}(T)]$ is upper bounded by Equation* (3).

The proof sketch of Theorem 3.2 is presented in Section 4 to explain our technical contributions. The complete proofs of Theorems 3.1 and 3.2 are provided in Appendix D and Appendix F, respectively.

Next, we compare the results of both theorems with the worst-case regrets in Zhang et al. (2020); Li et al. (2021) and the gap-dependent regrets in Yang et al. (2021); Xu et al. (2021).

**Comparisons with Zhang et al. (2020); Li et al. (2021).** Since the regrets showed in Equations (2) and (3) are logarithmic in $T$, they are better than the worst-case regret $\tilde{O}(\sqrt{H^2 SAT})$ when $T \ge \tilde{\Theta}(\mathrm{poly}(HSA, \Delta_{\min}^{-1}, \beta^{-1}))$ where $\mathrm{poly}(\cdot)$ represents some polynomial. In addition, our results imply new guidance on setting the hyper-parameter $\beta$ for the gap-dependent regret, which is different from $\beta = 1/\sqrt{H}$ in Zhang et al. (2020) and $\beta = 1$ in Li et al. (2021), respectively. When $\mathbb{Q}^\star = 0$ which happens when the MDP is deterministic, if we set $\beta = \tilde{\Theta}(H(S\Delta_{\min})^{1/4})$ for UCB-Advantage and $\beta = \tilde{\Theta}(H\Delta_{\min}^{1/3})$, the gap-dependent regrets will linearly depend on $\Delta_{\min}^{-1/2}$ and $\Delta_{\min}^{-1/3}$, respectively. This provides new guidance on setting $\beta$ when we have prior knowledge about $\Delta_{\min}$. When $0 < \mathbb{Q}^\star \le H^2$, the best available gap-dependent regret becomes $\tilde{\Theta}(\mathbb{Q}^\star H^2 SA)$ which holds when $\beta \le \sqrt{\mathbb{Q}^\star/H}$. Knowing that the gap-free terms in Equations (2) and (3) monotonically decrease in $\beta$, we will recommend setting $\beta = \tilde{O}(\sqrt{\mathbb{Q}^\star/H})$ if prior knowledge on $\mathbb{Q}^\star$ is available.

**Comparisons with Yang et al. (2021); Xu et al. (2021).** The gap-dependent regret for Yang et al. (2021) is provided in Equation (1). For Xu et al. (2021), their regret bound is given by:

$$O\left(\left(\sum_{h=1}^{H} \sum_{s \in \mathcal{S}} \sum_{a \ne \pi_h^\star(s)} \frac{1}{\Delta_h(s, a)} + \frac{|Z_{\mathrm{mul}}|}{\Delta_{\min}} + SA\right) H^5 \log(K)\right), \tag{6}$$

where $Z_{\mathrm{mul}} = \{(h, s, a) | \Delta_h(s, a) = 0 \wedge |Z_{\mathrm{opt}}^h(s)| > 1\}$ and $Z_{\mathrm{opt}}^h(s) = \{a | \Delta_h(s, a) = 0\}$. In MDPs where $\Delta_h(s, a) = \Theta(\Delta_{\min})$ for $\Theta(HSA)$ state-action-step triples (e.g. the example in Xu et al. (2021, Theorem 1.3)) or there are $\Theta(A)$ optimal actions for each state-step pair $(s, h)$, their regret reduces to Equation (1), which is worse than ours.

Next, we compare Equations (2) and (3) with Equation (1). Under the worst-case variance $\mathbb{Q}^\star = \Theta(H^2)$ and letting $\beta$ be $\Theta(1/\sqrt{H})$ or $\Theta(1)$ which are the recommendations in Zhang et al. (2020); Li et al. (2021) respectively for the worst-case MDPs, the common gap-dependent term Equations (2) and (3) becomes $\tilde{O}(H^5 SA/\Delta_{\min})$, which is better than Equation (1) by a factor of $H$. Under the best variance $\mathbb{Q}^\star = 0$, the gap-dependent term becomes $\tilde{O}(\beta^2 H^3 SA/\Delta_{\min})$, which is better than Equation (1) for any $\beta \in (0, H]$. In addition, our best possible gap-dependent regret that is sublinear in $\Delta_{\min}^{-1}$ is also intrinsically better. Here, we remark that the proof in Yang et al. (2021); Xu et al. (2021) cannot benefit from $\mathbb{Q}^\star = 0$ due to their use of Hoeffding-type bonuses.

We also comment on the gap-free terms in Equations (2) and (3). They are dominated by the gap-dependent term as long as $\Delta_{\min} \leq \tilde{O}(\text{poly}((HSA)^{-1}, \beta))$ for some polynomial $\text{poly}(\cdot)$. In addition, the gap-free term in Equation (3) is linear in $S$, which is better than that for Equation (2) thanks to the special design of Q-EarlySettled-Advantage algorithm. It utilizes both upper confidence bounds and lower confidence bounds for $V$-functions to settle the reference function.

## 3.2 OUR TECHNICAL TOOL: SURROGATE REFERENCE FUNCTIONS

We develop a new technical tool in the proofs of both Theorems 3.1 and 3.2: the surrogate reference functions. In this subsection, we explain it with the notations in the proof of Theorem 3.2 (Appendix F.1) for Q-EarlySettled-Advantage while all the ideas also apply to UCB-Advantage. A more detailed proof sketch will be provided in the next section. For a comprehensive explanation of Q-EarlySettled-Advantage, we refer readers to Appendix F.1.

Before introducing the surrogate reference function, we provide a brief overview of the key steps of Q-EarlySettled-Advantage. Denote the estimated $Q$-function, the estimated $V$-function, and the reference function before the start of episode $k$ as $Q_h^k(s,a), V_h^k(s), V_h^{\mathrm{R},k}(s)$ and episode $k$ as $\{(s_h^k, a_h^k)\}_{h=1}^H$. Let $N_h^k(s,a)$ be the number of visits to $(s,a,h)$ before the start of episode $k$. Let $N_h^{k+1}$ be short for $N_h^{k+1}(s_h^k, a_h^k)$ and $k^n$ be the episode index for the $n$-th visit to $(s_h^k, a_h^k, h)$. While remaining unchanged for the unvisited triples, the estimated $Q$-function is updated on the visited ones:

$$Q_h^{k+1}(s_h^k, a_h^k) = \min\{Q_h^{\mathrm{UCB},k+1}(s_h^k, a_h^k), Q_h^{\mathrm{R},k+1}(s_h^k, a_h^k), Q_h^k(s_h^k, a_h^k)\}, h \in [H]. \tag{7}$$

Here, $Q_h^{\mathrm{UCB},k+1}$ represents the Hoeffding-type estimation similar to Jin et al. (2018), and $Q_h^{\mathrm{R},k+1}(s_h^k, a_h^k)$ represents the reference-advantage type estimation as follows:

$$Q_h^{\mathrm{R},k+1}(s_h^k, a_h^k) = r_h^k(s_h^k, a_h^k) + \sum_{n=1}^{N_h^{k+1}} \left( \eta_n^{N_h^{k+1}} (V_{h+1}^{k^n} - V_{h+1}^{\mathrm{R},k^n}) + u_n^{N_h^{k+1}} V_{h+1}^{\mathrm{R},k^n} \right)(s_{h+1}^{k^n}) + \tilde{R}^{h,k+1}. \tag{8}$$

In Equation (8), $V_{h+1}^{k^n} - V_{h+1}^{\mathrm{R},k^n}$ represents the running estimation of the advantage function, and $\{\eta_n^{N_h^k}\}_{n=1}^{N_h^{k+1}}$ are the corresponding nonnegative weights that sum to 1. $\{u_n^{N_h^{k+1}}\}_{n=1}^{N_h^{k+1}}$ that sum to 1 are nonnegative weights for the reference function. $\tilde{R}^{h,k+1}$ is the cumulative bonus that dominates the variances in the two weighted sums. Next, the estimated $V$-function and the reference function are also updated. For any $(s,h)$, when some reference settling condition related to $\beta$ is triggered at the end of episode $k$, the reference function will be settled, which means that $V_h^{\mathrm{R},k'}(s) = V_h^{\mathrm{R},k+1}(s)$ for any $k' \geq k+1$. Thus, we call $V_h^{\mathrm{R},K+1}$, the reference function after the last episode as the settled reference function. Q-EarlySettled-Advantage guarantees that for any $(h,k) \in [H] \times [K]$

$$V_h^k(s) = \max_a Q_h^k(s,a), \ \pi_h^k(s) = \arg\max_a Q_h^k(s,a), \tag{9}$$

and

$$Q_h^\star \leq Q_h^{k+1} \leq Q_h^k \leq H, V_h^{k+1} \leq V_h^k \leq H, \ V_h^{\mathrm{R},k+1} \leq V_h^{\mathrm{R},k} \leq H, \ V_h^\star \leq V_h^k \leq V_h^{\mathrm{R},k}. \tag{10}$$

Equations (9) and (10) indicate that Q-EarlySettled-Advantage is an optimism-based method that updates the policy according to an upper bound $Q_h^k$ of $Q_h^\star$.

Next, we introduce our surrogate reference functions $\hat{V}_h^{\mathrm{R},k}$. They are defined as follows:

$$\hat{V}_h^{\mathrm{R},k}(s) := \max\left\{V_h^\star(s), \min\{V_h^\star(s) + \beta, V_h^{\mathrm{R},k}(s)\}\right\}, \forall(s,h,k). \tag{11}$$

We use the word "surrogate" because the algorithm does not rely on it, and $\hat{V}_h^{\mathrm{R},k}$ differs from the actual settled reference function $V_h^{\mathrm{R},K+1}$. $\hat{V}_h^{\mathrm{R},k}$ is determined before episode $k$. In addition, Equation (10) implies that

$$V_h^\star(s) \le \hat{V}_h^{\mathrm{R},k}(s) = \min\left\{V_h^\star(s) + \beta, V_h^{\mathrm{R},k}(s)\right\}, \forall(s,h,k), \tag{12}$$

and Lemma F.5 in Appendix F.2 shows that with high probability, $\hat{V}_h^{\mathrm{R},k}(s)$ coincides with the settled reference value $V_h^{\mathrm{R},K+1}(s)$ after the settling condition is triggered.

Next, we discuss the usage of $\hat{V}_h^{\mathrm{R},k}$ in our error decompositions. Our proof relies on relating the regret to multiple groups of estimation error sums that take the form $\sum_{k=1}^K \omega_{h,k}^{(i)}(Q_h^k - Q_h^\star)(s_h^k, a_h^k)$. Here $\{\omega_{h,k}^{(i)}\}_k$ are nonnegative weights and $i$ is the group index. Bounding the weighted sum via controlling each individual $(Q_h^k - Q_h^\star)(s_h^k, a_h^k)$ by recursion on $h$ is a common technique for model-free optimism-based algorithms, and it is also used by all of Yang et al. (2021); Zhang et al. (2020); Li et al. (2021). Yang et al. (2021) used it on gap-dependent regret analysis and Zhang et al. (2020); Li et al. (2021) used it to control the reference settling errors $\sum_{k=1}^K \left(V_h^{\mathrm{R},k+1} - V_h^{\mathrm{R},K+1}\right)(s_h^k)$. However, their techniques are only limited to the Hoeffding-type update, where the errors generated in the recursion take the simple form of $\tilde{O}\left(\sqrt{H^3/N_h^k}\right)$, where $N_h^k$ is short for $N_h^k(s_h^k, a_h^k)$. When analyzing the reference-advantage type update, we face a more complicated error (Equation (14) in Section 4) involving reference and advantage estimations, as well as bonuses with variance estimators.

Motivated by the structure of reference-advantage decomposition, we decompose the estimation error into several components, focusing on the following four main terms: $\mathcal{G}_1 := \sum_{n=1}^{N_h^k} \eta_n^{N_h^k}(\mathbb{P}_{s_h^k, a_h^k, h} - \mathbb{1}_{s_{h+1}^{k^n}})(\hat{V}_{h+1}^{\mathrm{R},k^n} - V_{h+1}^\star)$, $\mathcal{G}_2 := \sum_{n=1}^{N_h^k} u_n^{N_h^k}\left(\mathbb{1}_{s_{h+1}^{k^n}} - \mathbb{P}_{s_h^k, a_h^k, h}\right)\hat{V}_{h+1}^{\mathrm{R},k^n}$, $\mathcal{G}_3 := \sum_{n=1}^{N_h^k} \left(u_n^{N_h^k} - \eta_n^{N_h^k}\right)\mathbb{P}_{s_h^k, a_h^k, h}\hat{V}_{h+1}^{\mathrm{R},k^n} + \sum_{n=1}^{N_h^k} u_n^{N_h^k}(V_{h+1}^{\mathrm{R},k^n} - \hat{V}_{h+1}^{\mathrm{R},k^n})(s_{h+1}^{k^n})$ and the bonus term $\mathcal{G}_4$. The first three terms correspond to advantage estimation error, reference estimation error, and reference settling error, respectively. Here, we creatively use the surrogate $\hat{V}_{h+1}^{\mathrm{R},k}$ as it is determined before the start of episode $k$. Thus, $\mathcal{G}_1, \mathcal{G}_2$ are martingale sums and can be controlled by concentration inequalities. $\mathcal{G}_3$ corresponds to the reference settling error and can also be well-controlled given the settling conditions and properties of $\hat{V}_h^{\mathrm{R},k}(s)$. $\mathcal{G}_4$ is controlled using the same idea of bounding $\mathcal{G}_1, \mathcal{G}_2, \mathcal{G}_3$. $\hat{V}_{h+1}^{\mathrm{R},k}$ is crucial to this process and cannot be replaced by the actual settled reference function $V_{h+1}^{\mathrm{R},K+1}$ used in Zhang et al. (2020); Li et al. (2021). This is because $V_{h+1}^{\mathrm{R},K+1}$ depends on the whole learning process and causes a non-martingale issue in controlling $\mathcal{G}_1, \mathcal{G}_2$ and $\mathcal{G}_3$. To the best of our knowledge, we are the first to introduce the novel construction of reference surrogates for reference-advantage decomposition, which is of independent interest for future research on off-policy and offline methods.

### 3.3 GAP-DEPENDENT POLICY SWITCHING COST FOR UCB-ADVANTAGE

Different from Q-EarlySettled-Advantage, UCB-Advantage uses the stage design for updating the estimated $Q$-function. For each $(s, a, h)$, Zhang et al. (2020) divided the visits into consecutive stages with the stage size increasing exponentially. It updates the estimated $Q$-function only at the end of each stage so that the policy switches infrequently. Theorem 3.3 provides the policy switching cost for UCB-Advantage, and the proof is provided in Appendix E.

**Theorem 3.3.** *For UCB-Advantage (Algorithm 1 in Appendix D.1) with $\beta \in (0, H]$ and any $\delta \in (0, 1)$, with probability at least $1 - \delta$, $N_{\mathrm{switch}}$ is upper bounded by Equation* (4). *Here, $D_{\mathrm{opt}} = \{(s, a, h) \in \mathcal{S} \times \mathcal{A} \times [H] \mid a \in \mathcal{A}_h^\star(s)\}$, where $\mathcal{A}_h^*(s) = \{a \mid a = \arg\max_{a'} Q_h^*(s, a')\}$, and $D_{\mathrm{opt}}^c = (\mathcal{S} \times \mathcal{A} \times [H]) \backslash D_{\mathrm{opt}}$.*

**Comparisons with existing works.** The first term in Equation (4) logarithmically depends on $T$ and the second one logarithmically depends on $1/\Delta_{\min}$ and $\log T$. Next, we compare our result with $O(H^2 SA \log T)$ in Zhang et al. (2020), which is the best available policy switching cost for model-free methods in the literature. For the first term in Equation (4), knowing that $|D_{\mathrm{opt}}| < HSA$ for all non-degenerated MDPs where there exists at least one state such that not all actions are optimal, the coefficient is better than Equation (4). Specifically, if each state has a unique optimal action so

that $|D_{\text{opt}}| = SH$, Equation (4) becomes $O\Big(H^2 S \log\big(\frac{T}{H^2 S} + 1\big) + H^2 SA \log\big(\frac{H^{\frac{7}{2}} S^{\frac{1}{2}} \log(\frac{SAT}{\delta})}{\beta \Delta_{\min}}\big)\Big)$ where the coefficient in the first term outperforms that in Zhang et al. (2020) by a factor of $A$.

For the second term in Equation (4), when the total steps are sufficiently large such that $T = \tilde{\Omega}\big(\text{poly}\big(SH, (\beta\Delta_{\min})^{-1}\big)\big)$ for some polynomial $\text{poly}(\cdot)$, it is also better than $O(H^2 SA \log T)$.

**Key Ideas of the Proof.** The proof of Theorem 2 in Zhang et al. (2020) implies $N_{\text{switch}} \leq \sum_{s,a,h} 4H \log\Big(\frac{N_h^{K+1}(s,a)}{2H} + 1\Big)$, where $N_h^{K+1}(s,a)$ is the total number of visits to $(s,a,h)$. Under their worst-case MDP and noticing that $\sum_{s,a,h} N_h^{K+1}(s,a) \leq T$, Zhang et al. (2020) further proved their bound $O(H^2 SA \log T)$ by applying Jensen's inequality. In our gap-dependent analysis, Equation (79) in Appendix E shows that with high probability, $\sum_{(s,a,h) \in D_{\text{opt}}^c} N_h^{K+1}(s,a) \leq \tilde{O}\big(\frac{H^6 SA}{\Delta_{\min}} + \frac{H^8 S^2 A}{\beta^2}\big)$, which is much smaller than $T$ when $T$ is sufficiently large. This implies the discrepancy among the number of visits to state-action-step triples with optimal or suboptimal actions. Accordingly, we prove the bound in Equation (4) by using Jensen's inequality separately for triples with optimal or suboptimal actions.

## 4 PROOF SKETCH OF THEOREM 3.2

This section provides a proof sketch to outline the key steps for proving Theorem 3.2 on the gap-dependent regret of Q-EarlySettled-Advantage and explain our technical contributions. The key steps for proving Theorem 3.1 are similar except for different bounds on reference settling error and gap-free regret terms. For space consideration, the proofs of Theorem 3.1 are given in Appendix D.

**Notations.** First, we introduce the weights used in the algorithm. Let $\eta_n := \frac{H+1}{H+n}$. For $N \in \mathbb{N}_+$, denote $\eta_0^0 := 1$ and $\eta_0^N := \prod_{i=1}^N (1 - \eta_i)$. For integers $1 \leq n \leq N$, we also denote $\eta_n^N := \eta_n \prod_{i=n+1}^N (1 - \eta_i)$, and $u_n^N = \sum_{i=n}^N \eta_i^N / i$. When $N > 0$, they satisfy $1 - \eta_0^N = \sum_{n=1}^N \eta_n^N = \sum_{n=1}^N u_n^N$. For simplicity later, we use the notations $\hat{\mathbb{E}}_{h,k}^{\text{ref}} f := \sum_{n=1}^{N_h^k} u_n^{N_h^k} f(s_{h+1}^{k^n})$ and $\hat{\mathbb{E}}_{h,k}^{\text{ref}} f^{k^n} := \sum_{n=1}^{N_h^k} u_n^{N_h^k} f^{k^n}(s_{h+1}^{k^n})$ for any functions $f : \mathcal{S} \to \mathbb{R}$ and $f^k : \mathcal{S} \to \mathbb{R}$ with $k \in \mathbb{N}_+$, respectively. Similarly, we denote $\hat{\mathbb{E}}_{h,k}^{\text{adv}} f := \sum_{n=1}^{N_h^k} \eta_n^{N_h^k} f(s_{h+1}^{k^n})$ and $\hat{\mathbb{E}}_{h,k}^{\text{adv}} f^{k^n} := \sum_{n=1}^{N_h^k} \eta_n^{N_h^k} f^{k^n}(s_{h+1}^{k^n})$. We also denote $\mathbb{P}_{h,k}^{\text{ref}} f = \sum_{n=1}^{N_h^k} u_n^{N_h^k} \mathbb{P}_{s_h^k, a_h^k, h} f$, $\mathbb{P}_{h,k}^{\text{ref}} f^{k^n} = \sum_{n=1}^{N_h^k} u_n^{N_h^k} \mathbb{P}_{s_h^k, a_h^k, h} f^{k^n}$, $\mathbb{P}_{h,k}^{\text{adv}} f = \sum_{n=1}^{N_h^k} \eta_n^{N_h^k} \mathbb{P}_{s_h^k, a_h^k, h} f$ and $\mathbb{P}_{h,k}^{\text{adv}} f^{k^n} = \sum_{n=1}^{N_h^k} \eta_n^{N_h^k} \mathbb{P}_{s_h^k, a_h^k, h} f^{k^n}$.

In what follows, we present the proof sketch of Theorem 3.2.

**Step 1: Bounding $Q_h^k - Q_h^\star$ via decomposition and the surrogate reference function.** The update of the estimated $Q$-function in Equations (7) and (8) guarantees that

$$Q_h^k(s_h^k, a_h^k) \leq \eta_0^{N_h^k} H + r_h(s_h^k, a_h^k) + \hat{\mathbb{E}}_{h,k}^{\text{adv}}(V_{h+1}^{k^n} - V_{h+1}^{\text{R},k^n}) + \hat{\mathbb{E}}_{h,k}^{\text{ref}} V_{h+1}^{\text{R},k^n} + R^{h,k}. \quad (13)$$

Here, $R^{h,k}$ is the cumulative bonus provided in Equation (98) in Appendix F.3.2. Together with $Q_h^\star(s_h^k, a_h^k) \geq r_h(s_h^k, a_h^k) + (1 - \eta_0^{N_h^k}) \mathbb{P}_{s_h^k, a_h^k, h} V_{h+1}^\star$ by Equation (5) and $\hat{\mathbb{E}}_{h,k}^{\text{adv}}(V_{h+1}^{k^n} - V_{h+1}^{\text{R},k^n}) \leq \hat{\mathbb{E}}_{h,k}^{\text{adv}}(V_{h+1}^{k^n} - \hat{V}_{h+1}^{\text{R},k^n})$ implied by Equation (12), we have

$$(Q_h^k - Q_h^\star)(s_h^k, a_h^k) \leq \eta_0^{N_h^k} H + R^{h,k} + \hat{\mathbb{E}}_{h,k}^{\text{adv}}(V_{h+1}^{k^n} - \hat{V}_{h+1}^{\text{R},k^n}) + \hat{\mathbb{E}}_{h,k}^{\text{ref}} V_{h+1}^{\text{R},k^n} - \mathbb{P}_{h,k}^{\text{adv}} V_{h+1}^\star =: G_h^k.$$

Denote $\hat{V}_h^{\text{adv},k} = \hat{V}_h^{\text{R},k} - V_h^\star$, then:

$$G_h^k = \hat{\mathbb{E}}_{h,k}^{\text{adv}}(V_{h+1}^{k^n} - V_{h+1}^\star) + (\mathbb{P}_{h,k}^{\text{adv}} - \hat{\mathbb{E}}_{h,k}^{\text{adv}})\hat{V}_{h+1}^{\text{adv},k^n} + (\hat{\mathbb{E}}_{h,k}^{\text{ref}} - \mathbb{P}_{h,k}^{\text{ref}})\hat{V}_{h+1}^{\text{R},k^n} + R^{h,k} + R_{\text{else},0}^{h,k}. \quad (14)$$

Here, $R_{\text{else},0}^{h,k} = H\eta_0^{N_h^k} + \hat{\mathbb{E}}_{h,k}^{\text{ref}}(V_{h+1}^{\text{R},k^n} - \hat{V}_{h+1}^{\text{R},k^n}) + (\mathbb{P}_{h,k}^{\text{ref}} \hat{V}_{h+1}^{\text{R},k^n} - \mathbb{P}_{h,k}^{\text{adv}} \hat{V}_{h+1}^{\text{R},k^n})$. Equation (95) and Equation (96) in Appendix F.3.1 show that for all $(k,h)$ simultaneously, with high probability,

$$(\mathbb{P}_{h,k}^{\text{adv}} - \hat{\mathbb{E}}_{h,k}^{\text{adv}})\hat{V}_{h+1}^{\text{adv},k^n} \leq \tilde{O}\left(\sqrt{\frac{H\beta^2}{N_h^k}}\right), \quad (\hat{\mathbb{E}}_{h,k}^{\text{ref}} - \mathbb{P}_{h,k}^{\text{ref}})\hat{V}_{h+1}^{\text{R},k^n} \leq \tilde{O}\left(\sqrt{\frac{\mathbb{Q}^\star + \beta^2}{N_h^k}} + \frac{H}{N_h^k}\right). \quad (15)$$

Equation (15) corresponds to controlling $\mathcal{G}_1$ and $\mathcal{G}_2$, as discussed in Section 3.2, and holds because our surrogate reference function adapts to the learning process. To bound the bonus $R^{h,k}$, we also use the surrogate function $\hat{V}_h^{\mathrm{R},k}$. Equation (103) in Appendix F.3.2 shows that for all $(k,h)$ simultaneously, with high probability

$$R^{h,k} \le \tilde{O}\left(\sqrt{(\mathbb{Q}^\star + \beta^2 H)/N_h^k} + H^2/(N_h^k)^{\frac{3}{4}} + \sqrt{H\Psi_h^k/N_h^k}\right). \tag{16}$$

where $\Psi_h^k = \sum_{n=1}^{N_h^k}\left(V_{h+1}^{\mathrm{R},k^n} - \hat{V}_{h+1}^{\mathrm{R},k^n}\right)(s_{h+1}^{k^n})$. Equations (14) to (16) imply

$$(Q_h^k - Q_h^\star)(s_h^k, a_h^k) \le \hat{\mathbb{E}}_{h,k}^{\mathrm{adv}}\left(V_{h+1}^{k^n} - V_{h+1}^\star\right) + \tilde{O}\left(\sqrt{(\mathbb{Q}^\star + H\beta^2)/N_h^k} + H^2(N_h^k)^{-\frac{3}{4}}\right) + R_{\mathrm{else}}^{h,k}, \tag{17}$$

where $R_{\mathrm{else}}^{h,k} = \tilde{O}\left(\eta_0^{N_h^k} H + \hat{\mathbb{E}}_{h,k}^{\mathrm{ref}}\left(V_{h+1}^{\mathrm{R},k^n} - \hat{V}_{h+1}^{\mathrm{R},k^n}\right) + \left(\mathbb{P}_{h,k}^{\mathrm{ref}}\hat{V}_{h+1}^{\mathrm{R},k^n} - \mathbb{P}_{h,k}^{\mathrm{adv}}\hat{V}_{h+1}^{\mathrm{R},k^n}\right) + \left(\sqrt{H\Psi_h^k} + H\right)/N_h^k\right)$.

*Remark 1*: We use $\mathbb{Q}^\star$ in Equation (16) instead of its upper bound $\Theta(H^2)$ thanks to the variance estimator (line 16 of Algorithm 2 in Appendix F.1 ) used in Q-EarlySettled-Advantage algorithm.

**Step 2: Bounding the Weighted Sum.** For any given $h \in [H]$ and non-negative weights $\{\omega_{h,k}\}_{h,k\in[K]}$, we denote $\|\omega\|_{\infty,h} = \max_{k\in[K]}\omega_{h,k}$ and $\|\omega\|_{1,h} = \sum_{k\in[K]}\omega_{h,k}$. We also recursively define $\omega_{h',k}(h)$ for any $h \le h' < H, k \in [K]$ as follows:

$$\omega_{h,k}(h) := \omega_{h,k}; \quad \omega_{h'+1,j}(h) = \sum_{k=1}^K \sum_{n=1}^{N_{h'}^k} \omega_{h',k}(h)\eta_n^{N_{h'}^k}\mathbb{I}\left[k^n = j\right], \forall j \in [K], \; h \le h' < H. \tag{18}$$

Equation (18) implies the mapping from $\{\omega_{h,k}\}_{h,[K]}$ to $\{\omega_{h',k}(h)\}_{h',[K]}$ is linear. Let $\|\omega(h)\|_{\infty,h'} = \max_{k\in[K]}\omega_{h',k}(h)$ and $\|\omega(h)\|_{1,h'} = \sum_{k\in[K]}\omega_{h',k}(h)$. Then Equation (106) and Equation (107) in Appendix F.4.2 shows that

$$\|\omega(h)\|_{\infty,h'} \le (1 + 1/H)\|\omega(h)\|_{\infty,h'-1}, \; \|\omega(h)\|_{1,h'} \le \|\omega(h)\|_{1,h'-1}, \; \forall h' > h. \tag{19}$$

Next, we bound the weighted sum $\sum_{k=1}^K \omega_{h,k}(Q_h^k - Q_h^\star)(s_h^k, a_h^k)$. In Equation (17) where we take summations with regard to $k$ on both sides and apply the standard summation rearrangement technique given in Appendix F.4.1 to the first term $\hat{\mathbb{E}}_{h,k}^{\mathrm{adv}}\left(V_{h+1}^{k^n} - V_{h+1}^\star\right)$, we have $\sum_{k=1}^K \omega_{h,k}(Q_h^k - Q_h^\star)(s_h^k, a_h^k) \le \sum_{k=1}^K \omega_{h+1,k}(h)(Q_{h+1}^k - Q_{h+1}^\star)(s_{h+1}^k, a_{h+1}^k) + \sum_{k=1}^K \omega_{h,k}\tilde{O}\left(\sqrt{(\mathbb{Q}^\star + H\beta^2)/N_h^k} + H^2(N_h^k)^{-\frac{3}{4}}\right) + \sum_{k=1}^K \omega_{h,k}R_{\mathrm{else}}^{h,k}$. Recurring it with regard to $h, h+1\ldots, H$, we have

$$\sum_{k=1}^K \omega_{h,k}(Q_h^k - Q_h^\star)(s_h^k, a_h^k) \le R_c + \sum_{k=1}^K \sum_{h'=h}^H \omega_{h',k}(h)R_{\mathrm{else}}^{h',k}. \tag{20}$$

where $R_c = \sum_{k=1}^K \sum_{h'=h}^H \omega_{h',k}(h)\tilde{O}\left(\sqrt{(\mathbb{Q}^\star + H\beta^2)/N_h^k} + H^2(N_h^k)^{-3/4}\right)$. From Equation (19) and Lemma F.3 in Appendix F.2, it follows that

$$R_c \le \tilde{O}\left(H\sqrt{\mathbb{Q}^\star + \beta^2 H}\sqrt{SA\|\omega\|_{\infty,h}\|\omega\|_{1,h}} + H^3(SA\|\omega\|_{\infty,h})^{\frac{3}{4}}\|\omega\|_{1,h}^{\frac{1}{4}}\right). \tag{21}$$

**Step 3: Integrating Multiple Weighted Sums.** Next, we consider multiple groups of weights. We split the interval $[\Delta_{\min}, H]$ into $N$ disjoint intervals $\mathcal{I}_i := [2^{i-1}\Delta_{\min}, 2^i\Delta_{\min})$ for $i \in [N-1]$ and $\mathcal{I}_N := [2^{N-1}\Delta_{\min}, H]$. Here, $N = \lceil\log_2(H/\Delta_{\min})\rceil$. For any given $i \in [N]$ and $h \in [H]$, we denote $\omega_{h,k}^{(i)} = \mathbb{I}\left[(Q_h^k - Q_h^\star)(s_h^k, a_h^k) \in \mathcal{I}_i\right]$. Then we have $\|\omega^{(i)}\|_{\infty,h} = \max_{k\in[K]}\omega_{h,k}^{(i)} \le 1$ and

$$2^{i-1}\Delta_{\min}\|\omega^{(i)}\|_{1,h} \le \sum_{k=1}^K \omega_{h,k}^{(i)}(Q_h^k - Q_h^\star)(s_h^k, a_h^k) \le 2^i\Delta_{\min}\|\omega^{(i)}\|_{1,h}, \tag{22}$$

where $\|\omega^{(i)}\|_{1,h} = \sum_{k\in[K]}\omega_{h,k}^{(i)}$. Noticing that $\sum_{i=1}^N \sum_{k=1}^K \omega_{h,k}^{(i)}(Q_h^k - Q_h^\star)(s_h^k, a_h^k) = \sum_{k=1}^K \mathrm{clip}[(Q_h^k - Q_h^\star)(s_h^k, a_h^k)|\Delta_{\min}]$ where $\mathrm{clip}[x|\delta] := x\mathbb{I}[x \ge \delta]$, Equation (22) further implies

$$\sum_{k=1}^K \mathrm{clip}[(Q_h^k - Q_h^\star)(s_h^k, a_h^k) \mid \Delta_{\min}] = \Theta\left(\sum_{i=1}^N 2^i\Delta_{\min}\|\omega^{(i)}\|_{1,h}\right). \tag{23}$$

Letting $\omega_{h,k} = \omega_{h,k}^{(i)}$ in Equation (20) and applying Equations (21) and (22), we have

$$2^{i-1}\Delta_{\min}\|\omega^{(i)}\|_{1,h} \leq \tilde{O}\left(\theta_1\sqrt{\|\omega^{(i)}\|_{1,h}} + \theta_2\|\omega^{(i)}\|_{1,h}^{\frac{1}{4}} + \sum_{k=1}^{K}\sum_{h'=h}^{H}\omega_{h',k}^{(i)}(h)R_{\text{else}}^{h',k}\right). \quad (24)$$

Here, $\theta_1 = \sqrt{H^2SA(\mathbb{Q}^\star + \beta^2 H)}, \theta_2 = H^3(SA)^{\frac{3}{4}}$. The weight $\{\omega_{h',k}^{(i)}(h)\}_k$ is defined recursively by Equation (18) with $\omega_{h,k}^{(i)}(h) = \omega_{h,k}^{(i)}$. Solving this inequality (see Equation (112)), we have

$$\|\omega^{(i)}\|_{1,h} \leq \tilde{O}\left(\frac{(\mathbb{Q}^\star + \beta^2 H)SAH^2}{4^{i-1}\Delta_{\min}^2} + \frac{H^4SA}{(2^{i-1}\Delta_{\min})^{\frac{4}{3}}} + \frac{\sum_{k=1}^{K}\sum_{h'=h}^{H}\omega_{h',k}^{(i)}(h)R_{\text{else}}^{h',k}}{2^{i-1}\Delta_{\min}}\right).$$

This further implies

$$\sum_{i=1}^{N}2^i\Delta_{\min}\|\omega^{(i)}\|_{1,h} \leq \tilde{O}\left(\frac{(\mathbb{Q}^\star + \beta^2 H)SAH^2}{\Delta_{\min}} + \frac{H^4SA}{\Delta_{\min}^{\frac{1}{3}}} + \sum_{k=1}^{K}\sum_{h'=h}^{H}\hat{\omega}_{h',k}(h)R_{\text{else}}^{h',k}\right). \quad (25)$$

where $\hat{\omega}_{h',k}(h) = \sum_{i=1}^{N}\omega_{h',k}^{(i)}(h)$. Noticing that $\hat{\omega}_{h,k}(h) = \mathbb{I}[(Q_h^k - Q_h^\star)(s_h^k, a_h^k) \geq \Delta_{\min}]$, together with the linearity showed in Equation (18), Equation (19) implies $\hat{\omega}_{h',k}(h) \leq O(1)$ for any $h \leq h' \leq H$. Thus, $\sum_{k=1}^{K}\sum_{h'=h}^{H}\hat{\omega}_{h',k}(h)R_{\text{else}}^{h',k} \leq O(\sum_{k=1}^{K}\sum_{h'=1}^{H}R_{\text{else}}^{h',k})$. Appendix F.5.2 shows that with high probability,

$$\sum_{k=1}^{K}\sum_{h'=1}^{H}R_{\text{else}}^{h',k} \leq \tilde{O}(H^6SA/\beta). \quad (26)$$

Summarizing Equations (23), (25) and (26) and noticing that $H^4SA/\Delta_{\min}^{\frac{1}{3}} \leq O(\beta^2 H^3SA/\Delta_{\min} + H^4SA/\beta + H^5SA/\beta)$ that follows from the AM–GM inequality, we have

$$\sum_{k=1}^{K}\text{clip}[(Q_h^k - Q_h^\star)(s_h^k, a_h^k) \mid \Delta_{\min}] = \tilde{O}\left(SAH^2(\mathbb{Q}^\star + \beta^2 H)/\Delta_{\min} + H^6SA/\beta\right). \quad (27)$$

*Remark 2*: Integrating groups of sums is first introduced in Yang et al. (2021) and also applied in Li et al. (2021). It leads to regret dependency on $1/\Delta_{\min}$ instead of $1/\Delta_{\min}^2$ that will appear if we do not use integration. We extend this method in handling $R_{\text{else}}^{h,k}$ that only appears in our proof: we apply the upper bound in Equation (26) after the integration instead of Equation (24) before the integration. This helps us remove the dependency on $\Delta_{\min}$ in the second term in Equation (27).

*Remark 3*: Equation (26) can be interpreted as bounding the reference settling errors, which is related to $\hat{V}_h^{\text{R},k}$ and the reference settling design in Q-EarlySettled-Advantage. UCB-Advantage and Q-EarlySettled-Advantage mainly differ on the reference settling policy, which results in different bounds for reference settling error and the gap-free regret terms in Equations (2) and (3). We show the details in Appendix F.5.2.

**Step 4: Bounding the Expected Regret.** By Equation (9), $Q_h^k(s_h^k, a_h^k) = V_h^k(s_h^k) \geq V_h^\star(s_h^k)$. Thus,
$$\Delta_h(s_h^k, a_h^k) = \text{clip}[V_h^*(s_h^k) - Q_h^*(s_h^k, a_h^k) \mid \Delta_{\min}] \leq \text{clip}[(Q_h^k - Q_h^*)(s_h^k, a_h^k) \mid \Delta_{\min}], \forall(k, h).$$

Equation (4) of Yang et al. (2021) shows that $\mathbb{E}\left(\text{Regret}(K)\right) = \mathbb{E}\left[\sum_{k=1}^{K}\sum_{h=1}^{H}\Delta_h(s_h^k, a_h^k)\right]$, then

$$\mathbb{E}\left(\text{Regret}(K)\right) \leq \mathbb{E}\left[\sum_{k=1}^{K}\sum_{h=1}^{H}\text{clip}[(Q_h^k - Q_h^*)(s_h^k, a_h^k) \mid \Delta_{\min}]\right]. \quad (28)$$

Using the definition of expectation (see Equation (123) in Appendix F.6, which connects Equation (27) to Equation (28)), we can derive the gap-dependent regret bound presented in Theorem 3.2.

## 5  CONCLUSION

In this paper, we have presented the first gap-dependent regret analysis for $Q$-learning using reference-advantage decomposition and also provided the first gap-dependent analysis on the policy switching cost of $Q$-learning, which answers two important open questions. Our novel error decomposition approach and construction of surrogate reference functions can be used in other problems using reference-advantage decomposition such as the offline $Q$-learning and stochastic learning.

ACKNOWLEDGMENT

The work of Z. Zheng, H. Zhang, and L. Xue was supported by the U.S. National Science Foundation under the grants DMS-1811552, DMS-1953189, and CCF-2007823 and by the U.S. National Institutes of Health under the grant 1R01GM152812.

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

In the appendix, Appendix A reviews related works. Appendix B presents the results of our numerical experiments. Appendix C include some useful lemmas. Appendix D, Appendix E and Appendix F provides the proof for Theorem 3.1, Theorem 3.3 and Theorem 3.2, respectively.

# A    RELATED WORK

**On-policy RL for finite-horizon tabular MDPs with worst-case regret.** There are mainly two types of algorithms for reinforcement learning: model-based and model-free algorithms. Model-based algorithms learn a model from past experience and make decisions based on this model, while model-free algorithms only maintain a group of value functions and take the induced optimal actions. Due to these differences, model-free algorithms are usually more space-efficient and time-efficient compared to model-based algorithms. However, model-based algorithms may achieve better learning performance by leveraging the learned model.

Next, we discuss the literature on model-based and model-free algorithms for finite-horizon tabular MDPs with worst-case regret. Auer et al. (2008), Agrawal & Jia (2017), Azar et al. (2017), Kakade et al. (2018), Agarwal et al. (2020), Dann et al. (2019), Zanette & Brunskill (2019),Zhang et al. (2021a),Zhou et al. (2023) and Zhang et al. (2023) worked on model-based algorithms. Notably, Zhang et al. (2023) provided an algorithm that achieves a regret of $\tilde{O}(\min\{\sqrt{SAH^2T}, T\})$, which matches the information lower bound. Jin et al. (2018), Yang et al. (2021), Zhang et al. (2020), Li et al. (2021) and Ménard et al. (2021) work on model-free algorithms. The latter three have introduced algorithms that achieve minimax regret of $\tilde{O}(\sqrt{SAH^2T})$.

**Suboptimality Gap.** When there is a strictly positive suboptimality gap, it is possible to achieve logarithmic regret bounds. In RL, earlier work obtained asymptotic logarithmic regret bounds Auer & Ortner (2007); Tewari & Bartlett (2008). Recently, non-asymptotic logarithmic regret bounds were obtained (Jaksch et al. (2010); Ok et al. (2018); Simchowitz & Jamieson (2019); He et al. (2021). Specifically, Jaksch et al. (2010) developed a model-based algorithm, and their bound depends on the policy gap instead of the action gap studied in this paper. Ok et al. (2018) derived problem-specific logarithmic type lower bounds for both structured and unstructured MDPs. Simchowitz & Jamieson (2019) extended the model-based algorithm proposed by Zanette & Brunskill (2019) and obtained logarithmic regret bounds. Logarithmic regret bounds are also derived in linear function approximation settings (He et al., 2021). Additionally, Nguyen-Tang et al. (2023) provides a gap-dependent regret bounds for offline RL with linear funciton approximation.

Specifically, for model free algorithm, Yang et al. (2021) showed that the optimistic $Q$-learning algorithm by Jin et al. (2018) enjoyed a logarithmic regret $O(\frac{H^6SAT}{\Delta_{\min}})$, which was subsequently refined by Xu et al. (2021). In their work, Xu et al. (2021) introduced the Adaptive Multi-step Bootstrap (AMB) algorithm.

There are also some other works focusing on gap-dependent sample complexity bounds (Jonsson et al., 2020; Marjani & Proutiere, 2020; Al Marjani et al., 2021; Tirinzoni et al., 2022; Wagenmaker et al., 2022b; Wagenmaker & Jamieson, 2022; Wang et al., 2022; Tirinzoni et al., 2023).

**Variance reduction in RL.** The reference-advantage decomposition used in Zhang et al. (2020) and Li et al. (2021) is a technique of variance reduction that was originally proposed for finite-sum stochastic optimization (Gower et al., 2020; Johnson & Zhang, 2013; Nguyen et al., 2017). Later on, model-free RL algorithms also used variance reduction to improve the sample efficiency. For example, it was used in learning with generative models (Sidford et al., 2018; 2023; Wainwright, 2019), policy evaluation (Du et al., 2017; Khamaru et al., 2021; Wai et al., 2019; Xu et al., 2020), offline RL (Shi et al., 2022; Yin et al., 2021), and $Q$-learning (Li et al., 2020; Zhang et al., 2020; Li et al., 2021; Yan et al., 2023; Zheng et al., 2024b).

**RL with low switching cost**. Research in RL with low switching cost aims to minimize the number of policy switches while maintaining comparable regret bounds to fully adaptive counterparts. Bai et al. (2019) was the first to introduce the problem of RL with low-switching cost and proposed a $Q$-learning algorithm with lazy updates that achieves a low switching cost of $\tilde{O}(SAH^3 \log T)$. This work was advanced by Zhang et al. (2020), which improved the regret upper bound and the switching cost. Additionally, Wang et al. (2021) studied RL under the adaptivity constraint. Recently, Qiao et al. (2022) proposed a model-based algorithm with a switching cost of $\tilde{O}(\log \log T)$.

**Other problem-dependent performance.** In practice, RL algorithms often outperform what their worst-case performance guarantees would suggest. This motivates a recent line of works that investigate optimal performance in various problem-dependent settings (Fruit et al., 2018; Jin et al., 2020; Talebi & Maillard, 2018; Wagenmaker et al., 2022a; Zhao et al., 2023; Zhou et al., 2023).

# B    NUMERICAL EXPERIMENTS

In this section, we conduct experiments[1]. All the experiments are conducted in a synthetic environment to demonstrate the better gap-dependent regret of UCB-Advantage and Q-EarlySettled-Advantage compared to other two model-free algorithms: UCB-Hoeffding (Jin et al., 2018) and AMB (Xu et al., 2021). We will consider two different scales of experiments across two cases: a general MDP and a deterministic MDP.

We first set $H = 5$, $S = 3$, and $A = 2$. The reward $r_h(s, a)$ for each $(s, a, h)$ is generated independently and uniformly at random from $[0, 1]$. For general MDP, $\mathbb{P}_h(\cdot \mid s, a)$ is generated on the $S$-dimensional simplex independently and uniformly at random for $(s, a, h)$. For deterministic MDP, $\mathbb{P}_h(\cdot \mid s, a)$ is a randomly generated vector with only one element equal to 1, and all others equal to 0 for each $(s, a, h)$. Under the given MDP, we generate $3 \times 10^5$ episodes. For each episode, we randomly choose the initial state uniformly from the $S$ states. For all four algorithms, we set $\iota = 1$ and the hyper-parameter $c_1 = \sqrt{2}$ in the Hoeffding-type bonus. Here, $c_1$ represents the undefined constant in the bonus terms of the UCB-Hoeffding and AMB algorithms, as well as the multipliers in the bonus expressions in line 10 of Algorithm 1 (UCB-Advantage) and lines 2 and 4 of Algorithm 2 (Q-EarlySettled-Advantage). In both the UCB-Advantage and Q-EarlySettled-Advantage algorithms, we set the hyper-parameter $c_2 = 2$, where $c_2$ denotes the constant in the variance estimators of the advantage-type bonus, corresponding to the undefined constant in the second term of line 9 in Algorithm 1 and line 16 in Algorithm 2. Additionally, we set $c_3 = 1$, which is the multiplier in the last term of line 9 in Algorithm 1 and the last term of line 8 in Algorithm 2. For UCB-Advantage, we set $N_0 = 200$, and for Q-EarlySettled-Advantage, we set $\beta = 0.05$.

To show error bars, we collect 10 sample paths for all algorithms under the same MDP environment and show the relationship between $\text{Regret}(T)/\log(K + 1)$ and the total number of episodes $K$ in Figure 1. For both panels, the solid line represents the median of the 10 sample paths, while the shaded area shows the 10th and 90th percentiles.

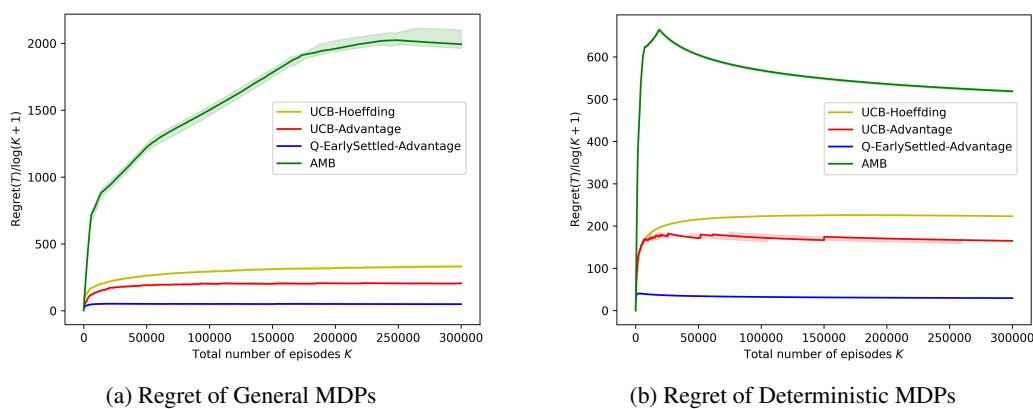

(a) Regret of General MDPs        (b) Regret of Deterministic MDPs

Figure 1: Numerical comparison of regrets with $H = 5$, $S = 3$, and $A = 2$

We also conduct a larger scale experiment with $H = 10$, $S = 5$, and $A = 5$ for $3 \times 10^6$ episodes in both types of MDPs. With all other settings unchanged, the result is shown in the following Figure 2:

---

[1]All the experiments are run on a server with Intel Xeon E5-2650v4 (2.2GHz) and 100 cores. Each replication is limited to a single core and 4GB RAM. The total execution time is less than 2 hours. The code for the numerical experiments is included in the supplementary materials along with the submission.

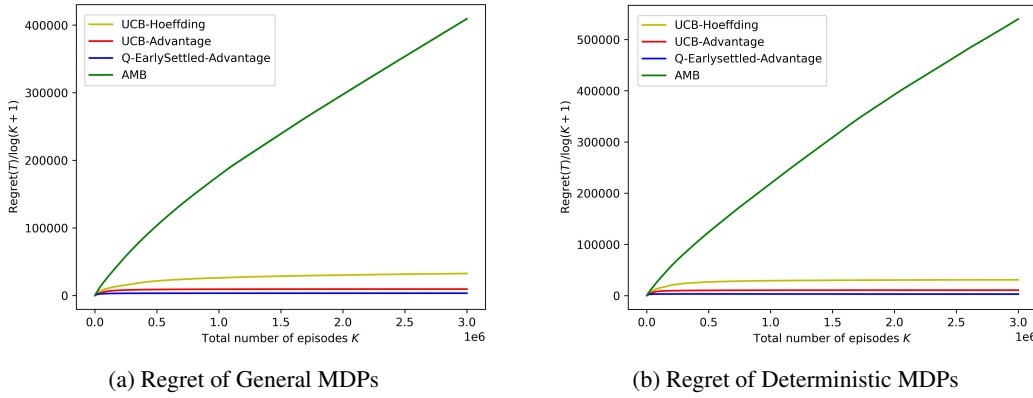

(a) Regret of General MDPs

(b) Regret of Deterministic MDPs

Figure 2: Numerical comparison of regrets with $H = 10$, $S = 5$, and $A = 5$

From the two figures, we observe that both UCB-Advantage and Q-EarlySettled-Advantage enjoy lower regret compared to UCB-Hoeffding and AMB. The y-axis represents $\text{Regret}(T)/\log(K+1)$, and we note that the curves for UCB-Advantage and Q-EarlySettled-Advantage approach horizontal lines as $K$ becomes sufficiently large. This suggests that the regret for these two algorithms grows logarithmically with $K$. In particular, Q-EarlySettled-Advantage achieves even lower regret than UCB-Advantage when $K$ is large. These features are consistent with our theoretical results.

We also conduct an experiment to evaluate the policy switching cost of the UCB-Advantage algorithm for $(H, S, A) = (5, 3, 2)$ and $(10, 5, 5)$ with the same experimental settings. The results are presented in the following figures:

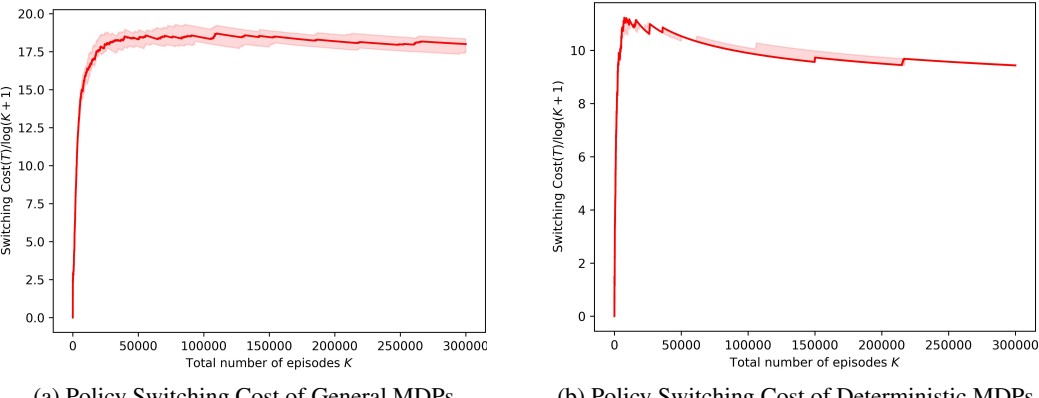

(a) Policy Switching Cost of General MDPs

(b) Policy Switching Cost of Deterministic MDPs

Figure 3: Policy switching cost of UCB-Advantage algorithm with $H = 5$, $S = 3$, and $A = 2$

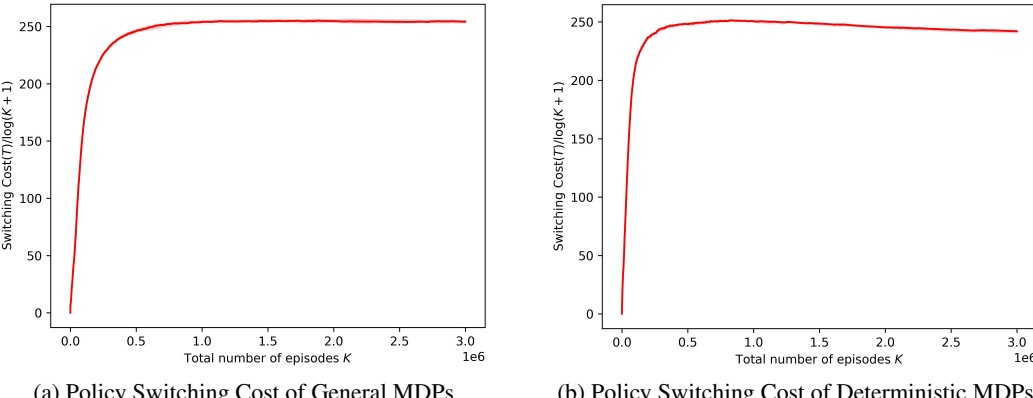

(a) Policy Switching Cost of General MDPs

(b) Policy Switching Cost of Deterministic MDPs

Figure 4: Policy switching cost of UCB-Advantage algorithm with $H = 10$, $S = 5$, and $A = 5$

In these two figures, the y-axis represents the ratio of policy switching cost to $\log(K+1)$. We note that all these four curves approach horizontal lines as $K$ becomes sufficiently large, which is consistent with our logarithmic policy switching cost shown in Equation (4).

## C    GENERAL LEMMAS

**Lemma C.1.** (Azuma-Hoeffding Inequality). *Suppose $\{X_k\}_{k=0}^{\infty}$ is a martingale and $|X_k - X_{k-1}| \leq c_k$, $\forall k \in \mathbb{N}_+$, almost surely. Then for any positive integers $N$ and any positive real number $\epsilon$, it holds that:*

$$\mathbb{P}\left(X_N - X_0 \geq \epsilon\right) \leq \exp\left(-\frac{\epsilon^2}{2\sum_{k=1}^{N} c_k^2}\right),$$

*and*

$$\mathbb{P}\left(|X_N - X_0| \geq \epsilon\right) \leq 2\exp\left(-\frac{\epsilon^2}{2\sum_{k=1}^{N} c_k^2}\right).$$

**Lemma C.2.** (Lemma 10 in Zhang et al. (2022a)). *Let $X_1, X_2, \ldots$ be a sequence of random variables taking value in $[0, l]$. Define $\mathcal{F}_k = \sigma(X_1, X_2, \ldots, X_k)$ and $Y_k = \mathbb{E}[X_k|\mathcal{F}_{k-1}]$ for $k \geq 1$. For any $\delta > 0$, we have that*

$$\mathbb{P}\left[\exists n, \sum_{k=1}^{n} X_k \geq 3\sum_{k=1}^{n} Y_k + l\log(1/\delta)\right] \leq \delta$$

*and*

$$\mathbb{P}\left[\exists n, \sum_{k=1}^{n} Y_k \geq 3\sum_{k=1}^{n} X_k + l\log(1/\delta)\right] \leq \delta.$$

**Lemma C.3.** (Lemma 11 in Zhang et al. (2021b)). *Let $\{M_n\}_{n \geq 0}$ be a martingale such that $M_0 = 0$ and $|M_n - M_{n-1}| \leq c$ for some $c > 0$ and any $n \geq 1$. Let*

$$\text{Var}_n = \sum_{k=1}^{n} \mathbb{E}\left[(M_k - M_{k-1})^2|\mathcal{F}_{k-1}\right]$$

*for $n \geq 0$, where $\mathcal{F}_k = \sigma(M_1, \ldots, M_k)$. Then for any positive integer $n$ and any $\epsilon, \delta > 0$, we have*

$$\mathbb{P}\left(|M_n| \geq 2\sqrt{2\text{Var}_n \ln\left(\frac{1}{\delta}\right)} + 2\sqrt{\epsilon \ln\left(\frac{1}{\delta}\right)} + 2c\ln\left(\frac{1}{\delta}\right)\right) \leq 2\left(\log_2\left(\frac{nc^2}{\epsilon}\right) + 1\right)\delta.$$

## D    PROOF OF THEOREM 3.1

### D.1    ALGORITHM DETAILS

The UCB-Advantage algorithm, first introduced in Zhang et al. (2020), achieves the information-theoretic bound on regret up to logarithmic factors, using a model-free algorithm. The key innovation of the algorithm lies in its combination of UCB exploration (Jin et al., 2018) with a newly introduced reference-advantage decomposition for updating $Q$-estimates.

Before discussing the algorithm in detail, we will first review the special stage design used in the algorithm. For any triple $(s, a, h)$, we divide the samples received for the triple into consecutive stages. Define $e_1 = H$ and $e_{i+1} = \lfloor(1 + \frac{1}{H})e_i\rfloor$ for all $i \geq 1$, standing for the length of the stages. We also let $\mathcal{L} := \{\sum_{i=1}^{j} e_i | j = 1, 2, 3, \ldots\}$ be the set of indices marking the ends of the stages.

We note that the definition of stages is with respect to the triple $(s, a, h)$. For any fixed pair of $k$ and $h$, let $(s_h^k, a_h^k)$ be the state-action pair at the $h$-th step during the $k$-th episode of the algorithm. We say that $(k, h)$ falls in the $j$-th stage of $(s, a, h)$ if and only if $(s, a) = (s_h^k, a_h^k)$ and the total visit number of $(s_h^k, a_h^k)$ after the $k$-th episode is in $(\sum_{i=1}^{j-1} e_i, \sum_{i=1}^{j} e_i]$.

Now we introduce the stage-based update framework. For any $(s, a, h)$ triple, we update $Q_h(s, a)$ when the total visit number of $(s, a, h)$ reaches the end of the current stage. For $k$-th episode at the end of a given stage, the $Q$-estimate $Q_h^{1,k+1}(s_h^k, a_h^k)$ learned from UCB is updated to:

$$Q_h^{1,k+1}(s_h^k, a_h^k) = r_h^k(s_h^k, a_h^k) + \frac{1}{\check{n}_h^{k+1}} \sum_{i=1}^{\check{n}_h^{k+1}} V_{h+1}^{k^{\check{l}_i}}(s_{h+1}^{\check{l}_i}) + c_1 \sqrt{\frac{H^2 \iota}{\check{n}_h^{k+1}}}. \tag{29}$$

Here we define $\check{n}_h^k = \check{n}_h^k(s_h^k, a_h^k)$ as the number of visits to $(s_h^k, a_h^k, h)$ during the stage immediately before the current stage of $(k, h)$ and $\check{l}_i = \check{l}_{h,k}^i$ denotes the index of the $i$-th episode among the $\check{n}_h^k$ episodes. $V_h^k(s)$ is the $V$-estimate at the end of the episode $k-1$ with the initial value $V_h^1(s) = H$. The term $c_1 \sqrt{\frac{H^2 \iota}{\check{n}_h^{k+1}}}$ represents the exploration bonus for $\check{n}_h^{k+1}$-th visit, where $c_1$ is a sufficiently large constant and $\iota = \log(\frac{2}{p})$ with $p \in (0, 1)$ being the failure probability. This type of bonus is commonly used in Hoeffding-type updates (Jin et al. (2018); Li et al. (2021); Zheng et al. (2024a)).

The other estimate, denoted by $Q_h^{2,k+1}(s_h^k, a_h^k)$, uses the reference-advantage decomposition technique. For $k$-th episode at the end of a given stage, it is updated to:

$$r_h^k(s_h^k, a_h^k) + \frac{1}{n_h^{k+1}} \sum_{i=1}^{n_h^{k+1}} V_{h+1}^{\text{ref},k^{l_i}}(s_{h+1}^{l_i}) + \frac{1}{\check{n}_h^{k+1}} \sum_{i=1}^{\check{n}_h^{k+1}} \left( V_{h+1}^{k^{\check{l}_i}} - V_{h+1}^{\text{ref},k^{\check{l}_i}} \right)(s_{h+1}^{\check{l}_i}) + b_h^{k+1}(s_h^k, a_h^k). \tag{30}$$

Here we define $n_h^k = n_h^k(s_h^k, a_h^k)$ be the number of visits to $(s_h^k, a_h^k, h)$ prior to the stage of $(k, h)$ and $l_i = l_{h,k}^i$ denotes the index of $i$-th episode among the $n_h^k$ episodes.

In Equation (30), $V_h^{\text{ref},k}(s)$ is the reference function learned at the end of episode $k-1$. We expect that for any $s \in \mathcal{S}$, sufficiently large $k$ and some given $\beta \in (0, H]$, it holds $|V_h^{\text{ref},k}(s) - V_h^\star(s)| \leq \beta$.

With these $Q$-estimates, we can update the final $Q$-estimate as follows:

$$Q_h^{k+1}(s_h^k, a_h^k) = \min\{Q_h^{1,k+1}(s_h^k, a_h^k), Q_h^{2,k+1}(s_h^k, a_h^k), Q_h^k(s_h^k, a_h^k)\}. \tag{31}$$

We also incorporate $Q_h^k(s_h^k, a_h^k)$ here to keep the mononicity of the update. After updating the $Q$-estimate, we can learn $V_h^{k+1}(s_h^k)$ by a greedy policy with respect to the $Q$-estimates, i.e., $V_h^{k+1}(s_h^k) = \max_a Q_h^{k+1}(s_h^k, a)$. If the number of visits to the state-step pair $(s, h)$ first exceeds $N_0 = O(\frac{SAH^5 \iota}{\beta^2})$ at $k$-th episode, then we update the final reference function $V_h^{\text{REF}}(s)$ to $V_h^{k+1}(s)$. For the reader's convenience, we have also provided the detailed algorithm below.

---

**Algorithm 1** UCB-Advantage

---

1: **Initialize:** set all accumulators to 0; for all $(s, a, h) \in \mathcal{S} \times \mathcal{A} \times [H]$, set $Q_h(s, a)$, $V_h(s) \leftarrow H - h + 1$; $V_h^{\text{ref}}(s) \leftarrow H$;
2: **for** episodes $k \leftarrow 1, 2, \ldots, K$ **do**
3:     observe $s_1$;
4:     **for** $h \leftarrow 1, 2, \ldots, H$ **do**
5:         Take action $a_h \leftarrow \arg\max_a Q_h(s_h, a)$, and observe $s_{h+1}$.
6:         Update the accumulators by $n := n_h(s_h, a_h) \overset{+}{\leftarrow} 1$, $\check{n} := \check{n}_h(s_h, a_h) \overset{+}{\leftarrow} 1$,
7:         and Equation (32), Equation (33), Equation (34).
8:         **if** $n \in \mathcal{L}$ **then**
9:             $b \leftarrow c_2 \sqrt{\frac{\sigma_h^{\text{ref}}/n - (\mu_h^{\text{ref}}/n)^2}{n}} \iota + c_2 \sqrt{\frac{\check{\sigma}/\check{n} - (\check{\mu}/\check{n})^2}{\check{n}}} \iota + c_3 \left( \frac{H\iota}{n} + \frac{H\iota}{\check{n}} + \frac{H\iota^{3/4}}{n^{3/4}} + \frac{H\iota^{3/4}}{\check{n}^{3/4}} \right)$;
10:             $\bar{b} \leftarrow c_1 \sqrt{\frac{H^2}{\check{n}} \iota}$;
11:             $Q_h(s_h, a_h) \leftarrow \min\{r_h(s_h, a_h) + \frac{\check{v}}{\check{n}} + \bar{b}, r_h(s_h, a_h) + \frac{\mu^{\text{ref}}}{n} + \frac{\check{\mu}}{\check{n}} + b, Q_h(s_h, a_h)\}$;
12:             $V_h(s_h) \leftarrow \max_a Q_h(s_h, a)$;
13:             $\check{n}_h(s_h, a_h), \check{\mu}_h(s_h, a_h), \check{v}_h(s_h, a_h), \check{\sigma}_h(s_h, a_h) \leftarrow 0$;
14:         **end if**
15:         **if** $\sum_a n_h(s_h, a) = N_0$ **then** $V_h^{\text{ref}}(s_h) \leftarrow V_h(s_h)$
16:         **end if**
17:     **end for**
18: **end for**

---

In the algorithm, $c_1, c_2, c_3 > 0$ are sufficiently large constant. The stage-wise accumulators in the algorithm are updated as follows.

$$\breve{\mu} := \breve{\mu}_h(s_h, a_h) \xleftarrow{+} V_{h+1}(s_{h+1}) - V_{h+1}^{\mathrm{ref}}(s_{h+1}); \quad \breve{v} := \breve{v}_h(s_h, a_h) \xleftarrow{+} V_{h+1}(s_{h+1}); \quad (32)$$

$$\breve{\sigma} := \breve{\sigma}_h(s_h, a_h) \xleftarrow{+} \left(V_{h+1}(s_{h+1}) - V_{h+1}^{\mathrm{ref}}(s_{h+1})\right)^2; \quad (33)$$

Meanwhile, the following two global accumulators are used for the samples in all stages

$$\mu^{\mathrm{ref}} := \mu_h^{\mathrm{ref}}(s_h, a_h) \xleftarrow{+} V_{h+1}^{\mathrm{ref}}(s_{h+1}); \quad \sigma^{\mathrm{ref}} := \sigma_h^{\mathrm{ref}}(s_h, a_h) \xleftarrow{+} \left(V_{h+1}^{\mathrm{ref}}(s_{h+1})\right)^2. \quad (34)$$

We use $\mu_h^{\mathrm{ref},k}, \sigma_h^{\mathrm{ref},k}, \breve{\mu}_h^k, \breve{v}_h^k, \breve{\sigma}_h^k, b_h^k$ to denote respectively the values of $\mu^{\mathrm{ref}}, \sigma^{\mathrm{ref}}, \breve{\mu}, \breve{v}, \breve{\sigma}, b$ at step $h$ by the start of the $k$-th episode.

UCB-Advantage assumes that the reward is known after the visit to the triple $(s, a, h)$, which is common in RL settings. When the reward is unknown after the visit to the triple, inverse reinforcement learning (Zeng et al., 2022; Liu & Zhu, 2022; 2023; 2024; Qiao et al., 2024; Liu & Zhu, 2025) provides a bi-level learning structure to help learn the reward.

## D.2 KEY LEMMAS

Before proceeding to the proof, we will first establish several key lemmas. In the algorithm, define $\iota = \log(2/p)$ with $p \in (0, 1)$ being the failure probability.

**Lemma D.1.** *Using $\forall(s, a, h, k)$ as the simplified notation for $\forall(s, a, h, k) \in \mathcal{S} \times \mathcal{A} \times [H] \times [K]$. For $\forall(s, a, h, k)$, let $N_h^k(s) = \sum_a n_h^k(s, a)$, $\lambda_h^k(s) = \mathbb{I}[N_h^k(s) < N_0]$ and define the surrogate function as $\hat{V}_h^{\mathrm{ref},k}(s) = \max\{V_h^\star(s), \min\{V_h^\star(s) + \beta, V_h^{\mathrm{ref},k}(s)\}\}$. Then we have the following conclusions:*

(a) *(Proposition 4 in Zhang et al. (2020)). With probability at least $1 - (4H^2T^4 + 12T)p$, the following event holds:*

$$\mathcal{E}_1 = \left\{Q_h^\star(s, a) \leq Q_h^{k+1}(s, a) \leq Q_h^k(s, a), \forall(s, a, h, k)\right\}.$$

(b) *(Corollary 6 in Zhang et al. (2020)). With probability at least $1 - (4H^2T^4 + 13T)p$, the following event holds:*

$$\mathcal{E}_2 = \left\{N_h^k(s) \geq N_0 \Rightarrow V_h^\star(s) \leq V_h^{\mathrm{ref},k} \leq V_h^\star(s) + \beta, \forall(s, h, k) \in \mathcal{S} \times [H] \times [K]\right\}.$$

(c) *With probability at least $1 - Hp$, the following event holds:*

$$\mathcal{E}_3 = \left\{\sum_{k=1}^K \mathbb{P}_{s_h^k, a_h^k, h} \lambda_{h+1}^k \leq 3 \sum_{k=1}^K \lambda_{h+1}^k(s_{h+1}^k) + \iota, \ \forall h \in [H]\right\}.$$

*Especially, $\lambda_{H+1}^k(s) = 0$.*

(d) *With probability at least $1 - SATp$, the following event holds:*

$$\mathcal{E}_4 = \left\{\frac{\left|\sum_{i=1}^{n_h^k} \left(\mathbb{1}_{s_{h+1}^{l_i}} - \mathbb{P}_{s,a,h}\right)\left(\hat{V}_{h+1}^{\mathrm{ref},l_i} - V_{h+1}^\star\right)\right|}{n_h^k(s, a)} \leq \beta\sqrt{\frac{2\iota}{n_h^k(s, a)}}, \forall(s, a, h, k)\right\}.$$

(e) *With probability at least $1 - SAT^2p$, the following event holds:*

$$\mathcal{E}_5 = \left\{\frac{\left|\sum_{i=1}^{n_h^k} \left(\mathbb{1}_{s_{h+1}^{l_i}} - \mathbb{P}_{s,a,h}\right)V_{h+1}^\star\right|}{n_h^k(s, a)} \leq 2\sqrt{\frac{2\mathbb{Q}^\star \iota}{n_h^k(s, a)}} + \frac{4H\iota}{n_h^k(s, a)}, \forall(s, a, h, k)\right\}.$$

(f) *With probability at least $1 - SAT^2p$, the following event holds:*

$$\mathcal{E}_6 = \left\{\frac{\left|\sum_{i=1}^{\breve{n}_h^k} \left(\mathbb{1}_{s_{h+1}^{\breve{l}_i}} - \mathbb{P}_{s,a,h}\right)\left(\hat{V}_{h+1}^{\mathrm{ref},\breve{l}_i} - V_{h+1}^\star\right)\right|}{\breve{n}_h^k(s, a)} \leq \beta\sqrt{\frac{2\iota}{\breve{n}_h^k(s, a)}}, \forall(s, a, h, k)\right\}.$$

*(g) With probability at least $1 - SATp$, the following event holds:*

$$\mathcal{E}_7 = \left\{ \frac{\left| \sum_{i=1}^{n_h^k} \left( \mathbb{1}_{s_{h+1}^{l_i}} - \mathbb{P}_{s,a,h} \right) (V_{h+1}^{\star})^2 \right|}{n_h^k(s,a)} \leq H^2 \sqrt{\frac{2\iota}{n_h^k(s,a)}}, \forall (s,a,h,k) \right\}.$$

*(h) With probability at least $1 - SATp$, the following event holds:*

$$\mathcal{E}_8 = \left\{ \frac{\left| \sum_{i=1}^{n_h^k} \left( \mathbb{1}_{s_{h+1}^{l_i}} - \mathbb{P}_{s,a,h} \right) V_{h+1}^{\star} \right|}{n_h^k(s,a)} \leq H \sqrt{\frac{2\iota}{n_h^k(s,a)}}, \forall (s,a,h,k) \right\}.$$

*Proof.* We only need prove (c) to (e).

(c) By using Lemma C.2 with $l = 1$ and $\delta = p$ and considering all possible values of $h \in [H]$, we can prove this conclusion.

(d) From the definition of $\hat{V}_h^{\text{ref},k}(s)$, we know that for any $k \in [K]$:

$$V_h^{\star}(s) \leq \hat{V}_h^{\text{ref},k}(s) \leq V_h^{\star}(s) + \beta. \tag{35}$$

Then the sequence

$$\left\{ \sum_{i=1}^{j} \left( \mathbb{1}_{s_{h+1}^{l_i}} - \mathbb{P}_{s,a,h} \right) \left( \hat{V}_{h+1}^{\text{ref},\check{l}_i} - V_{h+1}^{\star} \right) \right\}_{j \in \mathbb{N}^+}$$

is a martingale sequence with

$$\left| \left( \mathbb{1}_{s_{h+1}^{l_i}} - \mathbb{P}_{s,a,h} \right) \left( \hat{V}_{h+1}^{\text{ref},\check{l}_i} - V_{h+1}^{\star} \right) \right| \leq \beta.$$

Then according to Azuma-Hoeffding inequality, for any $p \in (0,1)$, with probability at least $1 - p$, it holds for given $n_h^k(s,a) = n \in \mathbb{N}_+$ that:

$$\frac{1}{n} \left| \sum_{i=1}^{n} \left( \mathbb{1}_{s_{h+1}^{l_i}} - \mathbb{P}_{s,a,h} \right) \left( \hat{V}_{h+1}^{\text{ref},l_i} - V_{h+1}^{\star} \right) \right| \leq \sqrt{\frac{2\beta^2\iota}{n}}.$$

For any all $(s,a,h,k) \in \mathcal{S} \times \mathcal{A} \times [H] \times [K]$, we have $n_h^k(s,a) \in [\frac{T}{H}]$. Considering all the possible combinations $(s,a,h,n) \in \mathcal{S} \times \mathcal{A} \times [H] \times [\frac{T}{H}]$, with probability at least $1 - SATp$, it holds simultaneously for all $(s,a,h,k) \in \mathcal{S} \times \mathcal{A} \times [H] \times [K]$ that:

$$\frac{1}{n_h^k(s,a)} \left| \sum_{i=1}^{n_h^k} \left( \mathbb{1}_{s_{h+1}^{l_i}} - \mathbb{P}_{s,a,h} \right) \left( \hat{V}_{h+1}^{\text{ref},l_i} - V_{h+1}^{\star} \right) \right| \leq \sqrt{\frac{2\beta^2\iota}{n_h^k(s,a)}}.$$

(e) The sequence

$$\left\{ \sum_{i=1}^{j} \left( \mathbb{1}_{s_{h+1}^{l_i}} - \mathbb{P}_{s,a,h} \right) V_{h+1}^{\star} \right\}_{j \in \mathbb{N}^+}$$

is a martingale sequence with

$$\left| \left( \mathbb{1}_{s_{h+1}^{l_i}} - \mathbb{P}_{s,a,h} \right) V_{h+1}^{\star} \right| \leq H.$$

Using Lemma C.3 with $c = H$, $\epsilon = H^2$ and $\delta = \frac{p}{2}$, for a given $n_h^k(s,a) = n \in [\frac{T}{H}]$, with probability at least $1 - (\log_2(n) + 1)p \geq 1 - Tp$, we have:

$$\frac{1}{n} \left| \sum_{i=1}^{n} \left( \mathbb{1}_{s_{h+1}^{l_i}} - \mathbb{P}_{s,a,h} \right) V_{h+1}^{\star} \right| \leq 2\sqrt{\frac{2\mathbb{Q}^{\star}\iota}{n}} + \frac{4H\iota}{n}$$

Considering all the possible combinations $(s, a, h, n) \in \mathcal{S} \times \mathcal{A} \times [H] \times [\frac{T}{H}]$, with probability at least $1 - SAT^2p$, it holds simultaneously for all $(s, a, h, k) \in \mathcal{S} \times \mathcal{A} \times [H] \times [K]$ that:

$$\frac{1}{n_h^k(s,a)} \left| \sum_{i=1}^{n_h^k} \left( \mathbb{1}_{s_{h+1}^{l_i}} - \mathbb{P}_{s,a,h} \right) V_{h+1}^\star \right| \le 2\sqrt{\frac{2\mathbb{Q}^\star}{n_h^k(s,a)}} + \frac{4H\iota}{n_h^k(s,a)}.$$

(f) The sequence

$$\left\{ \sum_{i=1}^{j} \left( \mathbb{1}_{s_{h+1}^{\breve{l}_i}} - \mathbb{P}_{s,a,h} \right) \left( \hat{V}_{h+1}^{\mathrm{ref},\breve{l}_i} - V_{h+1}^\star \right) \right\}_{j \in \mathbb{N}^+}$$

is a martingale sequence with

$$\left| \left( \mathbb{1}_{s_{h+1}^{\breve{l}_i}} - \mathbb{P}_{s,a,h} \right) \left( \hat{V}_{h+1}^{\mathrm{ref},\breve{l}_i} - V_{h+1}^\star \right) \right| \le \beta.$$

Then according to Azuma-Hoeffding inequality, for any $p \in (0,1)$, with probability at least $1 - p$, it holds for given $\breve{n}_h^k(s,a) = \breve{n} \in \mathbb{N}_+$ that:

$$\frac{1}{\breve{n}} \left| \sum_{i=1}^{\breve{n}} \left( \mathbb{1}_{s_{h+1}^{\breve{l}_i}} - \mathbb{P}_{s,a,h} \right) \left( \hat{V}_{h+1}^{\mathrm{ref},\breve{l}_i} - V_{h+1}^\star \right) \right| \le \sqrt{\frac{2\beta^2\iota}{\breve{n}}}.$$

For any all $(s, a, h, k) \in \mathcal{S} \times \mathcal{A} \times [H] \times [K]$, we have $\breve{n}_h^k(s,a) \in [\frac{T}{H}]$. Considering all the possible combinations $(s, a, h, k) \in \mathcal{S} \times \mathcal{A} \times [H] \times [K]$ and $\breve{n}_h^k(s,a) \in [\frac{T}{H}]$, with probability at least $1 - SAT^2/Hp \ge 1 - SAT^2p$, it holds simultaneously for all $(s, a, h, k) \in \mathcal{S} \times \mathcal{A} \times [H] \times [K]$ that:

$$\frac{1}{\breve{n}_h^k(s,a)} \left| \sum_{i=1}^{\breve{n}_h^k} \left( \mathbb{1}_{s_{h+1}^{\breve{l}_i}} - \mathbb{P}_{s,a,h} \right) \left( \hat{V}_{h+1}^{\mathrm{ref},\breve{l}_i} - V_{h+1}^\star \right) \right| \le \sqrt{\frac{2\beta^2\iota}{\breve{n}_h^k(s,a)}}.$$

(g) The sequence

$$\left\{ \sum_{i=1}^{j} \left( \mathbb{1}_{s_{h+1}^{l_i}} - \mathbb{P}_{s,a,h} \right) \left( V_{h+1}^\star \right)^2 \right\}_{j \in \mathbb{N}^+}$$

is a martingale sequence with

$$\left| \left( \mathbb{1}_{s_{h+1}^{l_i}} - \mathbb{P}_{s,a,h} \right) \left( V_{h+1}^\star \right)^2 \right| \le H^2.$$

Then according to Azuma-Hoeffding inequality, with probability at least $1 - p$, it holds for given $n_h^k(s,a) = n$ that:

$$\frac{1}{n} \left| \sum_{i=1}^{n} \left( \mathbb{1}_{s_{h+1}^{l_i}} - \mathbb{P}_{s,a,h} \right) (V_{h+1}^\star)^2 \right| \le H^2 \sqrt{\frac{2\iota}{n}}$$

Considering all the possible combinations $(s, a, h, n) \in \mathcal{S} \times \mathcal{A} \times [H] \times [\frac{T}{H}]$, with probability at least $1 - SATp$, it holds simultaneously for all $(s, a, h, k) \in \mathcal{S} \times \mathcal{A} \times [H] \times [K]$ that:

$$\frac{1}{n_h^k(s,a)} \left| \sum_{i=1}^{n_h^k} \left( \mathbb{1}_{s_{h+1}^{l_i}} - \mathbb{P}_{s,a,h} \right) (V_{h+1}^\star)^2 \right| \le H^2 \sqrt{\frac{2\iota}{n_h^k(s,a)}}.$$

(h) The sequence

$$\left\{ \sum_{i=1}^{j} \left( \mathbb{1}_{s_{h+1}^{l_i}} - \mathbb{P}_{s,a,h} \right) V_{h+1}^\star \right\}_{j \in \mathbb{N}^+}$$

is a martingale sequence with

$$\left| \left( \mathbb{1}_{s_{h+1}^{l_i}} - \mathbb{P}_{s,a,h} \right) V_{h+1}^\star \right| \le H.$$

Then according to Azuma-Hoeffding inequality, with probability at least $1 - p$, it holds for given $n_h^k(s, a) = n$ that:

$$\frac{1}{n} \left| \sum_{i=1}^{n} \left( \mathbb{1}_{s_{h+1}^{l_i}} - \mathbb{P}_{s,a,h} \right) V_{h+1}^{\star} \right| \leq H \sqrt{\frac{2\iota}{n}}$$

Considering all the possible combinations $(s, a, h, n) \in \mathcal{S} \times \mathcal{A} \times [H] \times [\frac{T}{H}]$, with probability at least $1 - SATp$, it holds simultaneously for all $(s, a, h, k) \in \mathcal{S} \times \mathcal{A} \times [H] \times [K]$ that:

$$\frac{1}{n_h^k(s, a)} \left| \sum_{i=1}^{n_h^k} \left( \mathbb{1}_{s_{h+1}^{l_i}} - \mathbb{P}_{s,a,h} \right) V_{h+1}^{\star} \right| \leq H \sqrt{\frac{2\iota}{n_h^k(s, a)}}.$$

$\square$

From this lemma, we know that the event $\bigcap_{i=1}^{8} \mathcal{E}_i$ holds with probability at least $1 - 40H^2 SAT^4 p$.

Next, we will discuss the relationship among the $V$-estimate $V_h^k(s)$, the reference function $V_h^{\text{ref},k}(s)$, the surrogate function $\hat{V}_h^{\text{ref},k}(s)$ and the final value $V_h^{\text{REF}}(s)$ of the reference function.

**Lemma D.2.** *Under the event $\mathcal{E}_1 \cap \mathcal{E}_2$ in Lemma D.1, we have the following conclusions:*

(a) $\hat{V}_h^{\text{ref},k}(s) = \min\{V_h^{\star}(s) + \beta, V_h^{\text{ref},k}(s)\}$

(b) $0 \leq V_h^{\text{ref},k}(s) - V_h^{\text{REF}}(s) \leq H\lambda_h^k(s).$

(c) $0 \leq V_h^{\text{ref},k}(s) - \hat{V}_h^{\text{ref},k}(s) \leq H\lambda_h^k(s).$

(d) $\left| \hat{V}_h^{\text{ref},k}(s) - V_h^{\text{REF}}(s) \right| \leq H\lambda_h^k(s).$

*Proof.*     (a)  Under the event $\mathcal{E}_1$ in Lemma D.1, we have $V_h^{\text{ref},k}(s) \geq V_h^k(s) \geq V_h^{\star}(s)$. Therefore, $\min\{V_h^{\star}(s) + \beta, V_h^{\text{ref},k}(s)\} \geq V_h^{\star}(s)$. According to the definition of $\hat{V}_h^{\text{ref},k}(s)$, we have $\hat{V}_h^{\text{ref},k}(s) = \min\{V_h^{\star}(s) + \beta, V_h^{\text{ref},k}(s)\}$.

(b)  For any $(s, h, k) \in \mathcal{S} \times [H] \times [K]$:
If $N_h^k(s) \geq N_0$, then $\lambda_h^k(s) = 0$. In this case, the reference function $V_h^{\text{ref},k}(s)$ is updated to its final value $V_h^{\text{REF}}(s)$ and then $V_h^{\text{ref},k}(s) - V_h^{\text{REF}}(s) = 0 = H\lambda_h^k(s)$.
If $N_h^k(s) < N_0$, then $\lambda_h^k(s) = 1$. Since the reference function is non-increasing and $V_h^{\text{ref},1}(s) = H$, we have $0 \leq V_h^{\text{ref},k}(s) - V_h^{\text{REF}}(s) \leq H = H\lambda_h^k(s)$.
Combining these two cases, we can prove the conclusion (b).

(c)  For any $(s, h, k) \in \mathcal{S} \times [H] \times [K]$:
If $N_h^k(s) \geq N_0$, then $\lambda_h^k(s) = 0$. Under the event $\mathcal{E}_2$ in Lemma D.1, we have $V_h^{\text{ref},k}(s) \leq V_h^{\star}(s) + \beta$. Therefore, it holds that $\hat{V}_h^{\text{ref},k}(s) = V_h^{\text{ref},k}(s)$ by (a). In this case, $V_h^{\text{ref},k}(s) - \hat{V}_h^{\text{ref},k}(s) = 0 = H\lambda_h^k(s)$.
If $N_h^k(s) < N_0$, then $\lambda_h^k(s) = 1$. Since the reference function is non-increasing and $V_h^{\text{ref},1}(s) = H$, we have $0 \leq V_h^{\text{ref},k}(s) - \hat{V}_h^{\text{ref},k}(s) \leq H$.
Combining these two cases, we can prove the conclusion (c).

(d)  For any $(s, h, k) \in \mathcal{S} \times [H] \times [K]$:
If $N_h^k(s) \geq N_0$, then $\lambda_h^k(s) = 0$. In this case, the reference function $V_h^{\text{ref},k}(s)$ is updated to its final value $V_h^{\text{REF}}(s)$. Under the event $\mathcal{E}_2$ in Lemma D.1, we have $V_h^{\text{REF}}(s) = V_h^{\text{ref},k}(s) \leq V_h^{\star}(s) + \beta$. In this case, we know $\hat{V}_h^{\text{ref},k}(s) = V_h^{\text{ref},k}(s) = V_h^{\text{REF}}(s)$. Therefore, it holds that $\hat{V}_h^{\text{ref},k}(s) - V_h^{\text{REF}}(s) = 0 = H\lambda_h^k(s)$.
If $N_h^k(s) < N_0$, then $\lambda_h^k(s) = 1$. Since the reference function is non-increasing and $V_h^{\text{ref},1}(s) = H$, we have $0 \leq V_h^{\text{REF}}(s) \leq V_h^{\text{ref},k}(s) \leq H$ and $0 \leq \hat{V}_h^{\text{ref},k}(s) \leq V_h^{\text{ref},k}(s) \leq$

$H$. Therefore, it holds that $\left|\hat{V}_h^{\text{ref},k}(s) - V_h^{\text{REF}}(s)\right| \leq H = H\lambda_h^k(s)$.

Combining these two cases, we can prove the conclusion (d).

$\square$

**Lemma D.3.** *For any* $(s, a, h, k) \in \mathcal{S} \times \mathcal{A} \times [H] \times [K]$ *such that* $\check{n}_h^k(s, a) \neq 0$*, it holds that:*

$$\frac{n_h^k(s, a)}{\check{n}_h^k(s, a)} \leq 4H$$

*Proof.* For $\check{n}_h^k(s, a) \neq 0$, there exists $j \in \mathbb{N}_+$ such that $\check{n}_h^k(s, a) = e_j$ and $n_h^k(s, a) = \sum_{i=1}^j e_i$. We will use the mathematical induction to prove that for any $j \in \mathbb{N}_+$, $\frac{\sum_{i=1}^j e_i}{e_j} \leq 4H$.

For $j = 1$, $\frac{\sum_{i=1}^j e_i}{e_j} = 1 \leq 4H$.

If $\frac{\sum_{i=1}^{j-1} e_i}{e_{j-1}} \leq 4H$, then for $j \in \mathbb{N}_+$ and $j \geq 2$, we have

$$e_j = \left\lfloor \left(1 + \frac{1}{H}\right) e_{j-1} \right\rfloor \geq \left(1 + \frac{1}{2H}\right) e_{j-1},$$

which implies:

$$\frac{\sum_{i=1}^j e_i}{e_j} = 1 + \frac{\sum_{i=1}^{j-1} e_i}{e_j} \leq 1 + \frac{\sum_{i=1}^{j-1} e_i}{(1 + \frac{1}{2H})e_{j-1}} \leq 1 + \frac{4H}{1 + \frac{1}{2H}} \leq 4H.$$

Therefore, we finish the proof.

$\square$

**Lemma D.4.** *For any non-negative weight sequence* $\{\omega_{h,k}\}_{h,k}$ *and* $\alpha \in (0, 1)$*, it holds that:*

$$\sum_{k=1}^K \frac{\omega_{h,k}\mathbb{I}[n_h^k(s_h^k, a_h^k) \neq 0]}{n_h^k(s_h^k, a_h^k)^\alpha} \leq \frac{2^{2-\alpha}}{1-\alpha}(SA\|\omega\|_{\infty,h})^\alpha \|\omega\|_{1,h}^{1-\alpha},$$

*and*

$$\sum_{k=1}^K \frac{\omega_{h,k}\mathbb{I}[\check{n}_h^k(s_h^k, a_h^k) \neq 0]}{\check{n}_h^k(s_h^k, a_h^k)^\alpha} \leq \frac{2^{2+\alpha}H^\alpha}{1-\alpha}(SA\|\omega\|_{\infty,h})^\alpha \|\omega\|_{1,h}^{1-\alpha}.$$

*Here,* $\|\omega\|_{\infty,h} = \max_k\{\omega_{h,k}\}$ *and* $\|\omega\|_{1,h} = \sum_{k=1}^K \omega_{h,k}$.

*For* $\alpha = 1$*, we have the following conclusions:*

$$\sum_{k=1}^K \frac{\mathbb{I}[n_h^k(s_h^k, a_h^k) \neq 0]}{n_h^k(s_h^k, a_h^k)} \leq 2SA\log(T),$$

*and*

$$\sum_{k=1}^K \frac{\mathbb{I}[\check{n}_h^k(s_h^k, a_h^k) \neq 0]}{\check{n}_h^k(s_h^k, a_h^k)} \leq 4SAH\log(T).$$

*Proof.*

$$\sum_{k=1}^K \frac{\omega_{h,k}\mathbb{I}[n_h^k(s_h^k, a_h^k) \neq 0]}{n_h^k(s_h^k, a_h^k)^\alpha} = \sum_{s,a}\sum_{k=1}^K \frac{\omega_{h,k}\mathbb{I}[n_h^k(s, a) \neq 0, (s_h^k, a_h^k) = (s, a)]}{n_h^k(s, a)^\alpha}$$

$$\triangleq \sum_{s,a}\sum_{k=1}^K \frac{\omega'_{h,k}(s, a)}{n_h^k(s, a)^\alpha} \tag{36}$$

Here we let $\omega'_{h,k}(s, a) = \omega_{h,k}\mathbb{I}[n_h^k(s, a) \neq 0, (s_h^k, a_h^k) = (s, a)]$ and $c_h(s, a) = \sum_{k=1}^K \omega'_{h,k}(s, a)$. Then $\omega'_{h,k}(s, a) \leq \|\omega\|_{\infty,h}$ and $\sum_{s,a} c_h(s, a) \leq \sum_{k=1}^K \omega_{h,k} = \|\omega\|_{1,h}$.

Because $n_h^k(s,a)$ is nondecreasing for $1 \leq k \leq K$, given the term $\sum_{k=1}^K \frac{\omega_{h,k}'}{n_h^k(s,a)^\alpha}$, when the weights $\omega_{h,k}'(s,a)$ concentrates on former terms, we can obtain the largest value. For a given state-action pair $(s,a)$ and $j \in \mathbb{N}_+$, according to the stage design, the set $\{k : n_h^k(s,a) = \sum_{i=1}^j e_i\}$ has at most $e_{j+1} \leq (1+\frac{1}{H})e_j$ elements. Thus, the upper bound for the sum of the coefficients of $n_h^k(s,a) = \sum_{i=1}^j e_i$ in Equation (36) is given by $(1+\frac{1}{H})e_j \|\omega\|_{\infty,h}$.

Let:

$$k_0 = \max \left\{ k : \sum_{j=1}^{k-1} \left(1+\frac{1}{H}\right) e_j \|\omega\|_{\infty,h} < c_h(s,a), k \in \mathbb{N}_+ \right\}.$$

Because $e_{j+1} \leq (1+\frac{1}{H})e_j$ for any $j \in \mathbb{N}_+$, we have

$$\sum_{j=2}^{k_0} e_j \|\omega\|_{\infty,h} < c_h(s,a),$$

and thus

$$\sum_{j=1}^{k_0} e_j \|\omega\|_{\infty,h} \leq \sum_{j=1}^{k_0-1} \left(1+\frac{1}{H}\right) e_j \|\omega\|_{\infty,h} + \sum_{j=2}^{k_0} e_j \|\omega\|_{\infty,h} < 2c_h(s,a). \tag{37}$$

Therefore, back to Equation (36), by concentrating the weight to the terms with $n_h^k(s,a) = \sum_{i=1}^j e_i$, $j \in \{1,2,...,k_0\}$, for any given state-action pair $(s,a) \in \mathcal{S} \times \mathcal{A}$, we have:

$$\sum_{k=1}^K \frac{\omega_{h,k}'}{n_h^k(s,a)^\alpha} \leq \sum_{j=1}^{k_0} \frac{(1+\frac{1}{H})e_j \|\omega\|_{\infty,h}}{\left(\sum_{i=1}^j e_i\right)^\alpha} = \left(1+\frac{1}{H}\right) \|\omega\|_{\infty,h} \left(\sum_{j=1}^{k_0} \frac{e_j}{\left(\sum_{i=1}^j e_i\right)^\alpha}\right). \tag{38}$$

For any $0 \leq y < x$ and $\alpha \in (0,1)$, we have:

$$\frac{x-y}{x^\alpha} \leq \frac{1}{1-\alpha}(x^{1-\alpha} - y^{1-\alpha}).$$

For any $j \in \mathbb{N}_+$, let $x = \sum_{i=1}^j e_i$ and $y = \sum_{i=1}^{j-1} e_i$, then we have:

$$\frac{e_j}{\left(\sum_{i=1}^j e_i\right)^\alpha} \leq \frac{1}{1-\alpha} \left( \left(\sum_{i=1}^j e_i\right)^{1-\alpha} - \left(\sum_{i=1}^{j-1} e_i\right)^{1-\alpha} \right).$$

Sum the above inequality from 1 to $k_0$, then it holds that:

$$\sum_{j=1}^{k_0} \frac{e_j}{\left(\sum_{i=1}^j e_i\right)^\alpha} \leq \frac{1}{1-\alpha} \left(\sum_{i=1}^{k_0} e_i\right)^{1-\alpha} < \frac{1}{1-\alpha} \left(\frac{2c_h(s,a)}{\|\omega\|_{\infty,h}}\right)^{1-\alpha}.$$

The last inequality is because of Equation (37). Applying this inequality to Equation (38), we have:

$$\sum_{k=1}^K \frac{\omega_{h,k}'}{n_h^k(s,a)^\alpha} \leq \frac{2^{2-\alpha}}{1-\alpha} \|\omega\|_{\infty,h}^\alpha c_h(s,a)^{1-\alpha}.$$

Using this inequality in Equation (36), we have:

$$\sum_{k=1}^K \frac{\omega_{h,k} \mathbb{I}[n_h^k(s,a) \neq 0]}{n_h^k(s_h^k,a_h^k)^\alpha} \leq \frac{2^{2-\alpha}}{1-\alpha} \|\omega\|_{\infty,h}^\alpha \sum_{s,a} c_h(s,a)^{1-\alpha} \leq \frac{2^{2-\alpha}}{1-\alpha} \left(SA\|\omega\|_{\infty,h}\right)^\alpha \|\omega\|_{1,h}^{1-\alpha}.$$

The last inequality holds due to Hölder's inequality, as $\sum_{s,a} c_h(s,a)^{1-\alpha} \leq (SA)^\alpha \|\omega\|_{1,h}^{1-\alpha}$.

By Lemma D.3, it is easy to prove prove the second conclusion:

$$\sum_{k=1}^K \frac{\omega_{h,k} \mathbb{I}[\check{n}_h^k(s_h^k,a_h^k) \neq 0]}{\check{n}_h^k(s_h^k,a_h^k)^\alpha} \leq \frac{2^{2+\alpha} H^\alpha}{1-\alpha} \left(SA\|\omega\|_{\infty,h}\right)^\alpha \|\omega\|_{1,h}^{1-\alpha}.$$

The case of $\alpha = 1$ is proved in Lemma 11 of Zhang et al. (2020). $\square$

**Lemma D.5.** *For any non-negative functions* $\{X_h^k : \mathcal{S} \to \mathbb{R} \mid k \in [K], h \in [H]\}$ *and any* $h \in [H]$, *we have that*

$$\sum_{k=1}^K \frac{\mathbb{I}\left[n_h^k(s_h^k, a_h^k) \neq 0\right]}{n_h^k(s_h^k, a_h^k)} \sum_{i=1}^{n_h^k} X_{h+1}^{l_i} \leq 3\log(T) \sum_{k=1}^K X_{h+1}^k,$$

$$\sum_{k=1}^K \frac{\mathbb{I}\left[\check{n}_h^k(s_h^k, a_h^k) \neq 0\right]}{\check{n}_h^k(s_h^k, a_h^k)} \sum_{i=1}^{\check{n}_h^k} X_{h+1}^{\check{l}_i} \leq \left(1 + \frac{1}{H}\right) \sum_{k=1}^K X_{h+1}^k.$$

*Here,* $X_{H+1}^k = 0$ *for any* $k \in [K]$ *and* $s \in \mathcal{S}$.

*Proof.* For the first conclusion,

$$\sum_{k=1}^K \frac{\mathbb{I}\left[n_h^k(s_h^k, a_h^k) \neq 0\right]}{n_h^k(s_h^k, a_h^k)} \sum_{i=1}^{n_h^k} X_{h+1}^{l_i} = \sum_{k=1}^K \frac{\sum_{i=1}^{n_h^k} X_{h+1}^{l_i}}{n_h^k(s_h^k, a_h^k)} \cdot \sum_{j=1}^K \mathbb{I}\left[l_i = j, n_h^k(s_h^k, a_h^k) \neq 0\right]$$

$$= \sum_{k=1}^K \sum_{i=1}^{n_h^k} \sum_{j=1}^K \frac{X_{h+1}^j}{n_h^k(s_h^k, a_h^k)} \cdot \mathbb{I}\left[l_i = j, n_h^k(s_h^k, a_h^k) \neq 0\right]$$

$$= \sum_{j=1}^K \left( \sum_{k=1}^K \frac{\sum_{i=1}^{n_h^k} \mathbb{I}\left[l_i = j, n_h^k(s_h^k, a_h^k) \neq 0\right]}{n_h^k(s_h^k, a_h^k)} \right) X_{h+1}^j. \quad (39)$$

For a given episode $k$, according to the definition of $l_i$, $\sum_{i=1}^{n_h^k} \mathbb{I}\left[l_i = j, n_h^k(s_h^k, a_h^k) \neq 0\right] = 1$ if and only if $(s_h^k, a_h^k) = (s_h^j, a_h^j)$ and $(j, h)$ falls in the stage before that $(k, h)$ falls in. As a result, if the $(k, h)$ belongs to stage $t$ of $(s_h^k, a_h^k, h)$, we have $n_h^k(s_h^k, a_h^k) = \sum_{i=1}^{t-1} e_i$ and the set $\{k : \sum_{i=1}^{n_h^k} \mathbb{I}\left[l_i = j, n_h^k(s_h^k, a_h^k) \neq 0\right] = 1\}$ has at most $e_t$ elements. Then it holds that:

$$\sum_{k=1}^K \frac{\sum_{i=1}^{n_h^k} \mathbb{I}\left[l_i = j, n_h^k(s_h^k, a_h^k) \neq 0\right]}{n_h^k(s_h^k, a_h^k)} \leq \sum_{t \in C} \frac{e_t}{\sum_{i=1}^{t-1} e_i} \leq \sum_{t \in C} \sum_{p=1}^{e_t} \frac{3}{\sum_{i=1}^{t-1} e_i + p} \leq 3\log(T) \quad (40)$$

Here, $C = \{j : H \leq \sum_{i=1}^{t-1} e_i \leq T, t \in \mathbb{N}_+\}$. The second inequality is because $e_t \leq (1 + \frac{1}{H})e_{t-1}$ and then for any $p \in [e_t]$, $\sum_{i=1}^{t-1} e_i + p \leq 3 \sum_{i=1}^{t-1} e_i$. Then we finish the proof of the first conclusion. For the second conclusion,

$$\sum_{k=1}^K \frac{\mathbb{I}\left[\check{n}_h^k(s_h^k, a_h^k) \neq 0\right]}{\check{n}_h^k(s_h^k, a_h^k)} \sum_{i=1}^{\check{n}_h^k} X_{h+1}^{\check{l}_i} = \sum_{k=1}^K \frac{\sum_{i=1}^{\check{n}_h^k} X_{h+1}^{\check{l}_i}}{\check{n}_h^k(s_h^k, a_h^k)} \cdot \sum_{j=1}^K \mathbb{I}\left[\check{l}_i = j, \check{n}_h^k(s_h^k, a_h^k) \neq 0\right]$$

$$= \sum_{k=1}^K \sum_{i=1}^{\check{n}_h^k} \sum_{j=1}^K \frac{X_{h+1}^j}{\check{n}_h^k(s_h^k, a_h^k)} \cdot \mathbb{I}\left[\check{l}_i = j, \check{n}_h^k(s_h^k, a_h^k) \neq 0\right]$$

$$= \sum_{j=1}^K \left( \sum_{k=1}^K \frac{\sum_{i=1}^{\check{n}_h^k} \mathbb{I}\left[\check{l}_i = j, \check{n}_h^k(s_h^k, a_h^k) \neq 0\right]}{\check{n}_h^k(s_h^k, a_h^k)} \right) X_{h+1}^j. \quad (41)$$

For a given episode $k$, according to the definition of $\check{l}_i$, $\sum_{i=1}^{n_h^k} \mathbb{I}\left[\check{l}_i = j, \check{n}_h^k(s_h^k, a_h^k) \neq 0\right] = 1$ if and only if $(s_h^k, a_h^k) = (s_h^j, a_h^j)$ and $(j, h)$ falls in the previous stage of that $(k, h)$ falls in. As a result, in the stage of $(j, h)$, the number of visits to $(s_h^k, a_h^k, h)$ is $\check{n}_h^k(s_h^k, a_h^k)$, and the set $\{k : \sum_{i=1}^{n_h^k} \mathbb{I}\left[\check{l}_i = j, \check{n}_h^k \neq 0\right] = 1\}$ has at most $(1 + \frac{1}{H})\check{n}_h^k(s_h^k, a_h^k)$ elements. Then it holds that:

$$\sum_{k=1}^K \frac{\sum_{i=1}^{\check{n}_h^k} \mathbb{I}\left[\check{l}_i = j, \check{n}_h^k \neq 0\right]}{\check{n}_h^k(s_h^k, a_h^k)} \leq 1 + \frac{1}{H} \quad (42)$$

Therefore, we prove the second conclusion. $\qquad\square$

### D.3 PROOF SKETCH OF THEOREM 3.1

Next, we will begin to prove Theorem 3.1 under $\bigcap_{i=1}^{8} \mathcal{E}_i$. Let $\mathcal{X} = (\mathcal{S}, \mathcal{A}, H, T, \iota)$. The notation $f(\mathcal{X}) \lesssim g(\mathcal{X})$ means that there exists a universal constant $C_1 > 0$ such that $f(\mathcal{X}) \leq C_1 g(\mathcal{X})$.

**Step 1: Bounding the term $Q_h^k - Q_h^\star$.** By Equation (30) and Bellman Optimality Equation (5), it holds that:

$$
Q_h^k(s_h^k, a_h^k) - Q_h^\star(s_h^k, a_h^k)
$$

$$
\leq \mathbb{I}\left[n_h^k \neq 0\right]\left(\frac{\sum_{i=1}^{n_h^k} V_{h+1}^{\mathrm{ref},l_i}(s_{h+1}^{l_i})}{n_h^k(s_h^k, a_h^k)} + \frac{\sum_{i=1}^{\check{n}_h^k}\left(V_{h+1}^{\check{l}_i} - V_{h+1}^{\mathrm{ref},\check{l}_i}\right)(s_{h+1}^{\check{l}_i})}{\check{n}_h^k(s_h^k, a_h^k)} + b_h^k(s_h^k, a_h^k)\right)
$$

$$
+ \mathbb{I}\left[n_h^k = 0\right] H - \mathbb{P}_{s_h^k, a_h^k, h} V_{h+1}^\star
$$

$$
\leq \mathbb{I}\left[n_h^k \neq 0\right]\left(\frac{\sum_{i=1}^{n_h^k} V_{h+1}^{\mathrm{ref},l_i}(s_{h+1}^{l_i})}{n_h^k(s_h^k, a_h^k)} + \frac{\sum_{i=1}^{\check{n}_h^k}\left(V_{h+1}^{\check{l}_i} - V_{h+1}^{\mathrm{REF}}\right)(s_{h+1}^{\check{l}_i})}{\check{n}_h^k(s_h^k, a_h^k)} + b_h^k(s_h^k, a_h^k)\right)
$$

$$
+ \mathbb{I}\left[n_h^k = 0\right] H - \mathbb{I}\left[\check{n}_h^k \neq 0\right] \mathbb{P}_{s_h^k, a_h^k, h} V_{h+1}^\star
$$

$$
= \mathbb{I}\left[n_h^k = 0\right] H + \mathbb{I}\left[n_h^k \neq 0\right]\left(G_1 + b_h^k(s_h^k, a_h^k)\right) + \mathbb{I}\left[\check{n}_h^k \neq 0\right]\left(G_2 + G_3\right)
$$

The second inequality is because $V_{h+1}^{\mathrm{ref},\check{l}_i}(s_{h+1}^{\check{l}_i}) \geq V_{h+1}^{\mathrm{REF}}(s_{h+1}^{\check{l}_i})$. In the last equality we use $\mathbb{I}\left[n_h^k(s_h^k, a_h^k) = 0\right] = \mathbb{I}\left[\check{n}_h^k(s_h^k, a_h^k) = 0\right]$. Here

$$
G_1 = \frac{\sum_{i=1}^{n_h^k}\left(V_{h+1}^{\mathrm{ref},l_i}(s_{h+1}^{l_i}) - \mathbb{P}_{s_h^k, a_h^k, h} V_{h+1}^{\mathrm{REF}}\right)}{n_h^k(s_h^k, a_h^k)},
$$

$$
G_2 = \frac{\sum_{i=1}^{\check{n}_h^k}\left(\mathbb{P}_{s_h^k, a_h^k, h} - \mathbb{1}_{s_{h+1}^{\check{l}_i}}\right)\left(V_{h+1}^{\mathrm{REF}} - V_{h+1}^\star\right)}{\check{n}_h^k(s_h^k, a_h^k)},
$$

$$
G_3 = \frac{\sum_{i=1}^{\check{n}_h^k}\left(V_{h+1}^{\check{l}_i}(s_{h+1}^{\check{l}_i}) - V_{h+1}^\star(s_{h+1}^{\check{l}_i})\right)}{\check{n}_h^k(s_h^k, a_h^k)}.
$$

The upper bounds of $G_1$, $G_2$ and $b_h^k$ is given in Appendix D.4. Combining the three upper bounds Equation (53), Equation (57) and Equation (62), the following inequality holds:

$$
(Q_h^k - Q_h^\star)(s_h^k, a_h^k) \lesssim \mathbb{I}\left[\check{n}_h^k \neq 0\right]\left(G_3 + \frac{H\iota^{\frac{3}{4}}}{\check{n}_h^k(s_h^k, a_h^k)^{\frac{3}{4}}}\right) + \mathbb{I}\left[n_h^k \neq 0\right]\sqrt{\frac{(\mathbb{Q}^\star + \beta^2 H)\iota}{n_h^k(s_h^k, a_h^k)}} + Y_h^k.
\tag{43}
$$

Here, for any $h' \in [H]$ and $k \in [K]$, $Y_{h'}^k$ is defined as:

$$
Y_{h'}^k = H\mathbb{I}\left[n_{h'}^k = 0\right] + \frac{\mathbb{I}\left[n_{h'}^k \neq 0\right]}{n_{h'}^k(s_{h'}^k, a_{h'}^k)}\left(\sum_{i=1}^{n_{h'}^k} H\left(\mathbb{1}_{s_{h'+1}^{l_i}} + \mathbb{P}_{s_{h'}^k, a_{h'}^k, h'}\right)\lambda_{h'+1}^{l_i} + \sqrt{H\Gamma_{h'}^k(s_{h'}^k, a_{h'}^k)\iota}\right)
$$

$$
+ \frac{\mathbb{I}\left[\check{n}_{h'}^k \neq 0\right]}{\check{n}_{h'}^k(s_{h'}^k, a_{h'}^k)}\left(\sum_{i=1}^{\check{n}_{h'}^k} H\left(\mathbb{P}_{s_{h'}^k, a_{h'}^k, h'} + \mathbb{1}_{s_{h'+1}^{\check{l}_i}}\right)\lambda_{h'+1}^{\check{l}_i} + \sqrt{H\check{\Gamma}_{h'}^k(s_{h'}^k, a_{h'}^k)\iota} + H\iota\right),
$$

where

$$
\Gamma_{h'}^k(s_{h'}^k, a_{h'}^k) = \sum_{i=1}^{n_{h'}^k}\left(V_{h'+1}^{\mathrm{ref},l_i}(s_{h'+1}^{l_i}) - \hat{V}_{h'+1}^{\mathrm{ref},l_i}(s_{h'+1}^{l_i})\right)
$$

and

$$
\check{\Gamma}_{h'}^k(s_{h'}^k, a_{h'}^k) = \sum_{i=1}^{\check{n}_{h'}^k}\left(V_{h'+1}^{\mathrm{ref},\check{l}_i}(s_{h'+1}^{\check{l}_i}) - \hat{V}_{h'+1}^{\mathrm{ref},\check{l}_i}(s_{h'+1}^{\check{l}_i})\right).
$$

**Step 2: Bounding the weighted sum.** For any given $h$ and non-negative constants $\{\omega_{h,k}\}_{h,[K]}$, we denote $\|\omega\|_{\infty,h} = \max_{k \in [K]} \omega_{h,k}$ and $\|\omega\|_{1,h} = \sum_{k \in [K]} \omega_{h,k}$. We also recursively define $\omega_{h',k}(h)$ for any $h \le h' \le H, k \in [K], j \in [K]$ as follows:

$$\omega_{h,k}(h) := \omega_{h,k}; \ \omega_{h'+1,j}(h) = \sum_{k=1}^{K} \omega_{h',k}(h) \frac{\sum_{i=1}^{\check{n}_{h'}^{k}} \mathbb{I}[\check{l}_i = j, \check{n}_{h'}^k \ne 0]}{\check{n}_{h'}^k(s_{h'}^k, a_{h'}^k)}. \tag{44}$$

By Equation (42), it is easy to show that

$$\|\omega(h)\|_{1,h'+1} \le \|\omega(h)\|_{1,h'}, \ \|\omega(h)\|_{\infty,h'+1} \le (1 + 1/H)\|\omega(h)\|_{\infty,h'}, \forall h' > h, \tag{45}$$

where

$$\|\omega(h))\|_{\infty,h'} = \max_k \omega_{h',k}(h) \le 1, \ \|\omega(h)\|_{1,h'} = \sum_{k=1}^{K} \omega_{h',k}(h).$$

Now given the weight $\{\omega_{h,k}\}_k$, we will bound the weighted sum $\sum_{k=1}^{K} \omega_{h,k}(Q_h^k - Q_h^\star)(s_h^k, a_h^k)$. Summing Equation (43) from 1 to $K$ with the weight $\{\omega_{h,k}\}_k$, we have:

$$\sum_{k=1}^{K} \omega_{h,k}(Q_h^k(s_h^k, a_h^k) - Q_h^\star(s_h^k, a_h^k))$$

$$\le \sum_{k=1}^{K} \omega_{h,k}\mathbb{I}\left[\check{n}_h^k \ne 0\right] G_3 + \sum_{k=1}^{K} \omega_{h,k}\left(\mathbb{I}\left[n_h^k \ne 0\right]\sqrt{\frac{(\mathbb{Q}^\star + \beta^2 H)\iota}{n_h^k(s_h^k, a_h^k)}} + \mathbb{I}\left[\check{n}_h^k \ne 0\right]\frac{H\iota^{\frac{3}{4}}}{\check{n}_h^k(s_h^k, a_h^k)^{\frac{3}{4}}}\right)$$

$$+ \sum_{k=1}^{K} \omega_{h,k}Y_h^k.$$

$$\lesssim \sum_{j=1}^{K} \omega_{h+1,j}(h)\left(Q_{h+1}^j - Q_{h+1}^\star\right)(s_{h+1}^j, a_{h+1}^j) + \sqrt{(\mathbb{Q}^\star + \beta^2 H)SA\|\omega\|_{\infty,h}\|\omega\|_{1,h}\iota}$$

$$+ H^{\frac{7}{4}}(SA\|\omega\|_{\infty,h}\iota)^{\frac{3}{4}}\|\omega\|_{1,h}^{\frac{1}{4}} + \sum_{k=1}^{K} \omega_{h,k}Y_h^k. \tag{46}$$

In the last inequality, the upper bound of $\sum_{k=1}^{K} \omega_{h,k}\mathbb{I}\left[\check{n}_h^k \ne 0\right] G_3$ is given in Appendix D.5. The upper bounds of the middle two terms is given by Lemma D.4 with $\alpha = \frac{1}{2}$ and $\alpha = \frac{3}{4}$.

Recurring Equation (46) for $h, h+1, ..., H$, since $Q_{H+1}^k(s,a) = Q_{H+1}^\star(s,a) = 0$, we have:

$$\sum_{k=1}^{K} \omega_{h,k}(Q_h^k(s_h^k, a_h^k) - Q_h^\star(s_h^k, a_h^k))$$

$$\lesssim H\sqrt{(\mathbb{Q}^\star + \beta^2 H)SA\|\omega\|_{\infty,h}\|\omega\|_{1,h}\iota} + H^{\frac{11}{4}}(SA\|\omega\|_{\infty,h}\iota)^{\frac{3}{4}}\|\omega\|_{1,h}^{\frac{1}{4}} + \sum_{h'=h}^{H}\sum_{k=1}^{K} \omega_{h',k}(h)Y_{h'}^k, \tag{47}$$

where $\omega_{h',k}(h)$ is defined in Equation (44).

**Step 3: Integrating multiple weighted sums.** For any $N = \lceil\log_2(H/\Delta_{\min})\rceil$, $n \in [N]$, $k \in [K]$ and the given $h \in [H]$, let:

$$\omega_{h,k}^{(n)} = \mathbb{I}\left[Q_h^k(s_h^k, a_h^k) - Q_h^\star(s_h^k, a_h^k) \in [2^{n-1}\Delta_{\min}, 2^n\Delta_{\min}]\right],$$

and

$$\omega_{h,k}^{(N)} = \mathbb{I}\left[Q_h^k(s_h^k, a_h^k) - Q_h^\star(s_h^k, a_h^k) \in [2^{N-1}\Delta_{\min}, H]\right].$$

We also denote

$$\|\omega^{(n)}\|_{\infty,h} = \max_k \omega_{h,k}^{(n)} \le 1, \ \|\omega^{(n)}\|_{1,h} = \sum_{k=1}^{K} \omega_{h,k}^{(n)}.$$

For $h \leq h' \leq H$ and any $n \in [N]$, the weight $\{\omega_{h',k}^{(n)}\}_k$ can be defined recursively by Equation (44):

$$\omega_{h,j}^{(n)}(h) = \omega_{h,j}^{(n)}; \; \omega_{h'+1,j}^{(n)}(h) = \sum_{k=1}^{K} \omega_{h',k}^{(n)}(h) \frac{\sum_{i=1}^{\check{n}_{h'}^{k}} \mathbb{I}[\check{l}_i = j, \check{n}_{h'}^{k} \neq 0]}{\check{n}_{h'}^{k}(s_{h'}^{k}, a_{h'}^{k})}.$$

Therefore, for any $j \in [K]$, it holds that:

$$\sum_{n=1}^{N} \omega_{h'+1,j}^{(n)}(h) = \sum_{k=1}^{K} \left( \sum_{n=1}^{N} \omega_{h',k}^{(n)}(h) \right) \frac{\sum_{i=1}^{\check{n}_{h'}^{k}} \mathbb{I}[\check{l}_i = j, \check{n}_{h'}^{k} \neq 0]}{\check{n}_{h'}^{k}(s_{h'}^{k}, a_{h'}^{k})}.$$

By mathematical induction on $h' \in [h, H]$, it is straightforward to prove that for any $j \in [K]$,

$$\sum_{n=1}^{N} \omega_{h',j}^{(n)}(h) \leq \left( 1 + \frac{1}{H} \right)^{h'-h} < 3, \tag{48}$$

given that for any $j \in [K]$

$$\sum_{k=1}^{K} \frac{\sum_{i=1}^{\check{n}_{h'}^{k}} \mathbb{I}[\check{l}_i = j, \check{n}_{h'}^{k} \neq 0]}{\check{n}_{h'}^{k}(s_{h'}^{k}, a_{h'}^{k})} \leq 1 + \frac{1}{H}$$

by Equation (42) and $\sum_{n=1}^{N} \omega_{h,j}^{(n)}(h) = \sum_{n=1}^{N} \omega_{h,j}^{(n)} \leq 1$ for $h' = h$.

Applying the weight $\{\omega_{h,k}^{(n)}\}_k$ to Equation (47), for any $n \in [N]$, it holds that:

$$\sum_{k=1}^{K} \omega_{h,k}^{(n)}(Q_h^k(s_h^k, a_h^k) - Q_h^\star(s_h^k, a_h^k))$$

$$\lesssim H\sqrt{(\mathbb{Q}^\star + \beta^2 H)SA\|\omega^{(n)}\|_{1,h}\iota} + H^{\frac{11}{4}}(SA\iota)^{\frac{3}{4}}\|\omega^{(n)}\|_{1,h}^{\frac{1}{4}} + \sum_{h'=h}^{H}\sum_{k=1}^{K} \omega_{h',k}^{(n)}(h)Y_{h'}^{k}.$$

On the other hand, according to the definition of $\omega_{h,k}^{(n)}$,

$$\sum_{k=1}^{K} \omega_{h,k}^{(n)}\left(Q_h^k(s_h^k, a_h^k) - Q_h^\star(s_h^k, a_h^k)\right) \geq 2^{n-1}\Delta_{\min}\|\omega^{(n)}\|_{1,h}.$$

Therefore, we obtain the following inequality:

$$2^{n-1}\Delta_{\min}\|\omega^{(n)}\|_{1,h}$$

$$\lesssim H\sqrt{(\mathbb{Q}^\star + \beta^2 H)SA\|\omega^{(n)}\|_{1,h}\iota} + H^{\frac{11}{4}}(SA\iota)^{\frac{3}{4}}\|\omega^{(n)}\|_{1,h}^{\frac{1}{4}} + \sum_{h'=h}^{H}\sum_{k=1}^{K} \omega_{h',k}^{(n)}(h)Y_{h'}^{k}. \tag{49}$$

Then at least one of the following three inequalities holds:

$$2^{n-1}\Delta_{\min}\|\omega^{(n)}\|_{1,h} \lesssim H\sqrt{(\mathbb{Q}^\star + \beta^2 H)SA\|\omega^{(n)}\|_{1,h}\iota},$$

$$2^{n-1}\Delta_{\min}\|\omega^{(n)}\|_{1,h} \lesssim H^{\frac{11}{4}}(SA\iota)^{\frac{3}{4}}(\|\omega^{(n)}\|_{1,h})^{\frac{1}{4}},$$

$$2^{n-1}\Delta_{\min}\|\omega^{(n)}\|_{1,h} \lesssim \sum_{h'=h}^{H}\sum_{k=1}^{K} \omega_{h',k}^{(n)}(h)Y_{h'}^{k}.$$

Solving this three inequalities, we know that:

$$\|\omega^{(n)}\|_{1,h} \leq O\left( \max\left\{ \frac{(\mathbb{Q}^\star + \beta^2 H)SAH^2\iota}{4^{n-2}\Delta_{\min}^2}, \frac{H^{\frac{11}{3}}SA\iota}{(2^{n-1}\Delta_{\min})^{\frac{4}{3}}}, \frac{\sum_{h'=h}^{H}\sum_{k=1}^{K}\omega_{h',k}^{(n)}(h)Y_{h'}^{k}}{2^{n-1}\Delta_{\min}} \right\} \right)$$

$$\leq O\left( \frac{(\mathbb{Q}^\star + \beta^2 H)SAH^2\iota}{4^{n-2}\Delta_{\min}^2} + \frac{H^{\frac{11}{3}}SA\iota}{(2^{n-1}\Delta_{\min})^{\frac{4}{3}}} + \frac{\sum_{h'=h}^{H}\sum_{k=1}^{K}\omega_{h',k}^{(n)}(h)Y_{h'}^{k}}{2^{n-1}\Delta_{\min}} \right).$$

By Equation (48), we have:

$$\sum_{n=1}^{N}\sum_{h'=h}^{H}\sum_{k=1}^{K}\omega_{h',k}^{(n)}(h)Y_{h'}^{k} = \sum_{h'=h}^{H}\sum_{k=1}^{K}\left(\sum_{n=1}^{N}\omega_{h',k}^{(n)}(h)\right)Y_{h'}^{k} \le 3\sum_{h'=1}^{H}\sum_{k=1}^{K}Y_{h'}^{k}.$$

Therefore,

$$\sum_{n=1}^{N}2^{n}\Delta_{\min}\|\omega^{(n)}\|_{1,h} \le O\left(\frac{\left(\mathbb{Q}^{\star}+\beta^{2}H\right)SAH^{2}\iota}{\Delta_{\min}} + \frac{H^{\frac{11}{3}}SA\iota}{(\Delta_{\min})^{\frac{1}{3}}} + \sum_{h'=1}^{H}\sum_{k=1}^{K}Y_{h'}^{k}\right). \tag{50}$$

From Appendix D.6, we know $\sum_{h'=1}^{H}\sum_{k=1}^{K}Y_{h'}^{k}$ can be bounded by $O(\frac{H^{7}S^{2}A\iota\log(T)}{\beta^{2}})$. Therefore, back to Equation (50), it holds that:

$$\sum_{n=1}^{N}2^{n}\Delta_{\min}\|\omega^{(n)}\|_{1,h} \le O\left(\frac{\left(\mathbb{Q}^{\star}+\beta^{2}H\right)SAH^{2}\iota}{\Delta_{\min}} + \frac{H^{\frac{11}{3}}SA\iota}{(\Delta_{\min})^{\frac{1}{3}}} + \frac{H^{7}S^{2}A\iota\log(T)}{\beta^{2}}\right)$$

$$\le O\left(\frac{\left(\mathbb{Q}^{\star}+\beta^{2}H\right)SAH^{2}\iota}{\Delta_{\min}} + \frac{H^{7}S^{2}A\iota\log(T)}{\beta^{2}}\right) \tag{51}$$

The last inequality is because:

$$\frac{H^{\frac{11}{3}}SA\iota}{(\Delta_{\min})^{\frac{1}{3}}} \lesssim \frac{\beta^{2}H^{3}SA\iota}{\Delta_{\min}} + \frac{H^{4}SA\iota}{\beta} + \frac{H^{4}SA\iota}{\beta} \lesssim \frac{\left(\mathbb{Q}^{\star}+\beta^{2}H\right)SAH^{2}\iota}{\Delta_{\min}} + \frac{H^{7}SA\iota\log(T)}{\beta^{2}}.$$

**Step 4: Bounding the expected gap-dependent regret.** Let $p = (40SAH^{2}T^{5})^{-1}$, then $\mathcal{E} = \bigcap_{i=1}^{7}\mathcal{E}_{i}$ holds with probability at least $1 - \frac{1}{T}$ and $\iota \lesssim \log(SAT)$. Therefore, by Equation (28), we have:

$$\mathbb{E}\left(\text{Regret}(K)\right) \le \mathbb{E}\left[\sum_{k=1}^{K}\sum_{h=1}^{H}\text{clip}[(Q_{h}^{k}-Q_{h}^{*})(s_{h}^{k},a_{h}^{k})\mid\Delta_{\min}]\right]$$

$$= \mathbb{E}\left[\sum_{k=1}^{K}\sum_{h=1}^{H}\text{clip}[(Q_{h}^{k}-Q_{h}^{*})(s_{h}^{k},a_{h}^{k})\mid\Delta_{\min}]\Big|\mathcal{E}\right]\mathbb{P}(\mathcal{E})$$

$$+ \mathbb{E}\left[\sum_{k=1}^{K}\sum_{h=1}^{H}\text{clip}[(Q_{h}^{k}-Q_{h}^{*})(s_{h}^{k},a_{h}^{k})\mid\Delta_{\min}]\Big|\mathcal{E}^{c}\right]\mathbb{P}(\mathcal{E}^{c})$$

$$\le \sum_{h=1}^{H}\sum_{n=1}^{N}2^{n}\Delta_{\min}\|\omega^{(n)}\|_{1,h} + \frac{1}{T}\cdot TH$$

$$\le O\left(\frac{\left(\mathbb{Q}^{\star}+\beta^{2}H\right)H^{3}SA\log(SAT)}{\Delta_{\min}} + \frac{H^{8}S^{2}A\log(SAT)\log(T)}{\beta^{2}}\right).$$

The last inequality is by Equation (51). The third inequality is because

$$\sum_{k=1}^{K}\text{clip}[(Q_{h}^{k}-Q_{h}^{*})(s_{h}^{k},a_{h}^{k})\mid\Delta_{\min}] = \sum_{k=1}^{K}\sum_{n=1}^{N}\omega_{h,k}^{(n)}(Q_{h}^{k}-Q_{h}^{*})(s_{h}^{k},a_{h}^{k})$$

$$\le \sum_{n=1}^{N}2^{n}\Delta_{\min}\sum_{k=1}^{K}\omega_{h,k}^{(n)} = \sum_{n=1}^{N}2^{n}\Delta_{\min}\|\omega^{(n)}\|_{1,h}.$$

## D.4 BOUNDING THE TERM $Q_{h}^{k} - Q_{h}^{\star}$

### D.4.1 BOUNDING THE TERM $G_{1}$

We can split $G_{1}$ into four terms:

$$\frac{\sum_{i=1}^{n_{h}^{k}}\left(V_{h+1}^{\text{ref},l_{i}}(s_{h+1}^{l_{i}}) - \mathbb{P}_{s_{h}^{k},a_{h}^{k},h}V_{h+1}^{\text{REF}}\right)}{n_{h}^{k}(s_{h}^{k},a_{h}^{k})} = G_{1,1}+G_{1,2}+G_{1,3}+G_{1,4}, \tag{52}$$

where

$$G_{1,1} = \frac{\sum_{i=1}^{n_h^k}\left(\mathbb{1}_{s_{h+1}^{l_i}} - \mathbb{P}_{s_h^k,a_h^k,h}\right)\left(V_{h+1}^{\mathrm{ref},l_i} - \hat{V}_{h+1}^{\mathrm{ref},l_i}\right)}{n_h^k(s_h^k,a_h^k)},$$

$$G_{1,2} = \frac{\sum_{i=1}^{n_h^k}\left(\mathbb{1}_{s_{h+1}^{l_i}} - \mathbb{P}_{s_h^k,a_h^k,h}\right)\left(\hat{V}_{h+1}^{\mathrm{ref},l_i} - V_{h+1}^{\star}\right)}{n_h^k(s_h^k,a_h^k)},$$

$$G_{1,3} = \frac{\sum_{i=1}^{n_h^k}\left(\mathbb{1}_{s_{h+1}^{l_i}} - \mathbb{P}_{s_h^k,a_h^k,h}\right)V_{h+1}^{\star}}{n_h^k(s_h^k,a_h^k)}$$

and

$$G_{1,4} = \frac{\sum_{i=1}^{n_h^k}\mathbb{P}_{s_h^k,a_h^k,h}\left(V_{h+1}^{\mathrm{ref},l_i} - V_{h+1}^{\mathrm{REF}}\right)}{n_h^k(s_h^k,a_h^k)}.$$

According to (c) in Lemma D.2, we have:

$$G_{1,1} \leq \frac{\sum_{i=1}^{n_h^k} H\left(\mathbb{1}_{s_{h+1}^{l_i}} + \mathbb{P}_{s_h^k,a_h^k,h}\right)\lambda_{h+1}^{l_i}}{n_h^k(s_h^k,a_h^k)}.$$

Under the event $\mathcal{E}_4$ in Lemma D.1, we can bound $G_{1,2}$:

$$G_{1,2} \leq \beta\sqrt{\frac{2\iota}{n_h^k(s_h^k,a_h^k)}}.$$

Under the event $\mathcal{E}_5$ in Lemma D.1, we can bound $G_{1,3}$:

$$G_{1,3} \leq 2\sqrt{\frac{2\mathbb{Q}^{\star}\iota}{n_h^k(s_h^k,a_h^k)}} + \frac{4H\iota}{n_h^k(s_h^k,a_h^k)}.$$

The upper bound of $G_{1,4}$ is given by (b) in Lemma D.2:

$$G_{1,4} \leq \frac{\sum_{i=1}^{n_h^k} H\mathbb{P}_{s_h^k,a_h^k,h}\lambda_{h+1}^{l_i}}{n_h^k(s_h^k,a_h^k)}.$$

Combining these four upper bounds together, we can bound $G_1$:

$$G_1 \lesssim \frac{\sum_{i=1}^{n_h^k} H\left(\mathbb{1}_{s_{h+1}^{l_i}} + \mathbb{P}_{s_h^k,a_h^k,h}\right)\lambda_{h+1}^{l_i}}{n_h^k(s_h^k,a_h^k)} + \sqrt{\frac{(\mathbb{Q}^{\star}+\beta^2)\iota}{n_h^k(s_h^k,a_h^k)}} + \frac{H\iota}{n_h^k(s_h^k,a_h^k)}. \tag{53}$$

### D.4.2 BOUNDING THE TERM $G_2$

We can split the term of $G_2$ into two terms:

$$G_2 = \frac{\sum_{i=1}^{\check{n}_h^k}\left(\mathbb{P}_{s_h^k,a_h^k,h} - \mathbb{1}_{s_{h+1}^{\check{l}_i}}\right)\left[\left(V_{h+1}^{\mathrm{REF}} - \hat{V}_{h+1}^{\mathrm{ref},\check{l}_i}\right) + \left(\hat{V}_{h+1}^{\mathrm{ref},\check{l}_i} - V_{h+1}^{\star}\right)\right]}{\check{n}_h^k(s_h^k,a_h^k)}. \tag{54}$$

According to (d) in Lemma D.2, we can bound the first term in Equation (54):

$$\frac{\sum_{i=1}^{\check{n}_h^k}\left(\mathbb{P}_{s_h^k,a_h^k,h} - \mathbb{1}_{s_{h+1}^{\check{l}_i}}\right)\left(V_{h+1}^{\mathrm{REF}} - \hat{V}_{h+1}^{\mathrm{ref},\check{l}_i}\right)}{\check{n}_h^k(s_h^k,a_h^k)} \leq \frac{\sum_{i=1}^{\check{n}_h^k} H\left(\mathbb{P}_{s_h^k,a_h^k,h} + \mathbb{1}_{s_{h+1}^{\check{l}_i}}\right)\lambda_{h+1}^{\check{l}_i}}{\check{n}_h^k(s_h^k,a_h^k)}. \tag{55}$$

The upper bound for the second term in Equation (54) is given by the event $\mathcal{E}_6$ in Lemma D.1:

$$\frac{\sum_{i=1}^{\check{n}_h^k}\left(\mathbb{P}_{s,a,h} - \mathbb{1}_{s_{h+1}^{\check{l}_i}}\right)\left(\hat{V}_{h+1}^{\mathrm{ref},\check{l}_i} - V_{h+1}^{\star}\right)}{\check{n}_h^k(s,a)} \leq \sqrt{\frac{2\beta^2\iota}{\check{n}_h^k(s,a)}} \lesssim \sqrt{\frac{\beta^2 H\iota}{n_h^k(s,a)}}. \tag{56}$$

The last inequality is because of Lemma D.3. Applying Equation (55) and Equation (56) to Equation (54), we have:

$$G_2 \lesssim \frac{\sum_{i=1}^{\check{n}_h^k} H\left(\mathbb{P}_{s_h^k,a_h^k,h} + \mathbb{1}_{s_{h+1}^{\check{l}_i}}\right)\lambda_{h+1}^{\check{l}_i}}{\check{n}_h^k(s_h^k,a_h^k)} + \sqrt{\frac{\beta^2 H\iota}{n_h^k(s,a)}}. \tag{57}$$

### D.4.3  Bounding the term $b_h^k(s_h^k, a_h^k)$

According to the definition of $b_h^k(s_h^k, a_h^k)$ in Appendix D.1, we have

$$b_h^k(s_h^k, a_h^k) \lesssim \sqrt{\frac{\nu_h^{\mathrm{ref},k}\iota}{n_h^k}} + \sqrt{\frac{\check{\nu}_h^k\iota}{\check{n}_h^k}} + \left( \frac{H\iota}{n_h^k} + \frac{H\iota}{\check{n}_h^k} + \frac{H\iota^{\frac{3}{4}}}{(n_h^k)^{\frac{3}{4}}} + \frac{H\iota^{\frac{3}{4}}}{(\check{n}_h^k)^{\frac{3}{4}}} \right), \tag{58}$$

where $\nu_h^{\mathrm{ref},k} = \sigma_h^{\mathrm{ref},k}/n_h^k - (\mu_h^{\mathrm{ref},k}/n_h^k)^2$ and $\check{\nu}_h^{\mathrm{ref},k} = \check{\sigma}_h^k/\check{n}_h^k - (\check{\mu}_h^k/\check{n}_h^k)^2$.

Since $V_{h+1}^{\mathrm{ref},l_i}(s_{h+1}^{l_i}) \geq \hat{V}_{h+1}^{\mathrm{ref},l_i}(s_{h+1}^{l_i})$, it holds that

$$\sqrt{\frac{\nu_h^{\mathrm{ref},k}\iota}{n_h^k}} = \sqrt{\frac{\frac{\sigma_h^{\mathrm{ref},k}(s_h^k,a_h^k)}{n_h^k(s_h^k,a_h^k)} - \left(\frac{\mu_h^{\mathrm{ref},k}(s_h^k,a_h^k)}{n_h^k(s_h^k,a_h^k)}\right)^2}{n_h^k(s_h^k,a_h^k)}}\,\iota \leq \sqrt{\frac{I_1^{h,k} + I_2^{h,k}}{n_h^k(s_h^k,a_h^k)}}\,\iota,$$

where:

$$I_1^{h,k} = \frac{\sum_{i=1}^{n_h^k} \left( \left(V_{h+1}^{\mathrm{ref},l_i}(s_{h+1}^{l_i})\right)^2 - \left(\hat{V}_{h+1}^{\mathrm{ref},l_i}(s_{h+1}^{l_i})\right)^2 \right)}{n_h^k(s_h^k, a_h^k)},$$

and

$$I_2^{h,k} = \frac{\sum_{i=1}^{n_h^k} \left(\hat{V}_{h+1}^{\mathrm{ref},l_i}(s_{h+1}^{l_i})\right)^2}{n_h^k(s_h^k, a_h^k)} - \left( \frac{\sum_{i=1}^{n_h^k} \hat{V}_{h+1}^{\mathrm{ref},l_i}(s_{h+1}^{l_i})}{n_h^k(s_h^k, a_h^k)} \right)^2.$$

Next we want to bound both $I_1^{h,k}$ and $I_2^{h,k}$.

$$I_1^{h,k} = \frac{\sum_{i=1}^{n_h^k} \left( V_{h+1}^{\mathrm{ref},l_i}(s_{h+1}^{l_i}) + \hat{V}_{h+1}^{\mathrm{ref},l_i}(s_{h+1}^{l_i}) \right) \left( V_{h+1}^{\mathrm{ref},l_i}(s_{h+1}^{l_i}) - \hat{V}_{h+1}^{\mathrm{ref},l_i}(s_{h+1}^{l_i}) \right)}{n_h^k(s_h^k, a_h^k)}$$

$$\leq \frac{\sum_{i=1}^{n_h^k} 2H\left( V_{h+1}^{\mathrm{ref},l_i}(s_{h+1}^{l_i}) - \hat{V}_{h+1}^{\mathrm{ref},l_i}(s_{h+1}^{l_i}) \right)}{n_h^k(s_h^k, a_h^k)} \triangleq \frac{2H\Gamma_h^k(s_h^k, a_h^k)}{n_h^k(s_h^k, a_h^k)}, \tag{59}$$

where

$$\Gamma_h^k(s_h^k, a_h^k) = \sum_{i=1}^{n_h^k} \left( V_{h+1}^{\mathrm{ref},l_i}(s_{h+1}^{l_i}) - \hat{V}_{h+1}^{\mathrm{ref},l_i}(s_{h+1}^{l_i}) \right).$$

For the second term $I_2^{h,k}$, because of Cauchy's Inequality, we have:

$$I_2^{h,k} = \frac{\sum_{i=1}^{n_h^k} \left( \hat{V}_{h+1}^{\mathrm{ref},l_i}(s_{h+1}^{l_i}) - \frac{\sum_{n=1}^{n_h^k} \hat{V}_{h+1}^{\mathrm{ref},l_n}(s_{h+1}^{l_n})}{n_h^k(s_h^k,a_h^k)} \right)^2}{n_h^k(s_h^k, a_h^k)} \leq 2\left( I_{2,1}^{h,k} + I_{2,2}^{h,k} \right),$$

where:

$$I_{2,1}^{h,k} = \frac{\sum_{i=1}^{n_h^k} \left( \hat{V}_{h+1}^{\mathrm{ref},l_i}(s_{h+1}^{l_i}) - V_{h+1}^\star(s_{h+1}^{l_i}) + \frac{\sum_{n=1}^{n_h^k} V_{h+1}^\star(s_{h+1}^{l_n})}{n_h^k(s_h^k,a_h^k)} - \frac{\sum_{n=1}^{n_h^k} \hat{V}_{h+1}^{\mathrm{ref},l_n}(s_{h+1}^{l_n})}{n_h^k(s_h^k,a_h^k)} \right)^2}{n_h^k(s_h^k, a_h^k)},$$

and

$$I_{2,2}^{h,k} = \frac{\sum_{i=1}^{n_h^k} \left( V_{h+1}^\star(s_{h+1}^{l_i}) - \frac{\sum_{n=1}^{n_h^k} V_{h+1}^\star(s_{h+1}^{l_n})}{n_h^k(s_h^k,a_h^k)} \right)^2}{n_h^k(s_h^k, a_h^k)}$$

$$= \frac{\sum_{i=1}^{n_h^k} \left( V_{h+1}^\star(s_{h+1}^{l_i}) \right)^2}{n_h^k(s_h^k, a_h^k)} - \left( \frac{\sum_{i=1}^{n_h^k} V_{h+1}^\star(s_{h+1}^{l_i})}{n_h^k(s_h^k, a_h^k)} \right)^2.$$

Since $V_{h+1}^\star(s_{h+1}^{l_i}) \le \hat{V}_{h+1}^{\mathrm{ref},l_i}(s_{h+1}^{l_i}) \le V_{h+1}^\star(s_{h+1}^{l_i}) + \beta$, it holds that:

$$\left| \hat{V}_{h+1}^{\mathrm{ref},l_i}(s_{h+1}^{l_i}) - V_{h+1}^\star(s_{h+1}^{l_i}) + \frac{\sum_{n=1}^{n_h^k} V_{h+1}^\star(s_{h+1}^{l_n})}{n_h^k(s_h^k, a_h^k)} - \frac{\sum_{n=1}^{n_h^k} \hat{V}_{h+1}^{\mathrm{ref},l_n}(s_{h+1}^{l_n})}{n_h^k(s_h^k, a_h^k)} \right|$$

$$\le \left| \hat{V}_{h+1}^{\mathrm{ref},l_i}(s_{h+1}^{l_i}) - V_{h+1}^\star(s_{h+1}^{l_i}) \right| + \left| \frac{\sum_{n=1}^{n_h^k} V_{h+1}^\star(s_{h+1}^{l_n})}{n_h^k(s_h^k, a_h^k)} - \frac{\sum_{n=1}^{n_h^k} \hat{V}_{h+1}^{\mathrm{ref},l_n}(s_{h+1}^{l_n})}{n_h^k(s_h^k, a_h^k)} \right| \le 2\beta.$$

Using this inequality, we have $I_{2,1}^{h,k} \le 4\beta^2$. Moreover, according to the definition of $\mathbb{Q}^\star$, it holds

$$I_{2,2}^{h,k} - \mathbb{Q}^\star \le I_{2,2}^{h,k} - \left( \mathbb{P}_{s_h^k,a_h^k,h}(V_{h+1}^\star)^2 - \left( \mathbb{P}_{s_h^k,a_h^k,h} V_{h+1}^\star \right)^2 \right)$$

$$= - \left( \frac{\sum_{i=1}^{n_h^k} V_{h+1}^\star(s_{h+1}^{l_i})}{n_h^k(s_h^k, a_h^k)} + \mathbb{P}_{s_h^k,a_h^k,h} V_{h+1}^\star \right) \left( \frac{\sum_{i=1}^{n_h^k} \left( \mathbb{1}_{s_{h+1}^{l_i}} - \mathbb{P}_{s_h^k,a_h^k,h} \right) V_{h+1}^\star}{n_h^k(s_h^k, a_h^k)} \right)$$

$$+ \frac{\sum_{i=1}^{n_h^k} \left( \mathbb{1}_{s_{h+1}^{l_i}} - \mathbb{P}_{s_h^k,a_h^k,h} \right) (V_{h+1}^\star)^2}{n_h^k(s_h^k, a_h^k)}$$

$$\le 2H \left| \frac{\sum_{i=1}^{n_h^k} \left( \mathbb{1}_{s_{h+1}^{l_i}} - \mathbb{P}_{s_h^k,a_h^k,h} \right) V_{h+1}^\star}{n_h^k(s_h^k, a_h^k)} \right| + \left| \frac{\sum_{i=1}^{n_h^k} \left( \mathbb{1}_{s_{h+1}^{l_i}} - \mathbb{P}_{s_h^k,a_h^k,h} \right) (V_{h+1}^\star)^2}{n_h^k(s_h^k, a_h^k)} \right|$$

$$\lesssim H^2 \sqrt{\frac{\iota}{n_h^k(s_h^k, a_h^k)}}.$$

The last inequality is because of the event $\mathcal{E}_7$ and the event $\mathcal{E}_8$ in Lemma D.1. Therefore, we have

$$I_{2,2}^{h,k} \lesssim \mathbb{Q}^\star + H^2 \sqrt{\frac{\iota}{n_h^k(s_h^k, a_h^k)}}.$$

By combining the upper bound of $I_1^{h,k}$ in Equation (59), along with those of $I_{2,1}^{h,k}$ and $I_{2,2}^{h,k}$, we have:

$$\sqrt{\frac{\nu_h^{\mathrm{ref},k} \iota}{n_h^k}} \lesssim \frac{\sqrt{H\Gamma_h^k(s_h^k, a_h^k)\iota}}{n_h^k(s_h^k, a_h^k)} + \sqrt{\frac{(\mathbb{Q}^\star + \beta^2)\iota}{n_h^k(s_h^k, a_h^k)}} + \frac{H\iota^{\frac{3}{4}}}{n_h^k(s_h^k, a_h^k)^{\frac{3}{4}}}. \tag{60}$$

Using the first inequality of inequality (80) in Zhang et al. (2020), we have:

$$\check{\nu}_h^k \lesssim \frac{1}{\check{n}_h^k} \sum_{i=1}^{\check{n}_h^k} \left( \left( V_{h+1}^{\mathrm{ref},\check{l}_i}(s_{h+1}^{\check{l}_i}) - \hat{V}_{h+1}^{\mathrm{ref},\check{l}_i}(s_{h+1}^{\check{l}_i}) \right)^2 + \left( \hat{V}_{h+1}^{\mathrm{ref},\check{l}_i}(s_{h+1}^{\check{l}_i}) - V_{h+1}^*(s_{h+1}^{\check{l}_i}) \right)^2 \right)$$

$$\lesssim \beta^2 + \frac{1}{\check{n}_h^k} \sum_{i=1}^{\check{n}_h^k} \left( \hat{V}_{h+1}^{\mathrm{ref},\check{l}_i}(s_{h+1}^{\check{l}_i}) - V_{h+1}^*(s_{h+1}^{\check{l}_i}) \right)^2$$

and thus

$$\sqrt{\frac{\check{\nu}_h^k \iota}{\check{n}_h^k}} \lesssim \sqrt{\frac{\beta^2 \iota}{\check{n}_h^k}} + \frac{\sqrt{\sum_{i=1}^{\check{n}_h^k} \left( V_{h+1}^{\mathrm{ref},\check{l}_i}(s_{h+1}^{\check{l}_i}) - \hat{V}_{h+1}^{\mathrm{ref},\check{l}_i}(s_{h+1}^{\check{l}_i}) \right)^2 \iota}}{\check{n}_h^k} \lesssim \sqrt{\frac{\beta^2 H \iota}{n_h^k}} + \frac{\sqrt{H\check{\Gamma}_h^k(s_h^k, a_h^k)\iota}}{\check{n}_h^k}. \tag{61}$$

where $\check{\Gamma}_h^k(s_h^k, a_h^k) = \sum_{i=1}^{\check{n}_h^k} \left( V_{h+1}^{\mathrm{ref},\check{l}_i}(s_{h+1}^{\check{l}_i}) - \hat{V}_{h+1}^{\mathrm{ref},\check{l}_i}(s_{h+1}^{\check{l}_i}) \right)$. The last inequality is by Lemma D.3 and $0 \le V_{h+1}^{\mathrm{ref},\check{l}_i}(s_{h+1}^{\check{l}_i}) - \hat{V}_{h+1}^{\mathrm{ref},\check{l}_i}(s_{h+1}^{\check{l}_i}) \le H$.

Applying Equation (60) and Equation (61) to Equation (58), we have:

$$b_h^k(s_h^k, a_h^k) \lesssim \frac{\sqrt{H\Gamma_h^k(s_h^k, a_h^k)\iota}}{n_h^k(s_h^k, a_h^k)} + \sqrt{\frac{(\mathbb{Q}^\star + \beta^2 H)\iota}{n_h^k(s_h^k, a_h^k)}} + \frac{H\iota^{\frac{3}{4}}}{\check{n}_h^k(s_h^k, a_h^k)^{\frac{3}{4}}} + \frac{\sqrt{H\check{\Gamma}_h^k(s_h^k, a_h^k)\iota} + H\iota}{\check{n}_h^k(s_h^k, a_h^k)}. \tag{62}$$

## D.5 REARRANGE THE WEIGHTED SUM OF $G_3$

Similar to Equation (41), it holds that:

$$
\sum_{k=1}^{K} \omega_{h,k} \mathbb{I}\left[\check{n}_h^k \neq 0\right] G_3 = \sum_{k=1}^{K} \omega_{h,k} \mathbb{I}\left[\check{n}_h^k \neq 0\right] \frac{\sum_{i=1}^{\check{n}_h^k}\left(V_{h+1}^{\check{l}_i}(s_{h+1}^{\check{l}_i}) - V_{h+1}^{\star}(s_{h+1}^{\check{l}_i})\right)}{\check{n}_h^k(s_h^k, a_h^k)}
$$

$$
= \sum_{j=1}^{K}\left(\sum_{k=1}^{K} \omega_{h,k} \frac{\sum_{i=1}^{\check{n}_h^k} \mathbb{I}\left[\check{l}_i = j, \check{n}_h^k \neq 0\right]}{\check{n}_h^k(s_h^k, a_h^k)}\right)\left(V_{h+1}^j(s_{h+1}^j) - V_{h+1}^{\star}(s_{h+1}^j)\right)
$$

$$
\leq \sum_{j=1}^{K}\left(\sum_{k=1}^{K} \omega_{h,k} \frac{\sum_{i=1}^{\check{n}_h^k} \mathbb{I}\left[\check{l}_i = j, \check{n}_h^k \neq 0\right]}{\check{n}_h^k(s_h^k, a_h^k)}\right)\left(Q_{h+1}^j - Q_{h+1}^{\star}\right)(s_{h+1}^j, a_{h+1}^j)
$$

(63)

$$
= \sum_{j=1}^{K} \omega_{h+1,j}(h)\left(Q_{h+1}^j(s_{h+1}^j, a_{h+1}^j) - Q_{h+1}^{\star}(s_{h+1}^j, a_{h+1}^j)\right).
$$

(64)

## D.6 BOUNDING THE TERM $\sum_{h'=1}^{H} \sum_{k=1}^{K} Y_{h'}^k$

$$
\sum_{k=1}^{K} Y_{h'}^k = \sum_{k=1}^{K} \mathbb{I}\left[n_{h'}^k = 0\right] H
$$

$$
+ \sum_{k=1}^{K} \frac{\mathbb{I}\left[n_{h'}^k \neq 0\right]}{n_{h'}^k(s_{h'}^k, a_{h'}^k)}\left(\sum_{i=1}^{n_{h'}^k} H\left(\mathbb{1}_{s_{h'+1}^{l_i}} + \mathbb{P}_{s_{h'}^k, a_{h'}^k, h'}\right)\lambda_{h'+1}^{l_i} + \sqrt{H\Gamma_{h'}^k(s_{h'}^k, a_{h'}^k)\iota}\right)
$$

$$
+ \sum_{k=1}^{K} \frac{\mathbb{I}\left[\check{n}_{h'}^k \neq 0\right]}{\check{n}_{h'}^k(s_{h'}^k, a_{h'}^k)}\left(\sum_{i=1}^{\check{n}_{h'}^k} H\left(\mathbb{P}_{s_{h'}^k, a_{h'}^k, h'} + \mathbb{1}_{s_{h'+1}^{\check{l}_i}}\right)\lambda_{h'+1}^{\check{l}_i} + \sqrt{H\check{\Gamma}_{h'}^k(s_{h'}^k, a_{h'}^k)\iota} + H\iota\right).
$$

(65)

In this equation,

$$
\sum_{h'=1}^{H} \sum_{k=1}^{K} \mathbb{I}\left[n_{h'}^k(s_{h'}^k, a_{h'}^k) = 0\right] H = \sum_{h'=1}^{H} \sum_{s,a} H \sum_{k=1}^{K} \mathbb{I}\left[n_{h'}^k(s, a) = 0, (s_{h'}^k, a_{h'}^k) = (s, a)\right] \leq H^3 SA.
$$

(66)

By Lemma D.4, we have the following inequalities:

$$
\sum_{h'=1}^{H} \sum_{k=1}^{K} \frac{\mathbb{I}\left[\check{n}_{h'}^k \neq 0\right]}{\check{n}_{h'}^k(s_{h'}^k, a_{h'}^k)} H\iota \lesssim H^3 SA\iota \log(T).
$$

(67)

According to Lemma D.5, we have:

$$
\sum_{k=1}^{K} \frac{\mathbb{I}\left[n_{h'}^k \neq 0\right]}{n_{h'}^k(s_{h'}^k, a_{h'}^k)} \sum_{i=1}^{n_{h'}^k} H\left(\mathbb{1}_{s_{h'+1}^{l_i}} + \mathbb{P}_{s_{h'}^k, a_{h'}^k, h'}\right)\lambda_{h'+1}^{l_i} \leq 3H \log T \sum_{k=1}^{K}\left(\mathbb{1}_{s_{h'+1}^k} + \mathbb{P}_{s_{h'}^k, a_{h'}^k, h'}\right)\lambda_{h'+1}^k.
$$

(68)

and

$$
\sum_{k=1}^{K} \frac{\mathbb{I}\left[\check{n}_{h'}^k \neq 0\right]}{\check{n}_{h'}^k(s_{h'}^k, a_{h'}^k)} \sum_{i=1}^{\check{n}_{h'}^k} H\left(\mathbb{P}_{s_{h'}^k, a_{h'}^k, h'} + \mathbb{1}_{s_{h'+1}^{\check{l}_i}}\right)\lambda_{h'+1}^{\check{l}_i} \leq 2H \sum_{k=1}^{K}\left(\mathbb{P}_{s_{h'}^k, a_{h'}^k, h'} + \mathbb{1}_{s_{h'+1}^k}\right)\lambda_{h'+1}^k.
$$

(69)

Let $k_0(s) = \max\{k \mid k \le K, N_{h'+1}^k(s) < N_0\}$. When there is no ambiguity, we use $k_0$ for short. Note that

$$\sum_{k=1}^K \lambda_{h'+1}^k(s_{h'+1}^k) = \sum_{k=1}^K \mathbb{I}\left[N_{h'+1}^k(s_{h'+1}^k) < N_0\right] = \sum_s \sum_{k=1}^K \mathbb{I}\left[N_{h'+1}^k(s) < N_0, s_{h'+1}^k = s\right]$$

$$= \sum_{s,a} \sum_{k=1}^{k_0} \mathbb{I}\left[s_{h'+1}^k = s, a_{h'+1}^k = a\right]. \quad (70)$$

Let $(k_0, h'+1)$ be in the $j(s,a,h)$-th state of $(s,a,h)$. Because $N_0 > H$, we know $j(s,a,h) > 1$. Then we have $N_{h'+1}^{k_0}(s,a) = \sum_{i=1}^{j-1} e_i$ and thus

$$\sum_{k=1}^{k_0} \mathbb{I}\left[s_{h'+1}^k = s, a_{h'+1}^k = a\right] \le N_{h'+1}^{k_0}(s,a) + e_j \le N_{h'+1}^{k_0}(s,a) + \left(1 + \frac{1}{H}\right)e_{j-1} \le 3N_{h'+1}^{k_0}(s,a). \quad (71)$$

Applying this inequality to Equation (70), we have

$$\sum_{k=1}^K \lambda_{h'+1}^k(s_{h'+1}^k) \le \sum_{s,a} 3N_{h'+1}^{k_0}(s,a) = 3\sum_s N_{h'+1}^{k_0}(s) \le 3SN_0. \quad (72)$$

Under the event $\mathcal{E}_3$ in Lemma D.1, it also holds that:

$$\sum_{k=1}^K \mathbb{P}_{s_{h'}^k, a_{h'}^k, h'} \lambda_{h'+1}^k \le 3\sum_{k=1}^K \lambda_{h'+1}^k + \iota \le 10SN_0. \quad (73)$$

Applying Equation (72) and Equation (73) to Equation (68) and Equation (69) respectively, then the following two inequalities holds:

$$\sum_{h'=1}^H \sum_{k=1}^K \frac{\mathbb{I}\left[n_{h'}^k \ne 0\right]}{n_{h'}^k(s_{h'}^k, a_{h'}^k)} \sum_{i=1}^{n_{h'}^k} H\left(\mathbb{1}_{s_{h'+1}^{l_i}} + \mathbb{P}_{s_{h'}^k, a_{h'}^k, h'}\right) \lambda_{h'+1}^{l_i} \lesssim H^2 SN_0 \log(T), \quad (74)$$

and

$$\sum_{h'=1}^H \sum_{k=1}^K \frac{\mathbb{I}\left[\check{n}_{h'}^k \ne 0\right]}{\check{n}_{h'}^k(s_{h'}^k, a_{h'}^k)} \sum_{i=1}^{\check{n}_{h'}^k} H\left(\mathbb{P}_{s_{h'}^k, a_{h'}^k, h'} + \mathbb{1}_{s_{h'+1}^{\check{l}_i}}\right) \lambda_{h'+1}^{\check{l}_i} \lesssim H^2 SN_0. \quad (75)$$

Meanwhile, according to Lemma D.2, we have:

$$\Gamma_{h'}^k(s_{h'}^k, a_{h'}^k) = \sum_{i=1}^{n_{h'}^k} \left(V_{h'+1}^{\text{ref}, l_i}(s_{h'+1}^{l_i}) - \hat{V}_{h'+1}^{\text{ref}, l_i}(s_{h'+1}^{l_i})\right) \le H\sum_{i=1}^{n_{h'}^k} \lambda_{h'+1}^{l_i}(s_{h'+1}^{l_i}) \triangleq \Theta_{h'}^k(s_{h'}^k, a_{h'}^k).$$

Then it holds that:

$$\sum_{k=1}^K \frac{\sqrt{\Gamma_{h'}^k(s_{h'}^k, a_{h'}^k)}}{n_{h'}^k(s_{h'}^k, a_{h'}^k)} \le \sum_{k=1}^K \frac{\sqrt{\Theta_{h'}^k(s_{h'}^k, a_{h'}^k)}}{n_{h'}^k(s_{h'}^k, a_{h'}^k)}$$

$$\le \sum_{s,a} \left(\sum_{j \in C} \frac{e_j}{\sum_{i=1}^{j-1} e_i}\right) \sqrt{\Theta_{h'}^K(s,a)\mathbb{I}\left[(s_{h'}^k, a_{h'}^k) = (s,a)\right]}$$

$$\lesssim \log T \sum_{s,a} \sqrt{\Theta_{h'}^K(s,a)\mathbb{I}\left[(s_{h'}^k, a_{h'}^k) = (s,a)\right]}$$

$$\le \log T \sqrt{SA \sum_{s,a} \Theta_{h'}^K(s,a)\mathbb{I}\left[(s_{h'}^k, a_{h'}^k) = (s,a)\right]}. \quad (76)$$

Here, $C = \{j : H \le \sum_{i=1}^{j-1} e_i \le T\}$. The second inequality is by Equation (40) and the mononicity of $\Theta_{h'}^n(s,a)$. The last inequality is by Cauchy's inequality. To continue, note that:

$$\sqrt{\sum_{s,a} \Theta_{h'}^K(s,a)\mathbb{I}\left[(s_{h'}^k, a_{h'}^k) = (s,a)\right]} \le \sqrt{H\sum_{k=1}^K \lambda_{h'+1}^k(s_{h'+1}^k)} \lesssim \sqrt{HSN_0}.$$

The last inequality is by Equation (71). Together with Equation (76), it holds:

$$\sum_{h'=1}^{H}\sum_{k=1}^{K}\frac{\sqrt{H\Gamma_{h'}^{k}(s_{h'}^{k},a_{h'}^{k})\iota}}{n_{h'}^{k}(s_{h'}^{k},a_{h'}^{k})} \lesssim H^2 S \log(T) \sqrt{AN_0\iota}. \tag{77}$$

Since $\check{\Gamma}_{h'}^{k}(s_{h'}^{k},a_{h'}^{k}) \leq \Gamma_{h'}^{k}(s_{h'}^{k},a_{h'}^{k})$ and $4H\check{n}_{h'}^{k}(s_{h'}^{k},a_{h'}^{k}) \geq n_{h'}^{k}(s_{h'}^{k},a_{h'}^{k})$ by Lemma D.3, it holds:

$$\sum_{h'=1}^{H}\sum_{k=1}^{K}\frac{\sqrt{H\check{\Gamma}_{h'}^{k}(s_{h'}^{k},a_{h'}^{k})\iota}}{\check{n}_{h'}^{k}(s_{h'}^{k},a_{h'}^{k})} \lesssim H^3 S \log(T) \sqrt{AN_0\iota}. \tag{78}$$

Applying the inequalities Equation (66), Equation (67), Equation (74), Equation (75), Equation (77) and Equation (78) to Equation (65), since $N_0 = O(\frac{SAH^5\iota}{\beta^2})$, we have:

$$\sum_{h'=1}^{H}\sum_{k=1}^{K}Y_{h'}^{k} \leq O\left(\frac{H^7 S^2 A\iota \log(T)}{\beta^2}\right).$$

## E    PROOF OF THEOREM 3.3

*Proof.* For $\delta \in (0,1)$, let $p \leftarrow \frac{\delta}{40SAH^2T^4}$, then $\iota = \log(\frac{2}{p}) = O(\frac{SAT}{\delta})$. Now with probability at least $1-\delta$, $\bigcap_{i=1}^{8}\mathcal{E}_i$ holds. Next, we will prove Theorem 3.3 under the event $\bigcap_{i=1}^{8}\mathcal{E}_i$.

From the proof of Theorem 2 in Zhang et al. (2020), we have:

$$N_{\text{switch}} \leq \sum_{s,a,h} 4H \log\left(\frac{N_h^{K+1}(s,a)}{2H} + 1\right).$$

Next for any $(s,a,h) \in \mathcal{S} \times \mathcal{A} \times [H]$, we will bound the term $N_h^{K+1}(s,a)$. Let $\mathcal{A}_h^*(s) = \{a \mid a = \arg\max_{a'} Q_h^*(s,a')\}$, which is the set of optimal actions for state-step pair $(s,h)$. For $a \notin \mathcal{A}_h^\star(s)$, we have $\Delta_h(s,a) > 0$ and then $\Delta_h(s,a) \geq \Delta_{\min}$. For any $h \in [H]$, let set $D_h$ be all triples of $(s,a,h)$ such that $a \notin \mathcal{A}_h^\star(s)$, i.e., $D_h = \{(s,a,h)|a \notin \mathcal{A}_h^\star(s)\}$.

We also let the set $D = \bigcup_{h=1}^{H} D_h$ and the set $D_{\text{opt}} = \{(s,a,h)|a \in \mathcal{A}_h^\star(s)\}$. Then we have $|D| + |D_{\text{opt}}| = SAH$. Since for every state-step pair $(s,h)$, there exists at least one optimal action. Therefore we know $|D_{\text{opt}}| \geq SH$ and then $0 \leq |D| \leq SA(H-1)$.

If for given $(h,k) \in [H] \times [k]$, $(s_h^k,a_h^k,h) \in D_h$, we have $\Delta_h(s_h^k,a_h^k) \geq \Delta_{\min}$. Then it holds that:

$$Q_h^k(s_h^k,a_h^k) - Q_h^\star(s_h^k,a_h^k) = V_h^k(s_h^k) - Q_h^\star(s_h^k,a_h^k) \geq \Delta_h(s_h^k,a_h^k) \geq \Delta_{\min}.$$

The first inequality is because $V_h^k(s) \geq V_h^\star(s)$. Therefore, we have

$$\sum_{(s,a,h) \in D_h}\mathbb{I}[(s_h^k,a_h^k) = (s,a)] = \mathbb{I}[(s_h^k,a_h^k,h) \in D_h]$$

$$\leq \mathbb{I}\left[Q_h^k(s_h^k,a_h^k) - Q_h^\star(s_h^k,a_h^k) \geq \Delta_{\min}\right] = \sum_{n=1}^{N}\omega_{h,k}^{(n)}.$$

and then

$$\sum_{(s,a,h) \in D} N_h^{K+1}(s,a) = \sum_{h=1}^{H}\sum_{(s,a,h) \in D_h} N_h^{K+1}(s,a) = \sum_{h=1}^{H}\sum_{(s,a,h) \in D_h}\sum_{k=1}^{K}\mathbb{I}[(s_h^k,a_h^k) = (s,a)]$$

$$\leq \sum_{h=1}^{H}\sum_{k=1}^{K}\sum_{n=1}^{N}\omega_{h,k}^{(n)} = \sum_{h=1}^{H}\sum_{n=1}^{N}\|\omega^{(n)}\|_{1,h}.$$

By Equation (51), we know:

$$\sum_{(s,a,h) \in D_{\text{opt}}^c} N_h^{K+1}(s,a) \leq \sum_{h=1}^{H}\sum_{n=1}^{N}\|\omega^{(n)}\|_{1,h} \leq O\left(\frac{(\mathbb{Q}^\star + \beta^2 H) SAH^3\iota}{\Delta_{\min}^2} + \frac{H^8 S^2 A\iota \log(T)}{\beta^2 \Delta_{\min}}\right). \tag{79}$$

Therefore we have:

$$
N_{\text{switch}} \leq \sum_{s,a,h} 4H \log \left( \frac{N_h^{K+1}(s,a)}{2H} + 1 \right)
$$

$$
= \sum_{(s,a,h) \in D_{\text{opt}}^c} 4H \log \left( \frac{N_h^{K+1}(s,a)}{2H} + 1 \right) + \sum_{(s,a,h) \notin D_{\text{opt}}} 4H \log \left( \frac{N_h^{K+1}(s,a)}{2H} + 1 \right) \qquad (80)
$$

$$
\leq 4H(SAH - |D_{\text{opt}}|) \log \left( 1 + \frac{\sum_{h=1}^H \sum_{n=1}^N \|\omega^{(n)}\|_{1,h}}{2H(SAH - |D_{\text{opt}}|)} \right) + 4H|D_{\text{opt}}| \log \left( \frac{T}{2H|D_{\text{opt}}|} + 1 \right)
$$

$$
\leq O \left( H(SAH - |D_{\text{opt}}|) \log \left( \frac{(\mathbb{Q}^\star + \beta^2 H)H^2 SA\iota}{(SAH - |D_{\text{opt}}|)\Delta_{\min}^2} + \frac{H^7 S^2 A\iota \log(T)}{\beta^2 (SAH - |D_{\text{opt}}|)\Delta_{\min}} \right) \right.
$$

$$
\left. + H|D_{\text{opt}}| \log \left( \frac{K}{|D_{\text{opt}}|} + 1 \right) \right). \qquad (81)
$$

The first inequality is because of Jensen's Inequality. The last inequality is by Equation (51). Since $\mathbb{Q}^\star \leq H^2$ and $\beta \leq H$, then we have:

$$
\frac{(\mathbb{Q}^\star + \beta^2 H)H^2 SA\iota}{(SAH - |D_{\text{opt}}|)\Delta_{\min}^2} \leq \frac{H^7 SA\iota}{\beta^2 (SAH - |D_{\text{opt}}|)\Delta_{\min}^2}.
$$

By $\Delta_{\min} \leq H$, we also have:

$$
\frac{H^7 S^2 A\iota \log(T)}{\beta^2 (SAH - |D_{\text{opt}}|)\Delta_{\min}} \leq \frac{H^8 S^2 A\iota \log(T)}{\beta^2 (SAH - |D_{\text{opt}}|)\Delta_{\min}^2}.
$$

For $\delta \in (0,1)$, let $p \leftarrow \frac{\delta}{60SAH^2 T^5}$, then $\iota = \log(\frac{2}{p}) \leq O(\log(\frac{SAT}{\delta}))$. Applying the above two inequalities to Equation (81), with probability at least $1 - \delta$, we have it holds that:

$$
N_{\text{switch}} \leq O \left( H(SAH - |D_{\text{opt}}|) \log \left( \frac{H^8 S^2 A\iota \log(T)}{\beta^2 (SAH - |D_{\text{opt}}|)\Delta_{\min}^2} \right) + H|D_{\text{opt}}| \log \left( \frac{K}{|D_{\text{opt}}|} + 1 \right) \right)
$$

$$
= O \left( H(SAH - |D_{\text{opt}}|) \log \left( \frac{H^4 SA^{\frac{1}{2}}\iota}{\beta \sqrt{(SAH - |D_{\text{opt}}|)\Delta_{\min}}} \right) + H|D_{\text{opt}}| \log \left( \frac{K}{|D_{\text{opt}}|} + 1 \right) \right)
$$

$$
= O \left( H|D_{\text{opt}}^c| \log \left( \frac{H^4 SA^{\frac{1}{2}} \log(\frac{SAT}{\delta})}{\beta \sqrt{|D_{\text{opt}}^c|\Delta_{\min}}} \right) + H|D_{\text{opt}}| \log \left( \frac{K}{|D_{\text{opt}}|} + 1 \right) \right).
$$

Especially, if the optimal policy is deterministic and unique, which means $|D_{\text{opt}}| = SH$, then the policy switching cost is upper bounded by:

$$
O \left( H^2 SA \log \left( \frac{H^{\frac{7}{2}} S^{\frac{1}{2}} \log(\frac{SAT}{\delta})}{\beta \Delta_{\min}} \right) + H^2 S \log \left( \frac{K}{HS} + 1 \right) \right).
$$

$\square$

# F  PROOF OF THEOREM 3.2

## F.1  ALGORITHM DETAILS

Before continuing, we briefly introduce the refined Q-EarlySettled-Advantage algorithm, which is similar to the original version in Li et al. (2021). We will first discuss the key auxiliary functions used for estimating the $Q$-value functions. For any $\delta \in [0,1]$, let $\iota = \log(\frac{SAT}{\delta})$.

The algorithm updates $\mu_h^{\text{ref}}$ and $\sigma_h^{\text{ref}}$ to represent the current mean and second moment of the reference function. $\mu_h^{\text{adv}}$ and $\sigma_h^{\text{adv}}$ denotes the current weighted mean and weighted second moment of

the reference function with weight to be the learning rate $\eta_n = \frac{H+1}{H+n}$. $b_h^R$ is the exploration bonus. Then we present the details of Q-EarlySettled-Advantage algorithm below.

---

**Algorithm 2** Auxiliary functions

---

1: **function** UPDATE-UCB-Q

2: $\qquad Q_h^{\text{UCB}}(s_h, a_h) \leftarrow (1 - \eta_n) Q_h^{\text{UCB}}(s_h, a_h) + \eta_n \left( r_h(s_h, a_h) + V_{h+1}(s_{h+1}) + c_b \sqrt{\frac{H^3 \iota}{n}} \right).$

3: **function** UPDATE-LCB-Q

4: $\qquad Q_h^{\text{LCB}}(s_h, a_h) \leftarrow (1 - \eta_n) Q_h^{\text{LCB}}(s_h, a_h) + \eta_n \left( r_h(s_h, a_h) + V_{h+1}^{\text{LCB}}(s_{h+1}) - c_b \sqrt{\frac{H^3 \iota}{n}} \right).$

5: **function** UPDATE-UCB-ADVANTAGE

6: $\qquad [\mu_h^{\text{ref}}, \sigma_h^{\text{ref}}, \mu_h^{\text{adv}}, \sigma_h^{\text{adv}}](s_h, a_h) \leftarrow$ UPDATE-MOMENTS;

7: $\qquad [\delta_h^R, B_h^R](s_h, a_h) \leftarrow$ UPDATE-BONUS;

8: $\qquad b_h^R \leftarrow B_h^R(s_h, a_h) + (1 - \eta_n)\frac{\delta_h^R(s_h, a_h)}{\eta_n} + c_b \frac{H^2 \iota}{n^{3/4}};$

9: $\qquad Q_h^R(s_h, a_h) \leftarrow (1 - \eta_n) Q_h^R(s_h, a_h)$
$\qquad\qquad\qquad\qquad + \eta_n \left( r_h(s_h, a_h) + V_{h+1}(s_{h+1}) - V_{h+1}^R(s_{h+1}) + \mu_h^{\text{ref}}(s_h, a_h) + b_h^R \right).$

10: **function** UPDATE-MOMENTS

11: $\qquad \mu_h^{\text{ref}}(s_h, a_h) \leftarrow \left(1 - \frac{1}{n}\right) \mu_h^{\text{ref}}(s_h, a_h) + \frac{1}{n} V_{h+1}^R(s_{h+1});$

12: $\qquad \sigma_h^{\text{ref}}(s_h, a_h) \leftarrow \left(1 - \frac{1}{n}\right) \sigma_h^{\text{ref}}(s_h, a_h) + \frac{1}{n} \left(V_{h+1}^R(s_{h+1})\right)^2;$

13: $\qquad \mu_h^{\text{adv}}(s_h, a_h) \leftarrow (1 - \eta_n) \mu_h^{\text{adv}}(s_h, a_h) + \eta_n \left(V_{h+1}(s_{h+1}) - V_{h+1}^R(s_{h+1})\right);$

14: $\qquad \sigma_h^{\text{adv}}(s_h, a_h) \leftarrow (1 - \eta_n) \sigma_h^{\text{adv}}(s_h, a_h) + \eta_n \left(V_{h+1}(s_{h+1}) - V_{h+1}^R(s_{h+1})\right)^2;$

15: **function** UPDATE-BONUS

16: $\qquad B_h^{\text{next}}(s_h, a_h) \leftarrow$
$\qquad c_b \sqrt{\frac{\iota}{n}} \left( \sqrt{\sigma_h^{\text{ref}}(s_h, a_h) - \left(\mu_h^{\text{ref}}(s_h, a_h)\right)^2} + \sqrt{H} \sqrt{\sigma_h^{\text{adv}}(s_h, a_h) - \left(\mu_h^{\text{adv}}(s_h, a_h)\right)^2} \right);$

17: $\qquad \delta_h^R(s_h, a_h) = B_h^{\text{next}}(s_h, a_h) - B_h^R(s_h, a_h);$

18: $\qquad B_h^R(s_h, a_h) \leftarrow B_h^{\text{next}}(s_h, a_h).$

---

**Algorithm 3** Refined Q-EarlySettled-Advantage

---

1: **Parameters:** Some universal constant $c_b > 0$ and probability of failure $\delta \in (0, 1)$;

2: **Initialize** $Q_h^1(s, a), Q_h^{\text{UCB},1}(s, a), Q_h^{R,1}(s, a) \leftarrow H$, $Q_h^{\text{LCB},1}(s, a) \leftarrow 0$; $V_h^{R,1}(s), V_h^1(s) \leftarrow H$;
$\quad N_h^1(s, a), \mu_h^{\text{ref}}(s, a), \sigma_h^{\text{ref}}(s, a), \mu_h^{\text{adv}}(s, a), \sigma_h^{\text{adv}}(s, a), \delta_h^R(s, a), B_h^R(s, a) \leftarrow 0$;
$\quad$ and $u_h^1(s) \leftarrow$ True, for all $(s, a, h) \in \mathcal{S} \times \mathcal{A} \times [H]$.

3: **for** Episode $k = 1$ to $K$ **do**

4: $\qquad$ Set initial state $s_1^k \leftarrow s_1^k$;

5: $\qquad$ **for** Step $h = 1$ to $H$ **do**

6: $\qquad\qquad$ Take action $a_h^k = \pi_h^k(s_h^k) = \arg\max_a Q_h^k(s_h^k, a)$, and draw $s_{h+1}^k \sim P_h(\cdot|s_h^k, a_h^k)$;

7: $\qquad\qquad N_h^k(s_h^k, a_h^k) \leftarrow N_h^{k-1}(s_h^k, a_h^k) + 1$; $n \leftarrow N_h^k(s_h^k, a_h^k)$;

8: $\qquad\qquad Q_h^{\text{UCB},k+1}(s_h^k, a_h^k) \leftarrow$ UPDATE-UCB-Q.

9: $\qquad\qquad Q_h^{\text{LCB},k+1}(s_h^k, a_h^k) \leftarrow$ UPDATE-LCB-Q.

10: $\qquad\qquad Q_h^{R,k+1}(s_h^k, a_h^k) \leftarrow$ UPDATE-UCB-ADVANTAGE.

11: $\qquad\qquad Q_h^{k+1}(s_h^k, a_h^k) \leftarrow \min\{Q_h^{R,k+1}(s_h^k, a_h^k), Q_h^{\text{UCB},k+1}(s_h^k, a_h^k), Q_h^k(s_h^k, a_h^k)\};$

12: $\qquad\qquad V_h^{k+1}(s_h^k) \leftarrow \max_a Q_h^{k+1}(s_h^k, a);$

13: $\qquad\qquad V_h^{\text{LCB},k+1}(s_h^k) \leftarrow \max\left\{ \max_a Q_h^{\text{LCB},k+1}(s_h^k, a), V_h^{\text{LCB},k}(s_h^k) \right\};$

14: $\qquad\qquad$ **if** $V_h^{k+1}(s_h^k) - V_h^{\text{LCB},k+1}(s_h^k) > \beta$ **then**

15: $\qquad\qquad\qquad V_h^{R,k+1}(s_h^k) \leftarrow V_h^{k+1}(s_h^k);$

16: $\qquad\qquad$ **else if** $u_h^k(s_h^k) =$ True **then**

17: $\qquad\qquad\qquad V_h^{R,k+1}(s_h^k) \leftarrow V_h^{k+1}(s_h^k)$; $u_h^{k+1}(s_h^k) =$ False.

---

At the beginning of the $k$-th episode, we can obtain $V$-estimate $V_h^k(s)$, the reference function $V_h^{\text{R},k}(s)$ and the policy $\pi^k$ from the previous episode $k-1$ and select a initial state $s_1^k$ (For the first episode, we randomly choose a policy $\pi^1$ and $V_h^1(s) = V_h^{\text{R},1} = H$). At step $h \in [H]$, we can process the trajectory with $a_h^k = \pi_h^k(s_h^k)$ and $s_{h+1}^k \sim \mathbb{P}_h(\cdot|s_h^k, a_h^k)$. Now we need to update the estimates of both $Q$-value and $V$-value functions at the end of $k$-th episode. In the algorithm, the estimate learned from the UCB by the end of $k$-th episode, denoted as $Q_h^{\text{UCB},k+1}$, is updated to:

$$Q_h^{\text{UCB},k+1} = r_h^k(s_h^k, a_h^k) + \sum_{n=1}^{N_h^{k+1}} \eta_n^{N_h^{k+1}} \left( V_{h+1}^{k^n}(s_{h+1}^{k^n}) + c_b\sqrt{\frac{H^3\iota}{n}} \right) \tag{82}$$

Here we define $N_h^k = N_h^k(s_h^k, a_h^k)$ as the number of times that the state-action pair $(s_h^k, a_h^k)$ has been visited at step $h$ at the beginning of the $k$-th episode and $k^n = k_h^n(s_h^k, a_h^k)$ denotes the index of the episode in which the state-action pair $(s_h^k, a_h^k)$ is visited for the $n$-th time at step $h$. The term $c_b\sqrt{\frac{H^3\iota}{n}}$ represents the exploration bonus for $n$-th visit, where $c_b > 0$ is a sufficiently large constant.

Another $Q$-estimate obtained from LCB at the end of $k$-th episode, denoted as $Q_h^{\text{LCB},k+1}$, is updated similarly to $Q_h^{\text{UCB},k+1}$, but with the exploration bonus subtracted instead.

The last estimate of $Q$-value function, denoted as $Q_h^{\text{R},k+1}$, uses reference-advantage decomposition techniques. At the end of $k$-th episode, $Q_h^{\text{R},k+1}$ is updated to:

$$Q_h^{\text{R},k+1} = r_h^k(s_h^k, a_h^k) + \sum_{n=1}^{N_h^{k+1}} \eta_n^{N_h^{k+1}} \left( V_{h+1}^{k^n}(s_{h+1}^{k^n}) - V_{h+1}^{\text{R},k^n}(s_{h+1}^{k^n}) + \frac{\sum_{i=1}^n V_{h+1}^{\text{R},k^i}(s_{h+1}^{k^i})}{n} + b_h^{\text{R},k^n+1} \right). \tag{83}$$

In Equation (83), $V_h^{\text{R},k}(s)$ is the reference function learned at the end of episode $k-1$. The key idea of the reference-advantage decomposition is that we expect to maintain a collection of reference values $\{V_h^{\text{R},k}(s)\}_{s,k,h}$, which form reasonable estimates of $\{V_h^{\star}(s)\}_{s,h}$ and become increasingly more accurate as the algorithm progresses. It means for any $s \in \mathcal{S}$, sufficiently large $k$ and some given $\beta \in (0, H]$, it holds $|V_h^{\text{R},k}(s) - V_h^{\star}(s)| \leq \beta$.

With two additional $Q$-estimates in hand — $Q_h^{\text{UCB},k+1}$ learned from UCB and $Q_h^{\text{R},k+1}$ obtained from the reference-advantage decomposition, we can update $Q_h^{k+1}(s_h^k, a_h^k)$ as follows:

$$Q_h^{k+1}(s_h^k, a_h^k) = \min\{Q_h^{\text{UCB},k+1}(s_h^k, a_h^k), Q_h^{\text{R},k+1}(s_h^k, a_h^k), Q_h^k(s_h^k, a_h^k)\}. \tag{84}$$

We also incorporate $Q_h^k(s_h^k, a_h^k)$ here to keep the monotonicity of the update. Then we can learn $V_h^{k+1}(s_h^k, a_h^k)$ and $V_h^{\text{LCB},k+1}(s_h^k, a_h^k)$ by a greedy policy with respect to these $Q$-estimates:

$$V_h^{k+1}(s_h^k) = \max_a Q_h^{k+1}(s_h^k, a), \quad V_h^{\text{LCB},k+1}(s_h^k) = \max\left\{ \max_a Q_h^{\text{LCB},k+1}(s_h^k, a), V_h^{\text{LCB},k}(s_h^k) \right\}.$$

In the algorithm, $V_h^{\text{LCB},k}(s)$ serves as a lower bound of $V_h^{\star}(s)$. We determine the final value $V_h^{\text{R},K+1}(s)$ of the reference function for the state-step pair $(s, h)$ when the condition $V_h^k(s) - V_h^{\text{LCB},k}(s) \leq \beta$ is met for the first time.

## F.2 AUXILIARY LEMMAS

As can be easily verified, we have

$$\sum_{n=1}^N \eta_n^N = \begin{cases} 1, & \text{if } N > 0, \\ 0, & \text{if } N = 0. \end{cases} \tag{85}$$

**Lemma F.1.** *For any integer $N > 0$, the following properties hold:*

$$\frac{1}{N^a} \leq \sum_{n=1}^N \frac{\eta_n^N}{n^a} \leq \frac{2}{N^a}, \quad \text{for all } \frac{1}{2} \leq a \leq 1, \tag{86}$$

$$\max_{1 \leq n \leq N} \eta_n^N \leq \frac{2H}{N}, \quad \sum_{n=1}^{N} (\eta_n^N)^2 \leq \frac{2H}{N}, \quad \sum_{N=n}^{\infty} \eta_n^N \leq 1 + \frac{1}{H}. \tag{87}$$

*Proof.* It is proved in Appendix B of Li et al. (2021). □

Let $u_i^N = \sum_{n=i}^{N} \frac{\eta_n^N}{n}$. Then according to Equation (86), we know $u_i^N \leq \frac{2}{N}$ for any $i \leq N \in \mathbb{N}_+$.

**Lemma F.2.** *Using $\forall(s,a,h,k)$ as the simplified notation for $\forall(s,a,h,k) \in \mathcal{S} \times \mathcal{A} \times [H] \times [K]$ and $\forall(s,h,k)$ as the simplified notation for $\forall(s,a,h,k) \in \mathcal{S} \times [H] \times [K]$. Then we have the following conclusions:*

*(a) (Lemma 2 of Li et al. (2021)) With probability at least $1 - \delta$, the following event holds:*

$$\mathcal{E}_1 = \left\{ Q_h^\star(s,a) \leq Q_h^{k+1}(s,a) \leq Q_h^k(s,a), \ V_h^\star(s) \leq V_h^k(s) \leq V_h^{\mathrm{R},k}(s), \ \forall(s,a,h,k) \right\}.$$

*(b) (Lemma 3 of Li et al. (2021)) With probability at least $1 - \delta$, the following event holds:*

$$\mathcal{E}_2 = \left\{ Q_h^{\mathrm{LCB},k}(s,a) \leq Q_h^\star(s,a), V_h^{\mathrm{LCB},k}(s) \leq V_h^\star(s), \ \forall(s,a,h,k) \text{ and} \right.$$
$$\left. \sum_{h=1}^{H} \sum_{k=1}^{K} \mathbb{I}\left[ Q_h^k(s_h^k, a_h^k) - Q_h^{\mathrm{LCB},k}(s_h^k, a_h^k) > \varepsilon \right] \lesssim \frac{H^6 S A \iota}{\varepsilon^2}, \text{ for any } \epsilon \in (0, H] \right\}.$$

*(c) (Paraphrased from Lemma 4 of Li et al. (2021)) With probability at least $1 - \delta$, the following event holds:*

$$\mathcal{E}_3 = \left\{ \left| V_h^k(s) - V_h^{\mathrm{R},k}(s) \right| \leq 2\beta \text{ and} \right.$$
$$\left. \sum_{h=1}^{H} \sum_{k=1}^{K} \left( V_h^k - V_h^{\mathrm{LCB},k} \right)(s_h^k) \ \mathbb{I}\left[ \left( V_h^k - V_h^{\mathrm{LCB},k} \right)(s_h^k) > \beta \right] \leq \frac{H^6 S A \iota}{\beta}, \forall(s,h,k) \right\}.$$

*(d) With probability at least $1 - \delta$, the following event holds:*

$$\mathcal{E}_4 = \left\{ \sum_{i=1}^{N_h^k} u_i^{N_h^k} \left( \mathbb{1}_{s_{h+1}^{k^i}} - \mathbb{P}_{s,a,h} \right) \left( \hat{V}_{h+1}^{\mathrm{R},k^i} - V_{h+1}^\star \right) \leq 2\sqrt{\frac{2\beta^2 \iota}{N_h^k(s,a)}}, \ \forall(s,a,h,k) \right\}.$$

*(e) With probability at least $1 - \delta$, the following event holds:*

$$\mathcal{E}_5 = \left\{ \sum_{i=1}^{N_h^k} u_i^{N_h^k} \left( \mathbb{1}_{s_{h+1}^{k^i}} - \mathbb{P}_{s,a,h} \right) V_{h+1}^\star \leq 8\sqrt{\frac{\mathbb{Q}^\star \iota}{N_h^k(s,a)}} + 16\frac{H\iota}{N_h^k(s,a)}, \ \forall(s,a,h,k) \right\}.$$

*(f) With probability at least $1 - \delta$, the following event holds:*

$$\mathcal{E}_6 = \left\{ \sum_{n=1}^{N_h^k} \eta_n^{N_h^k} \left( \mathbb{P}_{s,a,h} - \mathbb{1}_{s_{h+1}^{k^n}} \right) \left( \hat{V}_{h+1}^{\mathrm{R},k^i} - V_{h+1}^\star \right) \leq 2\sqrt{\frac{\beta^2 H\iota}{N_h^k(s,a)}}, \ \forall(s,a,h,k) \right\}.$$

*(g) With probability at least $1 - \delta$, the following event holds:*

$$\mathcal{E}_7 = \left\{ \sum_{h=1}^{H} \sum_{k=1}^{K} \mathbb{P}_{s_h^k, a_h^k, h} \left\{ \left( V_{h+1}^k - V_{h+1}^{\mathrm{LCB},k} \right)(s_{h+1}^k) \mathbb{I}\left[ \left( V_{h+1}^k - V_{h+1}^{\mathrm{LCB},k} \right)(s_{h+1}^k) > \beta \right] \right\} \right.$$
$$\left. \leq 3 \sum_{h=1}^{H} \sum_{k=1}^{K} \left( V_{h+1}^k - V_{h+1}^{\mathrm{LCB},k} \right)(s_{h+1}^k) \mathbb{I}\left[ \left( V_{h+1}^k - V_{h+1}^{\mathrm{LCB},k} \right)(s_{h+1}^k) > \beta \right] + H\iota \right\}.$$

*Proof.* (c) The proof is adapted from the proof of Equation (146) in Li et al. (2021), with the substitution of $\left(V_h^j - V_h^{\text{LCB},k}\right)(s_h^k) > 1$ by $\left(V_h^j - V_h^{\text{LCB},k}\right)(s_h^k) > \beta$.

(d) From the definition of $\hat{V}_h^{\text{R},k}(s)$, we know that for any $k \in [K]$:

$$V_h^\star(s) \le \hat{V}_h^{\text{R},k}(s) \le V_h^\star(s) + \beta. \tag{88}$$

Then the sequence

$$\left\{ \sum_{i=1}^{j} u_i^N \left( \mathbb{1}_{s_{h+1}^{k^i}} - \mathbb{P}_{s,a,h} \right) \left( \hat{V}_{h+1}^{\text{R},k^i} - V_{h+1}^\star \right) \right\}_{j \in \mathbb{N}^+}$$

is a martingale sequence with

$$\left| u_i^N \left( \mathbb{1}_{s_{h+1}^{k^i}} - \mathbb{P}_{s,a,h} \right) \left( \hat{V}_{h+1}^{\text{R},k^i} - V_{h+1}^\star \right) \right| \le \frac{2\beta}{N}.$$

Then according to Azuma-Hoeffding inequality, for any $\delta \in (0,1)$, with probability at least $1 - \frac{\delta}{SAT}$, it holds for given $N_H^k(s,a) = N \in \mathbb{N}_+$ that:

$$\sum_{i=1}^{N} \left( \mathbb{1}_{s_{h+1}^{k^i}} - \mathbb{P}_{s,a,h} \right) \left( \hat{V}_{h+1}^{\text{R},k^i} - V_{h+1}^\star \right) \le 2\sqrt{\frac{2\beta^2 \iota}{N}}.$$

For any all $(s,a,h,k) \in \mathcal{S} \times \mathcal{A} \times [H] \times [K]$, we have $N_h^k(s,a) \in [\frac{T}{H}]$. Considering all the possible combinations $(s,a,h,N) \in \mathcal{S} \times \mathcal{A} \times [H] \times [\frac{T}{H}]$, with probability at least $1 - \delta$, it holds simultaneously for all $(s,a,h,k) \in \mathcal{S} \times \mathcal{A} \times [H] \times [K]$ that:

$$\sum_{i=1}^{N_h^k} u_i^{N_h^k} \left( \mathbb{1}_{s_{h+1}^{k^i}} - \mathbb{P}_{s,a,h} \right) \left( \hat{V}_{h+1}^{\text{R},k^i} - V_{h+1}^\star \right) \le 2\sqrt{\frac{2\beta^2 \iota}{N_h^k(s,a)}}.$$

(e) The sequence

$$\left\{ \sum_{i=1}^{j} u_i^N \left( \mathbb{1}_{s_{h+1}^{k^i}} - \mathbb{P}_{s,a,h} \right) V_{h+1}^\star \right\}_{j \in \mathbb{N}_+}$$

is a martingale sequence with

$$\left| u_i^N \left( \mathbb{1}_{s_{h+1}^{l_i}} - \mathbb{P}_{s,a,h} \right) V_{h+1}^\star \right| \le \frac{2H}{N}.$$

Using Lemma C.3 with $c = \frac{2H}{N}$, $\epsilon = c^2$ and $\delta$ being $\frac{\delta}{SAT^2}$, for any given $N_h^k(s,a) = N \in [T/H]$, with probability at least $1 - 2(\log_2(N) + 1)\frac{\delta}{SAT^2} \ge 1 - \frac{\delta}{SAT}$, we have:

$$\sum_{i=1}^{N} u_i^N \left( \mathbb{1}_{s_{h+1}^{k^i}} - \mathbb{P}_{s,a,h} \right) V_{h+1}^\star \le 8\sqrt{\frac{\mathbb{Q}^\star \iota}{N}} + 16\frac{H\iota}{N}.$$

For any all $(s,a,h,k) \in \mathcal{S} \times \mathcal{A} \times [H] \times [K]$, we have $N_h^k(s,a) \in [\frac{T}{H}]$. Considering all the possible combinations $(s,a,h,N) \in \mathcal{S} \times \mathcal{A} \times [H] \times [\frac{T}{H}]$, with probability at least $1 - \delta$, it holds simultaneously for all $(s,a,h,k) \in \mathcal{S} \times \mathcal{A} \times [H] \times [K]$:

$$\sum_{i=1}^{N_h^k} u_i^{N_h^k} \left( \mathbb{1}_{s_{h+1}^{k^i}} - \mathbb{P}_{s,a,h} \right) V_{h+1}^\star \le 8\sqrt{\frac{\mathbb{Q}^\star \iota}{N_h^k(s,a)}} + 16\frac{H\iota}{N_h^k(s,a)}.$$

(f) The sequence

$$\left\{ \sum_{n=1}^{j} \eta_n^N \left( \mathbb{P}_{s,a,h} - \mathbb{1}_{s_{h+1}^{k^n}} \right) \left( \hat{V}_{h+1}^{\text{R},k^n} - V_{h+1}^\star \right) \right\}_{j \in \mathbb{N}^+}$$

is a martingale sequence with

$$\eta_n^N \left( \mathbb{P}_{s,a,h} - \mathbb{1}_{s_{h+1}^{k^n}} \right) \left( \hat{V}_{h+1}^{\text{R},k^n} - V_{h+1}^\star \right) \le \eta_n^N \beta.$$

Then according to Azuma-Hoeffding inequality and Equation (87), for any $\delta \in (0,1)$, with probability at least $1 - \frac{\delta}{SAT}$, it holds for given $N_h^k(s,a) = N \in \mathbb{N}_+$ that:

$$\sum_{n=1}^N \eta_n^N \left( \mathbb{P}_{s,a,h} - \mathbb{1}_{s_{h+1}^{k^n}} \right) \left( \hat{V}_{h+1}^{\text{R},k^i} - V_{h+1}^\star \right) \le 2\sqrt{\frac{\beta^2 H \iota}{N}}.$$

For any all $(s,a,h,k) \in \mathcal{S} \times \mathcal{A} \times [H] \times [K]$, we have $N_h^k(s,a) \in [\frac{T}{H}]$. Considering all the possible combinations $(s,a,h,N) \in \mathcal{S} \times \mathcal{A} \times [H] \times [\frac{T}{H}]$, with probability at least $1 - \delta$, it holds simultaneously for all $(s,a,h,k) \in \mathcal{S} \times \mathcal{A} \times [H] \times [K]$ that:

$$\sum_{n=1}^{N_h^k} \eta_n^{N_h^k} \left( \mathbb{P}_{s,a,h} - \mathbb{1}_{s_{h+1}^{k^n}} \right) \left( \hat{V}_{h+1}^{\text{R},k^i} - V_{h+1}^\star \right) \le 2\sqrt{\frac{\beta^2 H \iota}{N_h^k(s,a)}}$$

(g) This conclusion is directly proved by Lemma C.2 with $l = H$.

$\square$

**Lemma F.3.** *For any non-negative weight sequence $\{\omega_{h,k}\}_{h,k}$ and $\alpha \in (0,1)$, it holds that:*

$$\sum_{k=1}^K \frac{\omega_{h,k}}{N_h^k(s_h^k, a_h^k)^\alpha} \le \frac{1}{1-\alpha} (SA \|\omega\|_{\infty,h})^\alpha \|\omega\|_{1,h}^{1-\alpha},$$

*Here, $\|\omega\|_{\infty,h} = \max_k \{\omega_{h,k}\}$ and $\|\omega\|_{1,h} = \sum_{k=1}^K \omega_{h,k}$.*
*For $\alpha = 1$, we have the following conclusions:*

$$\sum_{k=1}^K \frac{1}{N_h^k(s_h^k, a_h^k)} \le SA \log(T),$$

*Proof.*

$$\sum_{k=1}^K \frac{\omega_{h,k}}{N_h^k(s_h^k, a_h^k)^\alpha} = \sum_{s,a} \sum_{i=1}^{N_h^K(s,a)} \frac{\omega_{h,k^i(s,a)}}{i^\alpha} \tag{89}$$

Here $k^i(s,a)$ is the episode index of the $i$-th visits to $(s,a,h)$. Let $c_h(s,a) = \sum_{i=1}^{N_h^K(s,a)} \omega_{h,k^i(s,a)}$ and then we have $\sum_{s,a} c_h(s,a) = \sum_{k=1}^K \omega_{h,k} = \|\omega\|_{1,h}$. Given the term $\sum_{k=1}^K \frac{\omega_{h,k^i(s,a)}}{i^\alpha}$, when the weights $\omega_{h,k^i(s,a)}$ concentrates on former terms, we can obtain the largest value. Let

$$k_{s,a,h} = \left\lceil \frac{c_h(s,a)}{\|\omega\|_{\infty,h}} \right\rceil \quad \text{and} \quad d_{s,a,h} = c_h(s,a) - (k_{s,a,h} - 1)\|\omega\|_{\infty,h}.$$

Then we have:

$$\sum_{k=1}^K \frac{\omega_{h,k}}{N_h^k(s_h^k, a_h^k)^\alpha} \le \sum_{s,a} \sum_{i=1}^{k_{s,a,h}-1} \frac{\|\omega\|_{\infty,h}}{i^\alpha} + \frac{d_{s,a,h}}{k_{s,a,h}^\alpha}$$

$$\le \sum_{s,a} \|\omega\|_{\infty,h} \sum_{i=1}^{k_{s,a,h}-1} \frac{i^{1-\alpha} - (i-1)^{1-\alpha}}{1-\alpha} + \frac{d_{s,a,h}}{k_{s,a,h}^\alpha}$$

$$= \sum_{s,a} \frac{\|\omega\|_{\infty,h}(k_{s,a,h} - 1)^{1-\alpha}}{1-\alpha} + \frac{d_{s,a,h}}{k_{s,a,h}^\alpha}$$

$$= \sum_{s,a} \|\omega\|_{\infty,h}^\alpha \left( \frac{[(k_{s,a,h} - 1)\|\omega\|_{\infty,h}]^{1-\alpha}}{1-\alpha} + \frac{d_{s,a,h}}{(k_{s,a,h}\|\omega\|_{\infty,h})^\alpha} \right)$$

$$\le \sum_{s,a} \|\omega\|_{\infty,h}^\alpha \left( \frac{[(k_{s,a,h} - 1)\|\omega\|_{\infty,h}]^{1-\alpha}}{1-\alpha} + \frac{d_{s,a,h}}{c_h(s,a)^\alpha} \right). \tag{90}$$

Here the last inequality is because $k_{s,a,h}\|\omega\|_{\infty,h} \geq c_h(s,a)$. The second inequality is because for any $0 < y < x$ and $\alpha \in (0,1)$, we have:

$$\frac{x-y}{x^\alpha} \leq \frac{1}{1-\alpha}(x^{1-\alpha} - y^{1-\alpha}).$$

Then, let $x = i$ and $y = i - 1$, it holds that:

$$\frac{1}{i^\alpha} \leq \frac{1}{1-\alpha}(i^{1-\alpha} - (i-1)^{1-\alpha}).$$

Also let $x = c_h(s,a)$ and $y = (k_{s,a,h} - 1)\|\omega\|_{\infty,h}$, we have:

$$\frac{d_{s,a,h}}{c_h(s,a)^\alpha} + \frac{[(k_{s,a,h} - 1)\|\omega\|_{\infty,h}]^{1-\alpha}}{1-\alpha} \leq \frac{c_h(s,a)^{1-\alpha}}{1-\alpha}.$$

Applying this inequality to Equation (90), we have:

$$\sum_{k=1}^{K} \frac{\omega_{h,k}}{N_h^k(s_h^k, a_h^k)^\alpha} \leq \sum_{s,a} \|\omega\|_{\infty,h}^\alpha \frac{c_h(s,a)^{1-\alpha}}{1-\alpha} \leq \frac{1}{1-\alpha}(SA\|\omega\|_{\infty,h})^\alpha \|\omega\|_{1,h}^{1-\alpha}$$

The last inequality is by Hölder's inequality, as $\sum_{s,a} c_h(s,a)^{1-\alpha} \leq (SA)^\alpha \|\omega\|_{1,h}^{1-\alpha}$.
For $\alpha = 1$, it holds that:

$$\sum_{k=1}^{K} \frac{1}{N_h^k(s_h^k, a_h^k)} = \sum_{s,a} \sum_{i=1}^{N_h^K(s,a)} \frac{1}{i} \leq \sum_{s,a} \left(\log(N_h^K(s,a)) + 1\right) \leq SA \log T.$$

$\square$

**Lemma F.4.** *For any non-negative functions $\{X_h^k : \mathcal{S} \to \mathbb{R} \mid k \in [K], h \in [H]\}$ and any $h \in [H]$, we have that*

$$\sum_{k=1}^{K} \sum_{n=1}^{N_h^k(s_h^k, a_h^k)} u_n^{N_h^k} X_{h+1}^{k^n} \lesssim \log(T) \sum_{k=1}^{K} X_{h+1}^k,$$

$$\sum_{k=1}^{K} \sum_{n=1}^{N_h^k(s_h^k, a_h^k)} \eta_n^{N_h^k} X_{h+1}^{k^n} \leq \left(1 + \frac{1}{H}\right) \sum_{k=1}^{K} X_{h+1}^k.$$

*Here, $X_{H+1}^k = 0$ for any $k \in [K]$ and $s \in \mathcal{S}$.*

*Proof.* For the first conclusion, we have

$$\sum_{k=1}^{K} \sum_{n=1}^{N_h^k(s_h^k, a_h^k)} u_n^{N_h^k} X_{h+1}^{k^n} = \sum_{k=1}^{K} \sum_{n=1}^{N_h^k} u_n^{N_h^k} X_{h+1}^{k^n} \left(\sum_{j=1}^{K} \mathbb{I}[k^n = j]\right)$$

$$= \sum_{j=1}^{K} \left(\sum_{k=1}^{K} \sum_{n=1}^{N_h^k} u_n^{N_h^k} \mathbb{I}[k^n = j]\right) X_{h+1}^j. \tag{91}$$

Here $\mathbb{I}[k^n = j] = 1$ if and only if $(s_h^j, a_h^j) = (s_h^k, a_h^k)$, $j \leq k - 1$ and $n = N_h^{j+1}(s_h^j, a_h^j) > 0$. Then we have:

$$\sum_{k=1}^{K} \sum_{n=1}^{N_h^k} u_n^{N_h^k} \mathbb{I}[k^n = j] = \sum_{k=j+1}^{K} u_{N_h^{j+1}}^{N_h^k} \mathbb{I}\left[(s_h^j, a_h^j) = (s_h^k, a_h^k)\right] \leq \sum_{t=N_h^{j+1}}^{N_h^K} u_{N_h^{j+1}}^t \lesssim \log T. \tag{92}$$

The last inequality is because for any $N \in \mathbb{N}_+$ and $i \in [N]$, $u_i^N \leq \frac{2}{N}$. Applying Equation (92) to Equation (91), we finish the proof of the first conclusion. For the second conclusion, it holds:

$$\sum_{k=1}^{K} \sum_{n=1}^{N_h^k(s_h^k, a_h^k)} \eta_n^{N_h^k} X_{h+1}^{k^n} = \sum_{k=1}^{K} \sum_{n=1}^{N_h^k} \eta_n^{N_h^k} X_{h+1}^{k^n} \left(\sum_{j=1}^{K} \mathbb{I}[k^n = j]\right)$$

$$= \sum_{j=1}^{K} \left(\sum_{k=1}^{K} \sum_{n=1}^{N_h^k} \eta_n^{N_h^k} \mathbb{I}[k^n = j]\right) X_{h+1}^j. \tag{93}$$

According to the definition of $k^n$, $\mathbb{I}[k^n = j] = 1$ if and only if $(s_h^j, a_h^j) = (s_h^k, a_h^k)$, $j \leq k-1$ and $n = N_h^{j+1}(s_h^j, a_h^j)$. Then by Equation (87) in Lemma F.1, we have:

$$\sum_{k=1}^{K} \sum_{n=1}^{N_h^k} \eta_n^{N_h^k} \mathbb{I}[k^n = j] = \sum_{k=j+1}^{K} \eta_{N_h^{j+1}}^{N_h^k} \mathbb{I}\left[(s_h^j, a_h^j) = (s_h^k, a_h^k)\right] \leq \sum_{t=N_h^{j+1}}^{\infty} \eta_{N_h^{j+1}}^{t} \leq 1 + \frac{1}{H}. \quad (94)$$

Applying Equation (94) to Lemma F.1, we have proven the second conclusion. $\qquad \square$

**Lemma F.5.** *For any $h \in [H]$ and $k \in [K]$, we have the following two conclusions:*

- *If $V_{h+1}^k(s) - V_{h+1}^{\mathrm{LCB},k}(s) \leq \beta$, then $V_{h+1}^{\mathrm{R},K+1}(s) = V_{h+1}^{\mathrm{R},k}(s) = \hat{V}_{h+1}^{\mathrm{R},k}(s)$.*

- *If $V_{h+1}^k(s) - V_{h+1}^{\mathrm{LCB},k}(s) > \beta$, then we have:*

$$0 \leq V_{h+1}^{\mathrm{R},k}(s) - \hat{V}_{h+1}^{\mathrm{R},k}(s) \leq V_{h+1}^k(s) - V_{h+1}^{\mathrm{LCB},k}(s),$$

*and*

$$|\hat{V}_{h+1}^{\mathrm{R},k}(s) - V_{h+1}^{\mathrm{R},K+1}(s)| \leq V_{h+1}^k(s) - V_{h+1}^{\mathrm{LCB},k}(s).$$

*Proof.* 
- If for given $k \in [K]$, $V_{h+1}^k(s) - V_{h+1}^{\mathrm{LCB},k}(s) \leq \beta$, then there exists $k_1 \in [K]$ such that:

$$k_1 = \min\left\{k : V_{h+1}^k(s) - V_{h+1}^{\mathrm{LCB},k}(s) \leq \beta\right\}.$$

Then according the algorithm, we have $u_{h+1}^{k_1-1}(s) = $ True, or it is contradictory to the minimality of $k_1$. Therefore, in this case we have:

$$V_{h+1}^{\mathrm{R},K+1}(s) = V_{h+1}^{\mathrm{R},k}(s) = V_{h+1}^{\mathrm{R},k_1}(s) = V_{h+1}^{k_1}(s) \leq V_{h+1}^{\mathrm{LCB},k_1}(s) + \beta \leq V_{h+1}^{\star}(s) + \beta,$$

and

$$V_{h+1}^{\mathrm{R},k}(s) = V_{h+1}^{\mathrm{R},k_1}(s) = V_{h+1}^{k_1}(s) \geq V_{h+1}^{\star}(s).$$

According to the definition of $\hat{V}_{h+1}^{\mathrm{R},k}(s)$, we have $\hat{V}_{h+1}^{\mathrm{R},k}(s) = V_{h+1}^{\mathrm{R},k}(s) = V_{h+1}^{\mathrm{R},K+1}(s)$. Thus $V_{h+1}^k(s) - V_{h+1}^{\mathrm{LCB},k}(s) \leq \beta$ is the sufficient condition of $V_{h+1}^{\mathrm{R},k}(s) = \hat{V}_{h+1}^{\mathrm{R},k}(s) = V_{h+1}^{\mathrm{R},K+1}(s)$.

- Moreover, if $V_{h+1}^k(s) - V_{h+1}^{\mathrm{LCB},k}(s) > \beta$, according to the algorithm, we have $V_{h+1}^{\mathrm{R},k}(s) = V_{h+1}^k(s)$ and then $0 \leq V_{h+1}^{\mathrm{R},k}(s) - \hat{V}_{h+1}^{\mathrm{R},k}(s) \leq V_{h+1}^k(s) - V_{h+1}^{\mathrm{LCB},k}(s)$.

In this case, we also have $V_{h+1}^{\mathrm{LCB},k}(s) \leq V_{h+1}^{*}(s) \leq V_{h+1}^{\mathrm{R},K+1}(s) \leq V_{h+1}^{\mathrm{R},k}(s) = V_{h+1}^k(s)$ and then $V_{h+1}^{\mathrm{LCB},k}(s) \leq V_{h+1}^{*}(s) \leq \hat{V}_{h+1}^{\mathrm{R},k}(s) \leq V_{h+1}^{\mathrm{R},k}(s) = V_{h+1}^k(s)$. These two inequalities imply that $|\hat{V}_{h+1}^{\mathrm{R},k}(s) - V_{h+1}^{\mathrm{R},K+1}(s)| \leq V_{h+1}^k(s) - V_{h+1}^{\mathrm{LCB},k}(s)$.

$\qquad \square$

## F.3 STEP 1: BOUNDING $Q_h^k - Q_h^{\star}$

### F.3.1 BOUNDING THE EMPIRICAL ESTIMATION ERRORS

By $\mathcal{E}_6$ in Lemma F.2 we have:

$$\left(\mathbb{P}_{h,k}^{\mathrm{adv}} - \hat{\mathbb{E}}_{h,k}^{\mathrm{adv}}\right)\hat{V}_{h+1}^{\mathrm{adv},k^n} = \sum_{n=1}^{N_h^k} \eta_n^{N_h^k}\left(\mathbb{P}_{s_h^k, a_h^k, h} - \mathbb{1}_{s_{h+1}^{k^n}}\right)\left(\hat{V}_{h+1}^{\mathrm{R},k^i} - V_{h+1}^{\star}\right) \leq 2\sqrt{\frac{\beta^2 H\iota}{N_h^k(s_h^k, a_h^k)}}. \quad (95)$$

By $\mathcal{E}_4$ in Lemma F.2, it holds that:

$$\left(\hat{\mathbb{E}}_{h,k}^{\mathrm{ref}} - \mathbb{P}_{h,k}^{\mathrm{ref}}\right)(\hat{V}_{h+1}^{\mathrm{R},k^n} - V_{h+1}^{\star}) = \sum_{i=1}^{N_h^k} u_i^{N_h^k}\left(\mathbb{1}_{s_{h+1}^{k^i}} - \mathbb{P}_{s_h^k, a_h^k, h}\right)(\hat{V}_{h+1}^{\mathrm{R},k^i} - V_{h+1}^{\star}) \leq 2\sqrt{\frac{2\beta^2\iota}{N_h^k(s_h^k, a_h^k)}}.$$

By $\mathcal{E}_5$ in Lemma F.2, it holds that:

$$\left(\hat{\mathbb{E}}_{h,k}^{\text{ref}} - \mathbb{P}_{h,k}^{\text{ref}}\right) V_{h+1}^{\star} = \sum_{i=1}^{N_h^k} u_i^{N_h^k} \left(\mathbb{1}_{s_{h+1}^{ki}} - \mathbb{P}_{s_h^k, a_h^k, h}\right) V_{h+1}^{\star} \leq 8 \sqrt{\frac{\mathbb{Q}^{\star} \iota}{N_h^k(s_h^k, a_h^k)}} + 16 \frac{H \iota}{N_h^k(s_h^k, a_h^k)}.$$

Therefore, combining these two inequalities, we have:

$$\left(\hat{\mathbb{E}}_{h,k}^{\text{ref}} - \mathbb{P}_{h,k}^{\text{ref}}\right) \hat{V}_{h+1}^{\text{R},k^n} \lesssim \sqrt{\frac{\mathbb{Q}^{\star} + \beta^2}{N_h^k(s_h^k, a_h^k)} \iota} + \frac{H \iota}{N_h^k(s_h^k, a_h^k)}. \tag{96}$$

### F.3.2 BOUNDING THE BONUS

Since the term $\iota^2$ in the last inequality of Lemma 7 in Li et al. (2021) can be easily improved to $\iota$, we can paraphrase the equation (87) and equation (88) of Li et al. (2021) to the following form:

$$b_h^{\text{R},k^n+1} = \left(1 - \frac{1}{\eta_n}\right) B_h^{\text{R},k^n}\left(s_h^k, a_h^k\right) + \frac{1}{\eta_n} B_h^{\text{R},k^n+1}\left(s_h^k, a_h^k\right) + \frac{c_b}{n^{3/4}} H^2 \iota. \tag{97}$$

This taken collectively with the definition of $\eta_n^N$ allows us to expand

$$R^{h,k} = \sum_{n=1}^{N_h^k} \eta_n^N b_h^{\text{R},k^n+1}$$

$$= \sum_{n=1}^{N_h^k} \eta_n \prod_{i=n+1}^{N_h^k} (1 - \eta_i) \left(\left(1 - \frac{1}{\eta_n}\right) B_h^{\text{R},k^n}\left(s_h^k, a_h^k\right) + \frac{1}{\eta_n} B_h^{\text{R},k^n+1}\left(s_h^k, a_h^k\right)\right) + c_b \sum_{n=1}^{N_h^k} \frac{\eta_n^{N_h^k}}{n^{3/4}} H^2 \iota$$

$$= B_h^{\text{R},k^{N_h^k}+1} + c_b \sum_{n=1}^{N_h^k} \frac{\eta_n^{N_h^k}}{n^{3/4}} H^2 \iota. \tag{98}$$

Then with $B_h^{\text{R},k^{N_h^k}+1} = B_h^{\text{R},k}$ and Equation (86) in Lemma F.1, it holds that

$$R^{h,k} \lesssim B_h^{\text{R},k} + \frac{H^2 \iota}{N_h^k(s_h^k, a_h^k)^{\frac{3}{4}}}. \tag{99}$$

Similar to equation (158) of Li et al. (2021), we have:

$$\sqrt{\frac{\sigma_h^{\text{adv},k}(s_h^k, a_h^k) - \left(\mu_h^{\text{adv},k}(s_h^k, a_h^k)\right)^2}{N_h^k(s_h^k, a_h^k)}} \leq \sqrt{\frac{\sum_{n=1}^{N_h^k} \eta_n^{N_h^k} \left(V_{h+1}^{k^n}(s_{h+1}^{k^n}) - V_{h+1}^{\text{R},k^n}(s_{h+1}^{k^n})\right)^2}{N_h^k(s_h^k, a_h^k)}} \leq 2\beta \tag{100}$$

Equation (100) is because $|V_{h+1}^{k^n}(s_{h+1}^{k^n}) - V_{h+1}^{\text{R},k^n}(s_{h+1}^{k^n})| \leq 2\beta$ by $\mathcal{E}_3$ in Lemma F.2 and $\sum_{n=1}^{N_h^k} \eta_n^{N_h^k} \leq 1$. Meanwhile, since $V_{h+1}^{\text{R},k^n}(s_{h+1}^{k^n}) \geq \hat{V}_{h+1}^{\text{R},k^n}(s_{h+1}^{k^n})$, it also holds that

$$\sqrt{\frac{\sigma_h^{\text{ref},k}(s_h^k, a_h^k) - \left(\mu_h^{\text{ref},k}(s_h^k, a_h^k)\right)^2}{N_h^k(s_h^k, a_h^k)}} \leq \sqrt{\frac{J_1^{h,k} + J_2^{h,k}}{N_h^k(s_h^k, a_h^k)}},$$

where:

$$J_1^{h,k} = \frac{\sum_{n=1}^{N_h^k} \left(\left(V_{h+1}^{\text{R},k^n}(s_{h+1}^{k^n})\right)^2 - \left(\hat{V}_{h+1}^{\text{R},k^n}(s_{h+1}^{k^n})\right)^2\right)}{N_h^k(s_h^k, a_h^k)},$$

and

$$J_2^{h,k} = \frac{\sum_{n=1}^{N_h^k} \left(\hat{V}_{h+1}^{\text{R},k^n}(s_{h+1}^{k^n})\right)^2}{N_h^k(s_h^k, a_h^k)} - \left(\frac{\sum_{n=1}^{N_h^k} \hat{V}_{h+1}^{\text{R},k^n}(s_{h+1}^{k^n})}{N_h^k(s_h^k, a_h^k)}\right)^2.$$

Next we want to bound both $J_1^{h,k}$ and $J_2^{h,k}$.

$$J_1^{h,k} = \frac{\sum_{n=1}^{N_h^k} \left( V_{h+1}^{\mathrm{R},k^n}(s_{h+1}^{k^n}) + \hat{V}_{h+1}^{\mathrm{R},k^n}(s_{h+1}^{k^n}) \right) \left( V_{h+1}^{\mathrm{R},k^n}(s_{h+1}^{k^n}) - \hat{V}_{h+1}^{\mathrm{R},k^n}(s_{h+1}^{k^n}) \right)}{N_h^k(s_h^k, a_h^k)}$$

$$\leq \frac{\sum_{n=1}^{N_h^k} 2H \left( V_{h+1}^{\mathrm{R},k^n}(s_{h+1}^{k^n}) - \hat{V}_{h+1}^{\mathrm{R},k^n}(s_{h+1}^{k^n}) \right)}{N_h^k(s_h^k, a_h^k)}.$$

Therefore, we have

$$J_1^{h,k} \leq \frac{2H \Psi_h^k(s_h^k, a_h^k)}{N_h^k(s_h^k, a_h^k)}, \tag{101}$$

where

$$\Psi_h^k(s_h^k, a_h^k) = \sum_{n=1}^{N_h^k} \left( V_{h+1}^{\mathrm{R},k^n}(s_{h+1}^{k^n}) - \hat{V}_{h+1}^{\mathrm{R},k^n}(s_{h+1}^{k^n}) \right).$$

For the second term $J_2^{h,k}$, because of Cauchy's Inequality, we have:

$$J_2^{h,k} = \frac{\sum_{n=1}^{N_h^k} \left( \hat{V}_{h+1}^{\mathrm{R},k^n}(s_{h+1}^{k^n}) - \frac{\sum_{i=1}^{N_h^k} \hat{V}_{h+1}^{\mathrm{R},k^n}(s_{h+1}^{k^n})}{N_h^k(s_h^k, a_h^k)} \right)^2}{N_h^k(s_h^k, a_h^k)} \leq 2(J_{2,1}^{h,k} + J_{2,2}^{h,k}),$$

where:

$$J_{2,1}^{h,k} = \frac{\sum_{n=1}^{N_h^k} \left( \hat{V}_{h+1}^{\mathrm{R},k^n}(s_{h+1}^{k^n}) - V_{h+1}^{\star}(s_{h+1}^{k^n}) + \frac{\sum_{i=1}^{N_h^k} V_{h+1}^{\star}(s_{h+1}^{k^n})}{N_h^k(s_h^k, a_h^k)} - \frac{\sum_{i=1}^{N_h^k} \hat{V}_{h+1}^{\mathrm{R},k^n}(s_{h+1}^{k^n})}{N_h^k(s_h^k, a_h^k)} \right)^2}{N_h^k(s_h^k, a_h^k)},$$

and

$$J_{2,2}^{h,k} = \frac{\sum_{n=1}^{N_h^k} \left( V_{h+1}^{\star}(s_{h+1}^{k^n}) - \frac{\sum_{i=1}^{N_h^k} V_{h+1}^{\star}(s_{h+1}^{k^n})}{N_h^k(s_h^k, a_h^k)} \right)^2}{N_h^k(s_h^k, a_h^k)}$$

$$= \frac{\sum_{n=1}^{N_h^k} \left( V_{h+1}^{\star}(s_{h+1}^{k^n}) \right)^2}{N_h^k(s_h^k, a_h^k)} - \left( \frac{\sum_{n=1}^{N_h^k} V_{h+1}^{\star}(s_{h+1}^{k^n})}{N_h^k(s_h^k, a_h^k)} \right)^2.$$

Since $V_{h+1}^{\star}(s_{h+1}^{k^n}) \leq \hat{V}_{h+1}^{\mathrm{R},k^n}(s_{h+1}^{k^n}) \leq V_{h+1}^{\star}(s_{h+1}^{k^n}) + \beta$, it holds that:

$$\left| \hat{V}_{h+1}^{\mathrm{R},k^n}(s_{h+1}^{k^n}) - V_{h+1}^{\star}(s_{h+1}^{k^n}) + \frac{\sum_{i=1}^{N_h^k} V_{h+1}^{\star}(s_{h+1}^{k^n})}{N_h^k(s_h^k, a_h^k)} - \frac{\sum_{i=1}^{N_h^k} \hat{V}_{h+1}^{\mathrm{R},k^n}(s_{h+1}^{k^n})}{N_h^k(s_h^k, a_h^k)} \right|$$

$$\leq \left| \hat{V}_{h+1}^{\mathrm{R},k^n}(s_{h+1}^{k^n}) - V_{h+1}^{\star}(s_{h+1}^{k^n}) \right| + \left| \frac{\sum_{i=1}^{N_h^k} V_{h+1}^{\star}(s_{h+1}^{k^n})}{N_h^k(s_h^k, a_h^k)} - \frac{\sum_{i=1}^{N_h^k} \hat{V}_{h+1}^{\mathrm{R},k^n}(s_{h+1}^{k^n})}{N_h^k(s_h^k, a_h^k)} \right| \leq 2\beta.$$

Therefore, applying this inequality to $J_{2,1}^{h,k}$, we have $J_{2,1}^{h,k} \leq 4\beta^2$. Moreover, according to equation (165) of Li et al. (2021), the following inequality holds:

$$J_{2,2}^{h,k} \lesssim \mathbb{Q}^{\star} + H^2 \sqrt{\frac{\iota}{N_h^k(s_h^k, a_h^k)}}.$$

Combining the upper bounds of $J_1^{h,k}$ Equation (101), $J_{2,1}^{h,k}$ and $J_{2,2}^{h,k}$, we have:

$$\sqrt{\frac{\sigma_h^{\mathrm{ref},k}(s_h^k, a_h^k) - \left( \mu_h^{\mathrm{ref},k}(s_h^k, a_h^k) \right)^2}{N_h^k(s_h^k, a_h^k)}} \lesssim \frac{\sqrt{H \Psi_h^k(s_h^k, a_h^k)}}{N_h^k(s_h^k, a_h^k)} + \sqrt{\frac{\mathbb{Q}^{\star} + \beta^2}{N_h^k(s_h^k, a_h^k)}} + \frac{H \iota^{\frac{1}{4}}}{N_h^k(s_h^k, a_h^k)^{\frac{3}{4}}}. \tag{102}$$

Back to the definition of $B_h^{\mathrm{R},k}$ in Algorithm 2, combining Equation (100) and Equation (102), it holds:

$$
B_h^{\mathrm{R},k} \leq c_b \sqrt{\iota} \sqrt{\frac{\sigma_h^{\mathrm{ref},k}(s_h^k, a_h^k) - \left(\mu_h^{\mathrm{ref},k}(s_h^k, a_h^k)\right)^2}{N_h^k(s_h^k, a_h^k)}} + c_b \sqrt{H\iota} \sqrt{\frac{\sigma_h^{\mathrm{adv},k}(s_h^k, a_h^k) - \left(\mu_h^{\mathrm{adv},k}(s_h^k, a_h^k)\right)^2}{N_h^k(s_h^k, a_h^k)}}
$$

$$
\lesssim \frac{\sqrt{H\Psi_h^k(s_h^k, a_h^k)\iota}}{N_h^k(s_h^k, a_h^k)} + \sqrt{\frac{(\mathbb{Q}^\star + \beta^2 H)\iota}{N_h^k(s_h^k, a_h^k)}} + \frac{H\iota^{\frac{3}{4}}}{N_h^k(s_h^k, a_h^k)^{\frac{3}{4}}}.
$$

Then by Equation (99), we have

$$
R^{h,k} \lesssim \frac{\sqrt{H\Psi_h^k(s_h^k, a_h^k)\iota}}{N_h^k(s_h^k, a_h^k)} + \sqrt{\frac{(\mathbb{Q}^\star + \beta^2 H)\iota}{N_h^k(s_h^k, a_h^k)}} + \frac{H^2\iota}{N_h^k(s_h^k, a_h^k)^{\frac{3}{4}}}. \tag{103}
$$

Applying Equation (95), Equation (96), Equation (103) to Equation (14), it holds that:

$$
(Q_h^k - Q_h^\star)(s_h^k, a_h^k) \leq \hat{\mathbb{E}}_{h,k}^{\mathrm{adv}}(V_{h+1}^{k^n} - V_{h+1}^\star) + \sqrt{\frac{(\mathbb{Q}^\star + \beta^2 H)\iota}{N_h^k(s_h^k, a_h^k)}} + \frac{H^2\iota}{N_h^k(s_h^k, a_h^k)^{\frac{3}{4}}} + R_{\mathrm{else}}^{h,k}. \tag{104}
$$

Here

$$
R_{\mathrm{else}}^{h,k} = \eta_0^{N_h^k} H + \hat{\mathbb{E}}_{h,k}^{\mathrm{ref}}(V_h^{\mathrm{R},k^n} - \hat{V}_h^{\mathrm{R},k^n}) + \left(\mathbb{P}_{h,k}^{\mathrm{ref}} - \mathbb{P}_{h,k}^{\mathrm{adv}}\right)\hat{V}_{h+1}^{\mathrm{R},k^n} + \frac{\sqrt{H\Psi_h^k(s_h^k, a_h^k)\iota}}{N_h^k(s_h^k, a_h^k)} + \frac{H\iota}{N_h^k(s_h^k, a_h^k)}.
$$

### F.4 STEP 2: BOUNDING THE WEIGHTED SUM

### F.4.1 REARRANGING THE SUMMATION

$$
\sum_{k=1}^K \omega_{h,k} \hat{\mathbb{E}}_{h,k}^{\mathrm{adv}}(V_{h+1}^{k^n} - V_{h+1}^\star) = \sum_{k=1}^K \sum_{n=1}^{N_h^k} \omega_{h,k} \eta_n^{N_h^k} \left(V_{h+1}^{k^n}(s_{h+1}^{k^n}) - V_{h+1}^\star(s_{h+1}^{k^n})\right)
$$

$$
= \sum_{j=1}^K \left(\sum_{k=1}^K \sum_{n=1}^{N_h^k} \omega_{h,k} \eta_n^{N_h^k} \mathbb{I}\left[k^n = j\right]\right) \left(V_{h+1}^j(s_{h+1}^j) - V_{h+1}^\star(s_{h+1}^j)\right)
$$

$$
\leq \sum_{j=1}^K \left(\sum_{k=1}^K \sum_{n=1}^{N_h^k} \omega_{h,k} \eta_n^{N_h^k} \mathbb{I}\left[k^n = j\right]\right) \left(Q_{h+1}^j - Q_{h+1}^\star\right)(s_{h+1}^j, a_{h+1}^j)
$$

$$
\triangleq \sum_{j=1}^K \omega_{h+1,j}(h) \left(Q_{h+1}^j(s_{h+1}^j, a_{h+1}^j) - Q_{h+1}^\star(s_{h+1}^j, a_{h+1}^j)\right). \tag{105}
$$

Here, for any $j \in [K]$

$$
\omega_{h+1,j}(h) = \sum_{k=1}^K \sum_{n=1}^{N_h^k} \omega_{h,k} \eta_n^{N_h^k} \mathbb{I}\left[k^n = j\right].
$$

The inequality is because $Q_{h+1}^j(s_{h+1}^j, a_{h+1}^j) = V_{h+1}^j(s_{h+1}^j)$, $Q_{h+1}^\star(s_{h+1}^j, a_{h+1}^j) \leq V_{h+1}^\star(s_{h+1}^j)$.

### F.4.2 PROOF OF EQUATION (20)

By Equation (94), for $h < h' \leq H$ and any $j \in [K]$, it holds that:

$$
\omega_{h',j}(h) \leq \|\omega(h)\|_{\infty,h'-1} \sum_{k=1}^K \sum_{n=1}^{N_h^k} \eta_n^{N_h^k} \mathbb{I}\left[k^n = j\right] \leq (1 + \frac{1}{H})\|\omega(h)\|_{\infty,h'-1}. \tag{106}
$$

It also holds that:

$$\sum_{j=1}^{K} \omega_{h',j}(h) = \sum_{k=1}^{K} \omega_{h,k} \sum_{n=1}^{N_h^k} \eta_n^{N_h^k} \le \|\omega(h)\|_{1,h'-1}. \tag{107}$$

Combining Equation (104) with Equation (105), the weighted sum $\sum_{k=1}^{K} \omega_{h,k}(Q_h^k - Q_h^\star)(s_h^k, a_h^k)$ can be bounded by

$$\sum_{k=1}^{K} \omega_{h,k}(Q_h^k(s_h^k, a_h^k) - Q_h^\star(s_h^k, a_h^k))$$

$$\lesssim \sum_{k=1}^{K} \omega_{h+1,k}(h)(Q_{h+1}^k - Q_{h+1}^\star)(s_{h+1}^k, a_{h+1}^k) + \sum_{k=1}^{K} \omega_{h,k}\left(\sqrt{\frac{(\mathbb{Q}^\star + \beta^2 H)\iota}{N_h^k(s_h^k, a_h^k)}} + \frac{H^2\iota}{(N_h^k)^{\frac{3}{4}}} + R_{\text{else}}^{h,k}\right)$$

$$\le \sum_{k=1}^{K} \omega_{h+1,k}(h)(Q_{h+1}^k - Q_{h+1}^\star)(s_{h+1}^k, a_{h+1}^k) + \sqrt{(\mathbb{Q}^\star + \beta^2)SA\|\omega\|_{\infty,h}\|\omega\|_{1,h}\iota}$$

$$+ H^2\iota(SA\|\omega\|_{\infty,h})^{\frac{3}{4}}\|\omega\|_{1,h}^{\frac{1}{4}} + \sum_{k=1}^{K} \omega_{h,k}R_{\text{else}}^{h,k}. \tag{108}$$

The last inequality is by Lemma F.3 with $\alpha = \frac{1}{2}$ and $\frac{3}{4}$. Recurring Equation (108) with regard to $h, h+1, \ldots, H$, since $Q_{H+1}^k(s,a) = Q_{H+1}^\star(s,a) = 0$ and the weight relationship Equation (106) and Equation (107), we have

$$\sum_{k=1}^{K} \omega_{h,k}(Q_h^k(s_h^k, a_h^k) - Q_h^\star(s_h^k, a_h^k))$$

$$\lesssim H\sqrt{(\mathbb{Q}^\star + \beta^2 H)SA\|\omega\|_{\infty,h}\|\omega\|_{1,h}\iota} + H^3\iota(SA\|\omega\|_{\infty,h})^{\frac{3}{4}}\|\omega\|_{1,h}^{\frac{1}{4}} + \sum_{k=1}^{K}\sum_{h'=h}^{H} \omega_{h',k}(h)R_{\text{else}}^{h',k}. \tag{109}$$

### F.5 STEP 3: INTEGRATING MULTIPLE WEIGHTED SUMS

#### F.5.1 PROOF OF EQUATION (25)

For any $N = \lceil \log_2(H/\Delta_{\min}) \rceil$, $i \in [N]$, $k \in [K]$ and the given $h \in [H]$, let:

$$\omega_{h,k}^{(i)} = \mathbb{I}\left[Q_h^k(s_h^k, a_h^k) - Q_h^\star(s_h^k, a_h^k) \in [2^{i-1}\Delta_{\min}, 2^i\Delta_{\min}]\right],$$

and

$$\omega_{h,k}^{(N)} = \mathbb{I}\left[Q_h^k(s_h^k, a_h^k) - Q_h^\star(s_h^k, a_h^k) \in [2^{N-1}\Delta_{\min}, H]\right].$$

Then

$$\|\omega^{(i)}\|_{\infty,h} = \max_k \omega_{h,k}^{(i)} \le 1, \ \|\omega^{(i)}\|_{1,h} = \sum_{k=1}^{K} \omega_{h,k}^{(i)}.$$

For any given $i \in [N]$ and $h \le h' \le H$, the weight $\{\omega_{h',k}^{(i)}\}_k$ can be defined recursively by Equation (18). Therefore, for any $j \in [K]$, it holds that:

$$\sum_{i=1}^{N} \omega_{h'+1,j}^{(i)}(h) = \sum_{k=1}^{K}\sum_{n=1}^{N_{h'}^k}\left(\sum_{i=1}^{N} \omega_{h',k}^{(i)}(h)\right)\eta_n^{N_{h'}^k}\mathbb{I}\left[k^n = j\right].$$

Here for any $i \in [N]$, $\omega_{h,k}^{(i)}(h) = \omega_{h,k}^{(i)}$. Then by mathematical induction on $h' \in [h, H]$, it is straightforward to prove that for any $j \in [K]$,

$$\sum_{i=1}^{N} \omega_{h',j}^{(i)}(h) \le \left(1 + \frac{1}{H}\right)^{h'-h} < 3, \tag{110}$$

given that for any $j \in [K]$

$$\sum_{k=1}^{K} \sum_{n=1}^{N_{h'}^{k}} \eta_n^{N_{h'}^{k}} \mathbb{I}[k^n = j] \leq 1 + \frac{1}{H}$$

by Equation (94) and $\sum_{i=1}^{N} \omega_{h,j}^{(i)}(h) = \sum_{i=1}^{N} \omega_{h,j}^{(i)} \leq 1$.

Applying the weight $\{\omega_{h,k}^{(i)}\}_k$ to Equation (109), since $\|\omega^{(i)}\|_{\infty,h} \leq 1$, then for any $i \in [N]$, it holds:

$$\sum_{k=1}^{K} \omega_{h,k}^{(i)}(Q_h^k(s_h^k, a_h^k) - Q_h^\star(s_h^k, a_h^k)) \lesssim H\sqrt{(\mathbb{Q}^\star + \beta^2 H)SA\|\omega^{(i)}\|_{1,h}\iota}$$

$$+ H^3\iota(SA)^{\frac{3}{4}}(\|\omega^{(i)}\|_{1,h})^{\frac{1}{4}} + \sum_{k=1}^{K}\sum_{h'=h}^{H} \omega_{h',k}^{(i)}(h)R_{\text{else}}^{h',k}.$$

On the other hand, according to the definition of $\omega_{h,k}^{(i)}$, we have

$$\sum_{k=1}^{K} \omega_{h,k}^{(i)}\left(Q_h^k(s_h^k, a_h^k) - Q_h^\star(s_h^k, a_h^k)\right) \geq 2^{i-1}\Delta_{\min}\|\omega^{(i)}\|_{1,h}.$$

Therefore, since $\|\omega^{(i)}\|_{\infty,h} \leq 1$, we obtain the following inequality for any $i \in [N]$:

$$2^{i-1}\Delta_{\min}\|\omega^{(i)}\|_{1,h} \lesssim H\sqrt{(\mathbb{Q}^\star + \beta^2 H)SA\|\omega^{(i)}\|_{1,h}\iota} + H^3\iota(SA)^{\frac{3}{4}}(\|\omega^{(i)}\|_{1,h})^{\frac{1}{4}}$$

$$+ \sum_{k=1}^{K}\sum_{h'=h}^{H} \omega_{h',k}^{(i)}(h)R_{\text{else}}^{h',k}. \tag{111}$$

Then at least one of the following three inequalities holds:

$$2^{i-1}\Delta_{\min}\|\omega^{(i)}\|_{1,h} \lesssim H\sqrt{(\mathbb{Q}^\star + \beta^2 H)SA\|\omega^{(i)}\|_{1,h}\iota}$$

$$2^{i-1}\Delta_{\min}\|\omega^{(i)}\|_{1,h} \lesssim H^3\iota(SA)^{\frac{3}{4}}(\|\omega^{(i)}\|_{1,h})^{\frac{1}{4}},$$

$$2^{i-1}\Delta_{\min}\|\omega^{(i)}\|_{1,h} \lesssim \sum_{k=1}^{K}\sum_{h'=h}^{H} \omega_{h',k}^{(i)}(h)R_{\text{else}}^{h',k}.$$

Solving this three inequalities, we know that:

$$\|\omega^{(i)}\|_{1,h} \leq O\left(\max\left\{\frac{(\mathbb{Q}^\star + \beta^2 H)SAH^2\iota}{4^{i-1}\Delta_{\min}^2}, \frac{H^4 SA\iota^{\frac{4}{3}}}{(2^{i-1}\Delta_{\min})^{\frac{4}{3}}}, \frac{\sum_{k=1}^{K}\sum_{h'=h}^{H} \omega_{h',k}^{(i)}(h)R_{\text{else}}^{h',k}}{2^{i-1}\Delta_{\min}}\right\}\right)$$

$$\leq O\left(\frac{(\mathbb{Q}^\star + \beta^2 H)SAH^2\iota}{4^{i-1}\Delta_{\min}^2} + \frac{H^4 SA\iota^{\frac{4}{3}}}{(2^{i-1}\Delta_{\min})^{\frac{4}{3}}} + \frac{\sum_{k=1}^{K}\sum_{h'=h}^{H} \omega_{h',k}^{(i)}(h)R_{\text{else}}^{h',k}}{2^{i-1}\Delta_{\min}}\right). \tag{112}$$

By Equation (110), we have:

$$\sum_{i=1}^{N}\sum_{k=1}^{K}\sum_{h'=h}^{H} \omega_{h',k}^{(i)}(h)R_{\text{else}}^{h',k} = \sum_{h'=h}^{H}\sum_{k=1}^{K}\left(\sum_{i=1}^{N} \omega_{h',k}^{(i)}(h)\right)R_{\text{else}}^{h',k} \leq 3\sum_{h'=1}^{H}\sum_{k=1}^{K} R_{\text{else}}^{h',k}.$$

Using this inequality, we have

$$\sum_{i=1}^{N} 2^i \Delta_{\min}\|\omega^{(i)}\|_{1,h} \leq O\left(\frac{(\mathbb{Q}^\star + \beta^2 H)SAH^2\iota}{\Delta_{\min}} + \frac{H^4 SA\iota^{\frac{4}{3}}}{(\Delta_{\min})^{\frac{1}{3}}} + \sum_{h'=1}^{H}\sum_{k=1}^{K} R_{\text{else}}^{h',k}\right). \tag{113}$$

### F.5.2 PROOF OF EQUATION (26) AND EQUATION (27)

Next we will bound the term $\sum_{h'=1}^{H} \sum_{k=1}^{K} R_{\text{else}}^{h',k}$, where

$$R_{\text{else}}^{h',k} = \eta_0^{N_{h'}^k} H + \hat{\mathbb{E}}_{h',k}^{\text{ref}} \left( V_{h'+1}^{\text{R},k^n} - \hat{V}_{h'+1}^{\text{R},k^n} \right) + \left( \mathbb{P}_{h',k}^{\text{ref}} \hat{V}_{h'+1}^{\text{R},k^n} - \mathbb{P}_{h',k}^{\text{adv}} \hat{V}_{h'+1}^{\text{R},k^n} \right) + \frac{\sqrt{H \Psi_{h'}^k \iota}}{N_{h'}^k} + \frac{H\iota}{N_{h'}^k}.$$

According to equation (149) of Li et al. (2021), we have:

$$\sum_{h'=1}^{H} \sum_{k=1}^{K} \eta_0^{N_{h'}^k} H \leq H^2 SA \leq \frac{H^6 SA \log(T)\iota}{\beta}. \tag{114}$$

By Lemma F.4, we have

$$\sum_{h'=1}^{H} \sum_{k=1}^{K} \hat{\mathbb{E}}_{h',k}^{\text{ref}} \left( V_{h'+1}^{\text{R},k^n} - \hat{V}_{h'+1}^{\text{R},k^n} \right) \lesssim \log T \sum_{h'=1}^{H} \sum_{j=1}^{K} \left( V_{h'+1}^{\text{R},j} - \hat{V}_{h'+1}^{\text{R},j} \right)(s_{h'+1}^{k^i}). \tag{115}$$

By Lemma F.5, the following inequality holds:

$$\sum_{h'=1}^{H} \sum_{j=1}^{K} \left( V_{h'+1}^{\text{R},j}(s_{h'+1}^j) - \hat{V}_{h'+1}^{\text{R},j}(s_{h'+1}^j) \right)$$

$$\leq \sum_{h'=1}^{H} \sum_{j=1}^{K} \left( V_{h'+1}^j(s_{h'+1}^j) - V_{h'+1}^{\text{LCB},j}(s_{h'+1}^j) \right) \mathbb{I}\left[ V_{h'+1}^j(s_{h'+1}^j) - V_{h'+1}^{\text{LCB},j}(s_{h'+1}^j) > \beta \right] \lesssim \frac{H^6 SA \iota}{\beta}.$$

The last inequality is by $\mathcal{E}_3$ in Lemma F.2. Applying this inequality to Equation (115), it holds that:

$$\sum_{h'=1}^{H} \sum_{k=1}^{K} \hat{\mathbb{E}}_{h',k}^{\text{ref}} \left( V_{h'}^{\text{R},k^n} - \hat{V}_{h'}^{\text{R},k^n} \right) \lesssim \frac{H^6 SA \log(T)\iota}{\beta} \tag{116}$$

For the third term in $R_{\text{else}}^{h',k}$, because $\sum_{n=1}^{N_{h'}^k} u_n^{N_{h'}^k} = \sum_{n=1}^{N_{h'}^k} \eta_n^{N_{h'}^k}$, then

$$\mathbb{P}_{h',k}^{\text{ref}} \hat{V}_{h'+1}^{\text{R},k^n} - \mathbb{P}_{h',k}^{\text{adv}} \hat{V}_{h'+1}^{\text{R},k^n}$$

$$= \sum_{n=1}^{N_{h'}^k} u_n^{N_{h'}^k} \mathbb{P}_{s_{h'}^k, a_{h'}^k, h'} (\hat{V}_{h'+1}^{\text{R},k^n} - V_{h'+1}^{\text{R},K+1}) - \sum_{n=1}^{N_{h'}^k} \eta_n^{N_{h'}^k} \mathbb{P}_{s_{h'}^k, a_{h'}^k, h'} (\hat{V}_{h'+1}^{\text{R},k^n} - V_{h'+1}^{\text{R},K+1})$$

$$\leq \sum_{n=1}^{N_{h'}^k} u_n^{N_{h'}^k} \mathbb{P}_{s_{h'}^k, a_{h'}^k, h'} \left| \hat{V}_{h'+1}^{\text{R},k^n} - V_{h'+1}^{\text{R},K+1} \right| + \sum_{n=1}^{N_{h'}^k} \eta_n^{N_{h'}^k} \mathbb{P}_{s_{h'}^k, a_{h'}^k, h'} \left| \hat{V}_{h'+1}^{\text{R},k^n} - V_{h'+1}^{\text{R},K+1} \right|$$

By Lemma F.4, we have:

$$\sum_{h'=1}^{H} \sum_{k=1}^{K} \sum_{n=1}^{N_{h'}^k} u_n^{N_{h'}^k} \mathbb{P}_{s_{h'}^k, a_{h'}^k, h'} \left| \hat{V}_{h'+1}^{\text{R},k^n} - V_{h'+1}^{\text{R},K+1} \right| \lesssim \log(T) \sum_{h'=1}^{H} \sum_{j=1}^{K} \mathbb{P}_{s_{h'}^k, a_{h'}^k, h'} \left| \hat{V}_{h'+1}^{\text{R},j} - V_{h'+1}^{\text{R},K+1} \right|.$$

and

$$\sum_{h'=1}^{H} \sum_{k=1}^{K} \sum_{n=1}^{N_{h'}^k} \eta_n^{N_{h'}^k} \mathbb{P}_{s_{h'}^k, a_{h'}^k, h'} \left| \hat{V}_{h'+1}^{\text{R},k^n} - V_{h'+1}^{\text{R},K+1} \right| \lesssim \sum_{h'=1}^{H} \sum_{j=1}^{K} \mathbb{P}_{s_{h'}^k, a_{h'}^k, h'} \left| \hat{V}_{h'+1}^{\text{R},j} - V_{h'+1}^{\text{R},K+1} \right|$$

Combining these two inequalities, we have:

$$\sum_{h'=1}^{H} \sum_{k=1}^{K} \left( \mathbb{P}_{h',k}^{\text{ref}} \hat{V}_{h'+1}^{\text{R},k^n} - \mathbb{P}_{h',k}^{\text{adv}} \hat{V}_{h'+1}^{\text{R},k^n} \right) \lesssim \log(T) \sum_{h'=1}^{H} \sum_{j=1}^{K} \mathbb{P}_{s_{h'}^k, a_{h'}^k, h'} \left| \hat{V}_{h'+1}^{\text{R},j} - V_{h'+1}^{\text{R},K+1} \right|. \tag{117}$$

According to Lemma F.5, the following inequality holds:

$$\sum_{h'=1}^{H} \sum_{j=1}^{K} \mathbb{P}_{s_{h'}^{k}, a_{h'}^{k}, h'} \left| \hat{V}_{h'+1}^{\mathrm{R},j}(s_{h'+1}^{j}) - V_{h'+1}^{\mathrm{R},K+1}(s_{h'+1}^{j}) \right|$$

$$\leq \sum_{h'=1}^{H} \sum_{j=1}^{K} \mathbb{P}_{s_{h'}^{k}, a_{h'}^{k}, h'} \left\{ \left( V_{h'+1}^{j} - V_{h'+1}^{\mathrm{LCB},j} \right)(s_{h'+1}^{j}) \mathbb{I} \left[ \left( V_{h'+1}^{j} - V_{h'+1}^{\mathrm{LCB},j} \right)(s_{h'+1}^{j}) > \beta \right] \right\} \lesssim \frac{H^6 S A \iota}{\beta}.$$

The last inequality is because of the events $\mathcal{E}_3$ and $\mathcal{E}_7$ in Lemma F.2. Together with Equation (117), we have:

$$\sum_{h'=1}^{H} \sum_{k=1}^{K} \left( \mathbb{P}_{h',k}^{\mathrm{ref}} \hat{V}_{h'+1}^{\mathrm{R},k^n} - \mathbb{P}_{h',k}^{\mathrm{adv}} \hat{V}_{h'+1}^{\mathrm{R},k^n} \right) \lesssim \frac{H^6 S A \log(T) \iota}{\beta}. \tag{118}$$

Now we move to the fourth term in $R_{\mathrm{else}}^{h,k}$. By Lemma F.5 we have:

$$\Psi_{h'}^{k}(s_{h'}^{k}, a_{h'}^{k}) = \sum_{n=1}^{N_{h'}^{k}} \left( V_{h'+1}^{\mathrm{R},k^n}(s_{h'+1}^{k^n}) - \hat{V}_{h'+1}^{\mathrm{R},k^n}(s_{h'+1}^{k^n}) \right)$$

$$\leq \sum_{n=1}^{N_{h'}^{k}} \left( V_{h'+1}^{k^n}(s_{h'+1}^{k^n}) - V_{h'+1}^{\mathrm{LCB},k^n}(s_{h'+1}^{k^n}) \right) \cdot \mathbb{I} \left[ V_{h'+1}^{k^n}(s_{h'+1}^{k^n}) - V_{h'+1}^{\mathrm{LCB},k^n}(s_{h'+1}^{k^n}) > \beta \right]$$

$$\triangleq \Phi_{h'}^{k}(s_{h'}^{k}, a_{h'}^{k})$$

Then it holds that:

$$\sum_{k=1}^{K} \frac{\sqrt{\Psi_{h'}^{k}(s_{h'}^{k}, a_{h'}^{k})}}{N_{h'}^{k}(s_{h'}^{k}, a_{h'}^{k})} \leq \sum_{k=1}^{K} \frac{\sqrt{\Phi_{h'}^{k}(s_{h'}^{k}, a_{h'}^{k})}}{N_{h'}^{k}(s_{h'}^{k}, a_{h'}^{k})}$$

$$= \sum_{s,a} \sum_{n=1}^{N_{h'}^{K}(s,a)} \frac{\sqrt{\Phi_{h'}^{k^n}(s,a) \mathbb{I} \left[ (s_{h'}^{k}, a_{h'}^{k}) = (s,a) \right]}}{n}$$

$$\leq \log T \sum_{s,a} \sqrt{\Phi_{h'}^{K}(s,a) \mathbb{I} \left[ (s_{h'}^{k}, a_{h'}^{k}) = (s,a) \right]}$$

$$\leq \log T \sqrt{SA \sum_{s,a} \Phi_{h'}^{K}(s,a) \mathbb{I} \left[ (s_{h'}^{k}, a_{h'}^{k}) = (s,a) \right]} \tag{119}$$

The second inequality is because of the mononicity of $\Phi_{h'}^{n}(s,a)$. The last inequality is by Cauchy-Schwartz inequality. To continue, note that:

$$\sum_{h'=1}^{H} \sqrt{\sum_{s,a} \Phi_{h'}^{K}(s,a) \mathbb{I} \left[ (s_{h'}^{k}, a_{h'}^{k}) = (s,a) \right]}$$

$$= \sum_{h'=1}^{H} \sqrt{\sum_{k=1}^{K} \left( V_{h'+1}^{k}(s_{h'+1}^{k}) - V_{h'+1}^{\mathrm{LCB},k}(s_{h'+1}^{k}) \right) \cdot \mathbb{I} \left[ V_{h'+1}^{k}(s_{h'+1}^{k}) - V_{h'+1}^{\mathrm{LCB},k}(s_{h'+1}^{k}) > \beta \right]}$$

$$\leq \sqrt{H \sum_{h'=1}^{H} \sum_{k=1}^{K} \left( V_{h'+1}^{k}(s_{h'+1}^{k}) - V_{h'+1}^{\mathrm{LCB},k}(s_{h'+1}^{k}) \right) \cdot \mathbb{I} \left[ V_{h'+1}^{k}(s_{h'+1}^{k}) - V_{h'+1}^{\mathrm{LCB},k}(s_{h'+1}^{k}) > \beta \right]}$$

$$\leq \sqrt{\frac{H^7 S A \iota}{\beta}}$$

The first inequality uses Cauchy-Schwartz inequality and the last inequality is by $\mathcal{E}_3$ in Lemma F.2. Together with Equation (119), it holds:

$$\sum_{h'=1}^{H} \sum_{k=1}^{K} \frac{\sqrt{H \Psi_{h'}^{k}(s_{h'}^{k}, a_{h'}^{k}) \iota}}{N_{h'}^{k}(s_{h'}^{k}, a_{h'}^{k})} \lesssim \frac{H^4 S A \log(T) \iota}{\sqrt{\beta}}. \tag{120}$$

By Lemma F.3 with $\alpha = 1$, we have:

$$\sum_{h'=1}^{H} \sum_{k=1}^{K} \frac{H\iota}{N_{h'}^{k}(s_{h'}^{k}, a_{h'}^{k})} \leq H^2 S A \log(T) \iota. \tag{121}$$

By summing Equation (114), Equation (116), Equation (118), Equation (120) and Equation (121), since $\beta \in (0, H]$, we can conclude that:

$$\sum_{h'=1}^{H} \sum_{k=1}^{K} R_{\text{else}}^{h',k} \lesssim \frac{H^6 S A \log(T) \iota}{\beta}.$$

Then we have

$$\sum_{k=1}^{K} \text{clip}[(Q_h^k - Q_h^\star)(s_h^k, a_h^k) \mid \Delta_{\min}] = O\left( \frac{(\mathbb{Q}^\star + \beta^2 H) S A H^2 \iota}{\Delta_{\min}} + \frac{H^4 S A \iota^{\frac{4}{3}}}{(\Delta_{\min})^{\frac{1}{3}}} + \frac{H^6 S A \log(T) \iota}{\beta} \right)$$

$$\leq O\left( \frac{(\mathbb{Q}^\star + \beta^2 H) S A H^2 \iota}{\Delta_{\min}} + \frac{H^6 S A \iota^2}{\beta} \right). \tag{122}$$

The last inequality is because

$$\frac{H^4 S A \iota^{\frac{4}{3}}}{(\Delta_{\min})^{\frac{1}{3}}} \leq \frac{\beta^2 H^3 S A \iota}{\Delta_{\min}} + \frac{H^5 S A \iota}{\beta} + \frac{H^5 S A \iota^2}{\beta}$$

by AM-GM inequality.

## F.6 STEP 4: BOUNDING THE EXPECTED GAP-DEPENDENT REGRET

By Equation (9), $Q_h^k(s_h^k, a_h^k) = V_h^k(s_h^k) \geq V_h^\star(s_h^k)$. Thus, for any episode-step pair $(k, h)$

$$\Delta_h(s_h^k, a_h^k) = \text{clip}[V_h^*(s_h^k) - Q_h^*(s_h^k, a_h^k) \mid \Delta_{\min}] \leq \text{clip}[(Q_h^k - Q_h^*)(s_h^k, a_h^k) \mid \Delta_{\min}].$$

By Equation (4) in Yang et al. (2021), we have $\mathbb{E}(\text{Regret}(K)) = \mathbb{E}\left[ \sum_{k=1}^{K} \sum_{h=1}^{H} \Delta_h(s_h^k, a_h^k) \right]$, which further implies

$$\mathbb{E}(\text{Regret}(K)) \leq \mathbb{E}\left[ \sum_{k=1}^{K} \sum_{h=1}^{H} \text{clip}[(Q_h^k - Q_h^*)(s_h^k, a_h^k) \mid \Delta_{\min}] \right].$$

Finally, let $\delta = \frac{1}{7T}$ and $\mathcal{E} = \bigcap_{i=1}^{7} \mathcal{E}_i$ with $\mathcal{E}_i$ in Lemma F.2. Then the event $\mathcal{E}$ holds with probability at least $1 - 7\delta = 1 - \frac{1}{T}$ and we also have:

$$\mathbb{E}(\text{Regret}(K)) \leq \mathbb{E}\left[ \sum_{k=1}^{K} \sum_{h=1}^{H} \text{clip}[(Q_h^k - Q_h^*)(s_h^k, a_h^k) \mid \Delta_{\min}] \Big| \mathcal{E} \right] \mathbb{P}(\mathcal{E})$$

$$+ \mathbb{E}\left[ \sum_{k=1}^{K} \sum_{h=1}^{H} \text{clip}[(Q_h^k - Q_h^*)(s_h^k, a_h^k) \mid \Delta_{\min}] \Big| \mathcal{E}^c \right] \mathbb{P}(\mathcal{E}^c)$$

$$\leq O\left( \frac{(\mathbb{Q}^\star + \beta^2 H) H^3 S A \iota}{\Delta_{\min}} + \frac{H^7 S A \iota^2}{\beta} \right) + \left( 1 - \frac{1}{T} \right) HT$$

$$= O\left( \frac{(\mathbb{Q}^\star + \beta^2 H) H^3 S A \iota}{\Delta_{\min}} + \frac{H^7 S A \iota^2}{\beta} \right). \tag{123}$$

The last inequality is because under the event $\mathcal{E}$, we have proved that

$$\sum_{k=1}^{K} \sum_{h=1}^{H} \text{clip}[(Q_h^k - Q_h^*)(s_h^k, a_h^k) \mid \Delta_{\min}] \leq O\left( \frac{(\mathbb{Q}^\star + \beta^2 H) H^3 S A \iota}{\Delta_{\min}} + \frac{H^7 S A \iota^2}{\beta} \right)$$

by Equation (122) and under the event $\mathcal{E}^c$,

$$\sum_{k=1}^{K} \sum_{h=1}^{H} \text{clip}[(Q_h^k - Q_h^*)(s_h^k, a_h^k) \mid \Delta_{\min}] \leq HT.$$

