# OpenReview forum: "Gap-Dependent Bounds for Q-Learning using Reference-Advantage Decomposition"
_ICLR.cc/2025/Conference — ICLR 2025 Spotlight_

### Official Review · Reviewer_Djdi · 2024-11-03

**Soundness:** 3
**Presentation:** 2
**Contribution:** 3
**Rating:** 6
**Confidence:** 4

**Summary:**

This work studies the instance-dependent regret guarantee in tabular Markov Decision Processes. The author focuses on the minimal sub-optimality gap structure and provides a logarithmic regret guarantee for two existing algorithms: UCB-Advantage and Q-EarlySettled-Advantage. Compared with previous instance-dependent guarantees, this work achieves a variance-aware regret bound that improves by a factor of H even under maximum variance. Additionally, when variance is low (e.g., deterministic transitions), the regret demonstrates improved dependency on the minimal sub-optimality gap. Furthermore, the author also proposes a gap-dependent policy-switching cost for the UCB-Advantage algorithm.

**Strengths:**

1. The author first proposes a novel algorithm that achieves a variance-aware regret bound with respect to the minimal sub-optimality gap.

2. The author also proposes an instance-dependent policy-switching cost for the UCB-Advantage algorithm, which could be of independent interest.

3. When variance is low (e.g., in deterministic transitions), the regret exhibits improved dependency on the minimal sub-optimality gap.

**Weaknesses:**

The main weakness is that the improvement in this work over existing results appears too limited.

1. As discussed in line 264, the instance-dependent regret bound depends on the point-wise sub-optimality gap. In comparison, this work relies on the minimal sub-optimality gap across all state-action pairs. In most situations, the sub-optimality gap varies significantly across different state-action pairs, leading to a weaker performance in the regret guarantee presented in this work.

2. For the instance-dependent guarantee with zero variance, this work achieves a sub-linear dependency on the sub-optimality gap. However, a similar result already exists without relying on the minimal sub-optimality gap assumption [1]. Compared with previous results, this work demonstrates worse dependency on the episode length H and the sub-optimality gap.
[1] Sharp Variance-Dependent Bounds in Reinforcement Learning: Best of Both Worlds in Stochastic and Deterministic Environments

3. Regarding the gap-dependent policy-switching cost, the claim in line 136 appears incorrect. When the optimal action set is small, the dominant term in equation (4) becomes the second term, resulting in an improvement of only
log T rather than a factor of A, which is minor.

4. Regarding technical novelty, the author claims the introduction of a surrogate reference function; however, the importance of this reference function is not clearly explained in section 3.2. It would be helpful to further highlight its effect in the proof sketch.

**Questions:**

1. In line 310, there is a typo fo$Q_h^k-Q_h^k$.

2. In line 313, it seems questionable that the first term in G3 does not diminish to zero, while the regret should converge to zero as the episode k becomes sufficiently large.

3. Lemma B.3 seems incorrect when $\check{n}(s,a)=1$ immediately after a reset to 0.

4. The $N(s,a)$ in Algorithm 1 should be $n(s,a)$.

---

> ### Author Response · Authors · 2024-11-19
> **Responses to Reviewer Djdi (part one)**
>
> We thank the reviewer for the careful reading and thoughtful comments. We have addressed the reviewer's questions in detail below and revised the paper accordingly. The changes are marked in blue in the revised manuscript. We hope that the responses provided and the updates made to the paper satisfactorily address the reviewer’s concerns.
>
> **Weakness 1:** Dependency on the minimal sub-optimality gap.
>
> Thanks for this question. It is important for us to explain the dependency on the sub-optimality gap.
>
> First, the regret upper bound in [2] does depend on the minimal sub-optimality gap explicitly. In our original draft, we only presented the main term of the regret bound in [2] for simplicity. In fact, the full regret upper bound in [2] is given by:
> $$O \left( \left( \sum_{h=1}^H\sum_{s\in\mathcal{S}}\sum_{a \neq \pi_h^\star(s)} \frac{1}{\Delta_h(s,a)} + \frac{|Z_{\textnormal{mul}}|}{\Delta_{\textnormal{min}}}+ SA \right) H^5 \log(K) \right),$$
> where $Z_{\textnormal{mul}} = \left\\{(h,s,a)|\Delta_h(s,a) = 0 \land |Z_{\textnormal{opt}}^h(s) | >1\right\\}$ and $Z_{\textnormal{opt}}^h(s) = \left\\{a|\Delta_h(s,a) = 0\right\\}$.
>
> In MDPs where $\Delta_h(s,a) = \Theta(\Delta_{\textnormal{min}})$ for $\Theta(HSA)$ state-action-step triples (e.g. the example in Theorem 1.3 of [2]), or when there are $\Omega(A)$ optimal actions for each state-step pair $(s,h)$, this upper bound becomes:
> $$O \left(\frac{H^6SA}{\Delta_{\textnormal{min}}}\log(K)\right).$$
> It coincides with [6] and is worse than ours. To avoid confusion, we have utilized the full regret upper bound and integrated these discussions into our revised draft (see lines 260--269).
>
> Second, Theorem 2.3 in Section 2.2 of [3] shows that the dependency on $\frac{S}{\Delta_{\textnormal{min}}}$ is unavoidable for optimism-based algorithms such as UCB-Advantage and Q-EarlySettled-Advantage that were analyzed in our paper. Therefore, it is reasonable to include the minimal sub-optimality gap term in the regret upper bound.
>
> Last but not least, we have conducted numerical experiments in Appendix F, despite the convergence result in [2], its computational efficiency is worse than other algorithms, including  UCB-Hoeffding analyzed in [6].
>
> **Weakness 2:** Comparison with the previous work [1].
>
> To the best of our knowledge, existing literature for model-free algorithms didn't reach gap-free regret and logarithmic regret simultaneously.
>
> For tabular MDPs, the variance-dependent regret bound in their model-free method UCB-Advantage-V of [1] is:
> $$\tilde{O}\left(\sqrt{\min\\{\textnormal{Var}_K^\Sigma, \textnormal{Var}^*K\\}HSA}+(H^{15}S^5A^3K)^{\frac{1}{4}}\right).$$
> In the case of deterministic MDPs where the variances are zero, the regret bound simplifies to  $\tilde{O}(T^{\frac{1}{4}})$. This polynomial dependency is much worse than our logarithmic bound when $T$ is sufficiently large. Thus, it is not a valid result for gap-dependent analysis that achieves a logarithmic dependency on $K$.
>
>
> **Weakness 3:** Regarding the gap-dependent policy-switching cost.
>
> Thank you for raising concerns regarding the gap-dependent policy-switching cost. It is important to highlight our improvements compared to the existing worst-case results and clarify their significance.
>
> In the revised manuscript (see lines 132–140), we address this by discussing the separate improvements of the two terms in Equation (4) compared to the $O(H^2SA\log T)$ bound in [7]. Specifically, the first term achieves an improvement by removing a factor of $A$, while the second term refines $\log T$ to $\log \log T$.
>
> Additionally, it is worth emphasizing that the improvement from $\log T$ to $\log \log T$ is significant. Two seminal works [4, 5] explicitly highlighted this as a key contribution, demonstrating how the policy-switching cost in [7, 8] was reduced from $\log T$ to $\log \log T$ and explaining the importance of this refinement in their analyses.

---

> ### Author Response · Authors · 2024-11-19
> **Responses to Reviewer Djdi (part two)**
>
> **Weakness 4:** The importance of the surrogate function.
>
> Following your suggestion, we have expanded the discussion on the surrogate reference function's theoretical contribution in Section 3.2, immediately after its definition (see lines 338–348). In the proof sketch, after Equation (16), we have added a sentence to emphasize the impact of the surrogate function and its connection to our discussion of $\mathcal{G}_1$ and $\mathcal{G}_2$ in Section 3.2. Furthermore, we provide a more detailed mathematical explanation. This content has also been included in Appendix G of our revised draft.
>
> Our proof relies on relating the regret to multiple groups of estimation error sums that take the form $\sum_{k=1}^K\omega_{h,k}^{(i)}(Q_h^k-Q_h^\star)(s_h^k,a_h^k)$. Here $\\{\omega_{h,k}^{(i)}\\}\_k$ are nonnegative weights and $i$ represents the group. Bounding the weighted sum via controlling each individual
> $Q_h^k(s_h^k,a_h^k) - Q_h^\star(s_h^k, a_h^k)$ by recursion on $h$ is a common technique for model-free optimism-based algorithms, which was used by all of [1, 2, 3]. [1] used it on gap-dependent regret analysis, and [2, 3] used it to control the reference setting errors $\sum_{k=1}^K (V_{h}^{\textnormal{R},k+1}(s_h^k) - V_{h}^{\textnormal{R},K+1}(s_h^k))$. However, their techniques are only limited to the Hoeffding-type update. In detail, the Hoeffding-type update in $Q$-function is given by
> $$Q_h^{k+1}(s_h^k,a_h^k) = r_h(s_h^k,a_h^k) + \sum_{n=1}^{N_h^{k+1}} \eta_n^{N_h^{k+1}} V_{h+1}^{k^n}(s_{h+1}^{k^n}) + \tilde{O}\left(\sqrt{H^3/N_h^{k+1}}\right),$$
> which is the key update of [1], and the update of $Q_h^{\textnormal{UCB},k+1}$ for [2, 3]. Accordingly, we can find that
> $$(Q_h^k - Q_h^\star)(s_h^k,a_h^k)\leq  H\eta_0^{N_h^k} + \sum_{n=1}^{N_h^{k}} \eta_n^{N_h^{k}} (V_{h+1}^{k^n} - V_{h+1}^\star)(s_{h+1}^{k^n})+ \tilde{O}\left(\sqrt{H^3/N_h^{k}}\right),$$
> which is the event in Definition 4.1 of [1]. Here, $\eta_0^{N_h^k} = 0$ when $N_h^k >0$. After taking the weighted sum with regard to $k\in [K]$ on both sides, we can establish recursions on $h$ where the main terms are $\sum_{k=1}^K\omega_{h,k}^{(i)}(Q_h^k-Q_h^\star)(s_h^k,a_h^k)$ and $\sum_{k=1}^K\omega_{h,k}^{(i)}\sum_{n=1}^{N_h^{k+1}} \eta_n^{N_h^{k+1}} (V_{h+1}^{k^n} - V_{h+1}^\star)(s_{h+1}^{k^n})$. With $\sum_{k=1}^K H\eta_0^{N_h^k}$ being easily controlled, the error generated by the recursion is mainly dominated by the weighted sum regarding the simple term $\tilde{O}\left(\sqrt{H^3/N_h^{k+1}}\right)$, which obviously vanishes when $k$ is large so that $N_h^k$ (the number of visit to $(s_h^k,a_h^k,h)$ is large.
>
> Here, we explain why [2, 3] only rely on the weighted sum $\sum_{k=1}^K\omega_{h,k}^{(i)}(Q_h^k-Q_h^\star)(s_h^k,a_h^k)$ with simple Hoeffding-type errors though their algorithms involve reference-advantage decomposition. Both methods incorporate a Hoeffding-type update (see $Q_h^{\textnormal{UCB},k+1}$ in Equation (7) in our revised draft), with which they bound the reference settling error by controlling the weighted sum. When analyzing the worst-case regret, they only need to relate the regret to $\sum_{k=1}^K(Q_h^k-Q_h^\star)(s_h^k,a_h^k)$, i.e., the sum instead of the weighted sum. However, in our gap-dependent regret analysis, because the weights do not adapt to the learning process (see our proof sketch for more details), we have to analyze each item $(Q_h^k-Q_h^\star)(s_h^k,a_h^k)$ individually in the weighted sum with complicated errors with new technical tools when we consider the reference-advantage update (Equation (8) in our revised draft).
>
>
>
>
> The reference-advantage update is listed as follows
> $$Q_h^{\textnormal{R},k+1}(s_h^k,a_h^k) = r_h^k(s_h^k,a_h^k)
> +\sum_{n=1}^{N_h^{k+1}}\Big(\eta_n^{N_h^{k+1}}(V_{h+1}^{k^n}-V_{h+1}^{\textnormal{R},k^n})+ u_n^{N_h^{k+1}}V_{h+1}^{\textnormal{R},k^n}\Big)(s_{h+1}^{k^n})+\tilde{R}^{h,k+1}. $$
> Here, $\\{\eta_n\^{N_h\^{k+1}}\\} \_{n=1}\^{N_h^{k+1}}$ are the corresponding nonnegative weights that sum to 1. $\\{u_n^{N_h^{k+1}}\\}\_{n=1}^{N_h^{k+1}}$ that sum to 1 are nonnegative weights for the reference function. $\tilde{R}^{h,k+1}$ is the cumulative bonus that contains variance estimators and dominates the variances in reference estimations and advantage estimations. Accordingly, we can find that
> $$(Q_h^k - Q_h^\star)(s_h^k,a_h^k)\leq H\eta_0^{N_h^k} +\sum_{n=1}^{N_h^{k}}\eta_n^{N_h^{k}}(V_{h+1}^{k^n}-V_{h+1}^*)(s_{h+1}^{k^n})  $$
> $$+\sum_{n=1}^{N_h^{k}}\Big(\eta_n^{N_h^{k}}(V_{h+1}^*-V_{h+1}^{\textnormal{R},k^n})+ u_n^{N_h^{k}}V_{h+1}^{\textnormal{R},k^n}\Big)(s_{h+1}^{k^n})- (1-\eta_0^{N_h^k})\mathbb{P}\_{(s_h^k,a_h^k,h)} V_{h+1}^\star+R^{h,k}.$$
> [cont'd on part three]

---

> ### Author Response · Authors · 2024-11-19
> **Responses to Reviewer Djdi (part three)**
>
> To establish the recursion on $h$ in the same way, when keeping the main terms unchanged and neglecting the term $H\eta_0^{N_h^k}$, the error term in our iteration becomes the weighted summation for
> $$ \sum_{n=1}^{N_h^{k}}\Big(\eta_n^{N_h^{k}}(V_{h+1}^{\star}-V_{h+1}^{\textnormal{R},k^n})+ u_n^{N_h^{k}}V_{h+1}^{\textnormal{R},k^n}\Big)(s_{h+1}^{k^n}) - (1-\eta_0^{N_h^k})\mathbb{P}\_{(s_h^k,a_h^k,h)} V_{h+1}^\star+R^{h,k}.$$
> It is much more complicated than $\tilde{O}(\sqrt{H^3/N_h^k})$ for the Hoeffding-type update.
>
> To handle this error, we propose a decomposition method following the reference-advantage structure. Naively, we can move towards advantage estimation errors (the first term), reference estimation errors (the second term), reference settling errors (the third term), the cumulative bonus (the fourth term), and a negative term (the last term), i.e.
> $$\sum_{n=1}^{N_h^{k}}\eta_n^{N_h^{k}}\left(\mathbb{P}\_{s_h^k,a_h^k,h}-\mathbb{1}\_{s_{h+1}^{k^n}} \right)(V_{h+1}^{\textnormal{R},K+1}-V_{h+1}^{\star})+ \sum_{n=1}^{N_h^{k}}u_n^{N_h^{k}}\left(\mathbb{1}\_{s_{h+1}^{k^n}} - \mathbb{P}\_{s_h^k,a_h^k,h}\right)V_{h+1}^{\textnormal{R},K+1}(s_{h+1}^{k^n})$$
> $$+\sum_{n=1}^{N_h^{k}}u_n^{N_h^{k}} (V_{h+1}^{\textnormal{R},k^n}-V_{h+1}^{\textnormal{R},K+1})(s_{h+1}^{k^n}) +R^{h,k}+ \sum_{n=1}^{N_h^{k}}\eta_n^{N_h^{k}}(V_{h+1}^{\textnormal{R},K+1}-V_{h+1}^{\textnormal{R},k^n})(s_{h+1}^{k^n})$$
> because the properties of the settled reference function $V_{h+1}^{\textnormal{R},K+1}$ is well-studied in [2, 3]. However, it will cause a non-martingale issue when we try to apply concentration inequalities as $V_{h+1}^{\textnormal{R},K+1}$ depends on the whole learning process. To solve this issue, we propose our **surrogate reference function** $\hat{V}\_{h}^{\textnormal{R},k}$ and decompose the error above as $\mathcal{G} \_1 := \sum_{n=1}^{N_h^k} \eta_n^{N_h^k} (\mathbb{P} \_{s_h^k,a_h^k,h}-\mathbb{1} \_{s_{h+1}^{k^n}})(\hat{V}\_{h+1}^{\textnormal{R},k^n} - V_{h+1}^\star),$ $\mathcal{G}\_2 := \sum_{n=1}^{N_h^k} u_n^{N_h^k} (\mathbb{1}\_{s_{h+1}^{k^n}} - \mathbb{P}\_{s_h^k,a_h^k,h})\hat{V}\_{h+1}^{\textnormal{R},k^n}$, $\mathcal{G}\_3 := \sum_{n=1}^{N_h^k} (u_n^{N_h^k} - \eta_n^{N_h^k}) \mathbb{P}\_{s_h^k,a_h^k,h}\hat{V}\_{h+1}^{\textnormal{R},k^n} + \sum_{n=1}^{N_h^k} u_n^{N_h^k}(V_{h+1}^{\textnormal{R},k^n} - \hat{V}\_{h+1}^{\textnormal{R},k^n})(s_{h+1}^{k^n})$, the bonus term $\mathcal{G}\_4 = R^{h,k}$, and a negative negligible term $\sum_{n=1}^{N_h^k} \eta_n^{N_h^k}(\hat{V}\_{h+1}^{\textnormal{R},k^n}-V_{h+1}^{\textnormal{R},k^n})(s_{h+1}^{k^n})$.  The first three terms correspond to advantage estimation error, reference estimation error, and reference settling error, respectively. Here, we creatively use the surrogate $\hat{V}_{h+1}^{\textnormal{R},k}$ as it is determined before the start of episode $k$. Thus, $\mathcal{G}_1,\mathcal{G}_2$ are martingale sums and can be controlled by concentration inequalities that are given in Equation (16), so the non-martingale challenge can be addressed. $\mathcal{G}_3$ corresponds to the reference settling error and can also be controlled given the settling conditions and properties of $\hat{V}_h^{\textnormal{R},k}(s)$. The bonus $\mathcal{G}_4$ is controlled using the same idea of bounding $\mathcal{G}_1,\mathcal{G}_2,\mathcal{G}_3$.
>
> Our decomposition above expands the technique of bounding the weighted sum of estimation errors to reference-advantage type estimations. In addition, we are the first to use the novel construction of the reference surrogates for reference-advantage decomposition in the literature, which makes a separate contribution to future work on off-policy methods and offline methods.

---

> ### Author Response · Authors · 2024-11-19
> **Responses to Reviewer Djdi (part four)**
>
> **Question 1:** Typo related to $Q_h^k$
>
> Thank you for your careful reading. We have corrected this typo and marked the changes in blue in the revised manuscript.
>
> **Question 2:** Diminishing of the first term in $\mathcal{G}_3$
>
> We provide an explanation on how the term $\sum_{n=1}^{N_h^k} (u_n^{N_h^k} - \eta_n^{N_h^k}) \mathbb{P}\_{s_h^k,a_h^k,h}\hat{V}\_{h+1}^{\textnormal{R},k^n}$ diminishes.
>
> First, we explain some basic facts about the weights. $\\{\eta_n^{N_h^k}\\}\_{n=1}^{N_h^k}$ and $\\{u_n^{N_h^k}\\}\_{n=1}^{N_h^k}$ correspond to the nonnegative weights for advantage estimations and reference estimations, respectively. The weights of each of them sum to 1. $\\{\eta_n^{N_h^k}\\}\_{n=1}^{N_h^k}$ concentrates on the lasted visits of proportion $\Theta(1/H)$ and $\\{u_n^{N_h^k}\\}\_{n=1}^{N_h^k}$ spreads evenly on all the $N_h^k$ visits. Thus, $\max_n\\{\eta_n^{N_h^k}\\}\_{n=1}^{N_h^k}\leq O(H/N_h^k)$ and $\max_n\\{u_n^{N_h^k}\\}\_{n=1}^{N_h^k}\leq O(1/N_h^k)$ according to Lemma D.1.
>
>
> Next, we explain facts about our surrogate reference function $\hat{V}\_{h+1}^{\textnormal{R},k^n}$. For each $(s,h)$, similar to the running reference function $V_{h+1}^{\textnormal{R},k^n}$ used in the algorithm, when some triggering condition is satisfied, $\hat{V}\_{h+1}^{\textnormal{R},k}(s)$ will settle on the settled reference function $V_{h+1}^{\textnormal{R},K+1}(s)$. Here, $K$ is the index for the last episode. Thus, $\hat{V}\_{h+1}^{\textnormal{R},k}$ will become a fixed function when $k$ is large. Mathematically, we can bound the cumulative difference with high probability as follows (similar to the proof of Equation (113), using the Lemma A.2):
> $$\sum_{k=1}^K \mathbb{P}\_{s_h^k,a_h^k,h}|\hat{V}\_{h+1}^{\textnormal{R},k+1} - \tilde{V}\_{h+1}^{\textnormal{R}}|\leq \tilde{O}(\mbox{poly}(HSA,\beta^{-1})),$$
> which is logarithmic in $K$. Here, we introduce $\tilde{V}\_{h}^{\textnormal{R}} = \min\\{V_{h}^{\textnormal{R},K+1},V_{h}^\star + \beta\\}$, the projected settled reference function, to incorporate situations that the reference function on some $(s,h)$ pair never settle.
>
> Now, we are ready to explain how this term diminishes. We can find that
> $$\left|\sum_{n=1}^{N_h^k} (u_n^{N_h^k} - \eta_n^{N_h^k}) \mathbb{P}\_{s_h^k,a_h^k,h}\hat{V}\_{h+1}^{\textnormal{R},k^n}\right|\leq \left|\sum_{n=1}^{N_h^k} (u_n^{N_h^k} - \eta_n^{N_h^k}) \mathbb{P}\_{s_h^k,a_h^k,h}\tilde{V}\_{h+1}^{\textnormal{R}}\right| + \sum_{n=1}^{N_h^k} (u_n^{N_h^k} + \eta_n^{N_h^k}) \left|\mathbb{P}\_{s_h^k,a_h^k,h}\hat{V}\_{h+1}^{\textnormal{R},k^n} - \mathbb{P}\_{s_h^k,a_h^k,h}\tilde{V}\_{h+1}^{\textnormal{R}}\right|.$$
> The first term of RHS is 0 as both groups of weights sum to 1. The second term can be upper bounded by $\tilde{O}(H\mbox{poly}(HSA,\beta^{-1}))/N_h^k$ given our discussion about the weights and the reference settling error. When the number of visits $N_h^k$ is large, this term diminishes.
>
> In our paper, we handle the reference settling error in the term $R_{\textnormal{else}}^{h,k}$. Please refer to our proof of Equation (27) in Appendix D.5.2 for more details.
>
> **Question 3:** The explanation of Lemma B.3
>
> Thanks for your careful reading. In Lemma B.3, the variable we used is $\check{n}_h^k(s,a)$, as defined on line 864, page 16 of the revised version. It is the number of visits to $(s,a,h)$ during the stage immediately before the stage of $k$-th episode. Based on the stage design reviewed from lines 848--857, Appendix C.1, we know that $\check{n}_h^k(s,a) = 0$ for $k$ in the first stage and $\check{n}_h^k(s,a) = e_i$ for $k$ in the stage $i+1$. Thus, we can proceed with the proof of Lemma B.3. Moreover, although $\check{n}(s,a)$ helps record the value of $\check{n}_h^k(s,a)$, it is a local counting variable used only in the UCB-Advantage algorithm.
>
> **Question 4:** Typo related to $N(s,a)$.
>
> Thank you for your careful reading. We have corrected this typo and marked the changes in blue in the revised manuscript.

---

> ### Author Response · Authors · 2024-11-19
> **Responses to Reviewer Djdi (part five)**
>
> [1] Runlong Zhou, Zhang Zihan, and Simon Shaolei Du. "Sharp variance-dependent bounds in reinforcement learning: Best of both worlds in stochastic and deterministic environments." ICML, 2023.
>
> [2] Haike Xu, Tengyu Ma, and Simon Du. "Fine-grained gap-dependent bounds for tabular MDPs via adaptive multi-step bootstrap." COLT, 2021.
>
> [3] Max Simchowitz, and Kevin G. Jamieson. "Non-asymptotic gap-dependent regret bounds for tabular MDPs." NeurIPS, 32 (2019).
>
> [4] Zihan Zhang, Yuhang Jiang, Yuan Zhou, and Xiangyang Ji. "Near-optimal regret bounds for multi-batch reinforcement learning." NeurIPS, 35 (2022): 24586-24596.
>
> [5] Dan Qiao, Ming Yin, Ming Min, and Yu-Xiang Wang. "Sample-efficient reinforcement learning with loglog (t) switching cost." ICML, 2022.
>
> [6] Kunhe Yang, Lin Yang, and Simon Du. "Q-learning with logarithmic regret." AISTATS, 2021.
>
> [7] Zihan Zhang, Yuan Zhou, and Xiangyang Ji. "Almost optimal model-free reinforcement learning via reference-advantage decomposition." NeurIPS, 33 (2020): 15198-15207.
>
> [8] Yu Bai, Tengyang Xie, Nan Jiang, and Yu-Xiang Wang. "Provably efficient q-learning with low switching cost." NeurIPS, 32 (2019).

---

> ### Author Response · Authors · 2024-11-25
> **Following up on the rebuttal**
>
> Thanks again for your insightful comments and valuable advice! We have uploaded the revised draft and replied to your suggested weaknesses and questions. If you have further questions or comments, we are happy to reply in the author-reviewer discussion period, which ends on Nov 26th at 11:59 pm, AoE. If our response resolves your concerns, we kindly ask you to consider raising the rating of our work. Thank you very much for your time and efforts!

---

> > ### Comment · Reviewer_Djdi · 2024-12-03
> >
> > The author's feedback addresses my concerns and I will improve my score.

---

### Official Review · Reviewer_mKwy · 2024-11-03

**Soundness:** 3
**Presentation:** 3
**Contribution:** 3
**Rating:** 8
**Confidence:** 3

**Summary:**

The paper analyzes the UCB-Advantage algorithm and a slightly modified version of the Q-EarlySettled-Advantage algorithms and provides the gap-dependent regret bounds and switching-cost bounds for them. Those two algorithms are worst-case optimal algorithms via references. Similarly, the gap-dependent regret bounds of such algorithms provided in this paper are better than the gap-dependent bounds of the algorithms without references in the literature. Discussions on the choice of hyperparameter $\beta$ and sketch of the proofs are clearly presented. For switching cost, analysis by separating the impact of the optimal and suboptimal actions is provided, so that the multiplicative factor before the leading order log(T) only depends on the tuples with optimal actions.

**Strengths:**

The analysis of "gap-dependent bound + reference-based algorithm" is novel and of interest to the RL theory study.

The proof sketch is clearly written. I checked some technical parts of the paper, and they are correct to me.

The technique of introducing an auxiliary "surrogate reference function" via cut-off based on optimal value function and $\beta$ to avoid non-martingale if using the last step reference function is new to gap-dependent bound.

**Weaknesses:**

I did not see major weaknesses in the paper. Here are some minor/barely ones.

In the discussion "Comparisons with Zhang et al. (2020); Li et al. (2021" after Theorem 3.3. The claim of better than worst-case since one is log(T) and the other is sqrt{T} is not quite fair. Either say it is asymptotic/for sufficiently large T, or discuss whether the proposed gap-dependent bounds can degrade to the worst-case bound naturally. The latter is worth investigating, but I do not see an immediate solution to this.

**Questions:**

Since the hyperparameter $\beta$ plays a more important bound-dependent role in the gap-dependent bound compared to that of the worst-case bound. Is there an adaptive way of updating the hyperparameter \beta? Say initialize \beta to be sufficiently large at the beginning while decreasing it gradually as the estimates get more accurate.

---

> ### Author Response · Authors · 2024-11-19
> **Responses to Reviewer mKwy**
>
> We thank the reviewer for the careful reading and thoughtful comments. We have addressed the reviewer's questions in detail below and revised the paper accordingly. The changes are marked in blue in the revised manuscript. We hope that the responses provided and the updates made to the paper satisfactorily address the reviewer’s concerns.
>
> **Weakness:** Comparison with the worst-case regret.
>
> Thank you for providing the suggestion on the comparison with the worst-case regret. In Section 3.2 of our revised draft, when comparing our results with the worst-case regret, we have added the description that $T\geq \tilde{\Theta}(\mbox{poly}(HSA, \Delta_{\textnormal{min}}^{-1}, \beta^{-1}))$ to be more precise.
>
>
> **Question:** The possibility of adaptively updating the hyper-parameter $\beta$.
>
> Thanks for your insightful comment. Algorithms that adaptively update $\beta$ can potentially avoid the hyper-parameter tuning. Currently, two technical challenges exist for designing an adaptive way of updating $\beta$.
>
> First, an important property of UCB-Advantage and Q-EarlySettled-Advantage is that the reference settling error can be well-controlled:
> $$\sum_{k=1}^K (V_{h}^{\textnormal{R},k+1}(s_h^k) - V_{h}^{\textnormal{R},K+1}(s_h^k))\leq \tilde{O}(\mbox{poly}(HSA,\beta^{-1})).$$
> If we need to adaptively update the parameter $\beta$, new technical efforts are needed.
>
> Second, we have proved that UCB-Advantage guarantees a gap-dependent expected regret of
>   $$O\left( \frac{\left(\mathbb{Q}^\star+\beta^2 H \right)H^3SA\log(SAT) } {\Delta_{\textnormal{min}}}+\frac{H^8S^2A\log(SAT)
>     \log(T)}{\beta^2}\right),$$
>     and
>     Q-EarlySettled-Advantage guarantees a gap-dependent expected regret of
> $$
> O\left( \frac{\left(\mathbb{Q}^\star+\beta^2 H \right)H^3SA\log (SAT) }{\Delta_{\textnormal{min}}}+ \frac{H^7SA\log^2(SAT)}{\beta}\right).
> $$
> Both upper bounds imply that optimal $\beta$ should strike a balance between the first terms and the second terms. Thus, we also need to find a valid termination condition when adaptively updating $\beta$.
>
> In summary, we agree with you that it is important to adaptively update $\beta$, and the technical challenges outlined above offer valuable directions for future work.

---

> ### Author Response · Authors · 2024-11-25
> **Following up on the rebuttal**
>
> Thanks again for your insightful comments and valuable advice! We have uploaded the revised draft and replied to your suggested weaknesses and questions. If you have further questions or comments, we are happy to reply in the author-reviewer discussion period, which ends on Nov 26th at 11:59 pm, AoE. If our response resolves your concerns, we kindly ask you to consider raising the rating of our work. Thank you very much for your time and efforts!

---

### Official Review · Reviewer_8zvK · 2024-11-04

**Soundness:** 3
**Presentation:** 3
**Contribution:** 3
**Rating:** 8
**Confidence:** 3

**Summary:**

The paper provides gap-dependent regret bounds for Q-learning-like algorithms which use variance estimation/also achieve variance dependent regret. They also provide an algorithm with a gap-dependent policy switching cost. The algorithms used (or small variations) appear in prior work. The authors describe a novel error decomposition and a surrogate reference function technique (which assists in the application of concentration inequalities) as main analytical contributions.

**Strengths:**

The regret bounds achieved by the paper improve upon those of prior works.

The gap-dependent analysis of the switching cost is new, and I think it is interesting to expand gap-dependent analyses beyond the regret performance metric.

I am somewhat unclear on the level of technical contribution of the paper (see questions), but it seems like the analysis techniques may be useful for future work involving reference-advantage decomposition algorithmic ideas.

**Weaknesses:**

The proof sketch is not very easy to follow and does not seem very useful for an initial read of the paper. This is especially due to the fact that the statements of the algorithms are only provided in the appendix and many forward references are made. I think it would be more helpful if the algorithms (or maybe just one) were provided in the main body of the text and the proof sketch were shortened to focus on higher-level steps and main differences compared to prior works.

The contribution appears to be somewhat limited, since it is a re-analysis of existing algorithms and the level of technical contribution of the analysis is not fully clear to me (see questions below). It is very common in RL for the analysis of the same/similar algorithms to be gradually refined, but then I think it is very important that the authors do a good job highlighting the analytical improvements.

**Questions:**

I would like to better understand the level of technical contribution of this paper.
Why are surrogate reference functions needed in your analyses but not those of the previous works (Zhang et al 2020, Li et al 2021)?
Could you provide more discussion on exactly how the error/regret decomposition differs from previous work and why it is novel/what issues are being solved?

Could you provide more comparison and discussion of related work which is model-based and tries to achieve similar goals (gap and variance dependent guarantees)?

---

> ### Author Response · Authors · 2024-11-19
> **Responses to Reviewer 8zvK (part one)**
>
> We thank the reviewer for the careful reading and thoughtful comments. We have addressed the reviewer's questions in detail below and revised the paper accordingly. The changes are marked in blue in the revised manuscript. We hope that the responses provided and the updates made to the paper satisfactorily address the reviewer’s concerns.
>
> **Weakness 1:** The proof sketch is not very easy to follow.
>
> Thank you for providing two helpful suggestions to improve the presentation of the proof sketch.
>
> On the one hand, in our revised draft, we have followed your suggestion to include the key steps of Q-EarlySettled-Advantage at the beginning of Section 3.2 (see lines 292-323 in the main body of the text), preceding the introduction of our surrogate reference function. Given its complexity and lengthy details, the full description of the algorithm remains in Appendix D.1.
>
> On the other hand, following your suggestion, we have shortened the proof sketch and also explained the main technical differences in bounding the weighted sum compared to prior works on gap-dependent regret analysis as in the updated Section 3.2 (see lines 338-348). A more detailed explanation has been provided in Appendix G in our revised draft.

---

> ### Author Response · Authors · 2024-11-19
> **Responses to Reviewer 8zvK (part two)**
>
> **Weakness 2 and Question 1:** On the theoretical contribution and the significance of the surrogate reference function as well as the error decomposition.
>
> Following your suggestion, we have included more discussion about the surrogate reference function's theoretical contribution in Section 3.2 after its definition. Next, we explain it in a more mathematical manner. The following content is also included in Appendix G in our revised draft.
>
> Our proof relies on relating the regret to multiple groups of estimation error sums that take the form $\sum_{k=1}^K\omega_{h,k}^{(i)}(Q_h^k-Q_h^\star)(s_h^k,a_h^k)$. Here $\\{\omega_{h,k}^{(i)}\\}\_k$ are nonnegative weights and $i$ represents the group. Bounding the weighted sum via controlling each individual
> $Q_h^k(s_h^k,a_h^k) - Q_h^\star(s_h^k, a_h^k)$ by recursion on $h$ is a common technique for model-free optimism-based algorithms, which was used by all of [1, 2, 3]. [1] used it on gap-dependent regret analysis, and [2, 3] used it to control the reference setting errors $\sum_{k=1}^K (V_{h}^{\textnormal{R},k+1}(s_h^k) - V_{h}^{\textnormal{R},K+1}(s_h^k))$. However, their techniques are only limited to the Hoeffding-type update. In detail, the Hoeffding-type update in $Q$-function is given by
> $$Q_h^{k+1}(s_h^k,a_h^k) = r_h(s_h^k,a_h^k) + \sum_{n=1}^{N_h^{k+1}} \eta_n^{N_h^{k+1}} V_{h+1}^{k^n}(s_{h+1}^{k^n}) + \tilde{O}\left(\sqrt{H^3/N_h^{k+1}}\right),$$
> which is the key update of [1], and the update of $Q_h^{\textnormal{UCB},k+1}$ for [2, 3]. Accordingly, we can find that
> $$(Q_h^k - Q_h^\star)(s_h^k,a_h^k)\leq  H\eta_0^{N_h^k} + \sum_{n=1}^{N_h^{k}} \eta_n^{N_h^{k}} (V_{h+1}^{k^n} - V_{h+1}^\star)(s_{h+1}^{k^n})+ \tilde{O}\left(\sqrt{H^3/N_h^{k}}\right),$$
> which is the event in Definition 4.1 of [1]. Here, $\eta_0^{N_h^k} = 0$ when $N_h^k >0$. After taking the weighted sum with regard to $k\in [K]$ on both sides, we can establish recursions on $h$ where the main terms are $\sum_{k=1}^K\omega_{h,k}^{(i)}(Q_h^k-Q_h^\star)(s_h^k,a_h^k)$ and $\sum_{k=1}^K\omega_{h,k}^{(i)}\sum_{n=1}^{N_h^{k+1}} \eta_n^{N_h^{k+1}} (V_{h+1}^{k^n} - V_{h+1}^\star)(s_{h+1}^{k^n})$. With $\sum_{k=1}^K H\eta_0^{N_h^k}$ being easily controlled, the error generated by the recursion is mainly dominated by the weighted sum regarding the simple term $\tilde{O}\left(\sqrt{H^3/N_h^{k+1}}\right)$, which obviously vanishes when $k$ is large so that $N_h^k$ (the number of visit to $(s_h^k,a_h^k,h)$ is large.
>
> Here, we explain why [2, 3] only rely on the weighted sum $\sum_{k=1}^K\omega_{h,k}^{(i)}(Q_h^k-Q_h^\star)(s_h^k,a_h^k)$ with simple Hoeffding-type errors though their algorithms involve reference-advantage decomposition. Both methods incorporate a Hoeffding-type update (see $Q_h^{\textnormal{UCB},k+1}$ in Equation (7) in our revised draft), with which they bound the reference settling error by controlling the weighted sum. When analyzing the worst-case regret, they only need to relate the regret to $\sum_{k=1}^K(Q_h^k-Q_h^\star)(s_h^k,a_h^k)$, i.e., the sum instead of the weighted sum. However, in our gap-dependent regret analysis, because the weights do not adapt to the learning process (see our proof sketch for more details), we have to analyze each item $(Q_h^k-Q_h^\star)(s_h^k,a_h^k)$ individually in the weighted sum with complicated errors with new technical tools when we consider the reference-advantage update (Equation (8) in our revised draft).
>
>
>
>
> The reference-advantage update is listed as follows
> $$Q_h^{\textnormal{R},k+1}(s_h^k,a_h^k) = r_h^k(s_h^k,a_h^k)
> +\sum_{n=1}^{N_h^{k+1}}\Big(\eta_n^{N_h^{k+1}}(V_{h+1}^{k^n}-V_{h+1}^{\textnormal{R},k^n})+ u_n^{N_h^{k+1}}V_{h+1}^{\textnormal{R},k^n}\Big)(s_{h+1}^{k^n})+\tilde{R}^{h,k+1}. $$
> Here, $\\{\eta_n\^{N_h\^{k+1}}\\} \_{n=1}\^{N_h^{k+1}}$ are the corresponding nonnegative weights that sum to 1. $\\{u_n^{N_h^{k+1}}\\}\_{n=1}^{N_h^{k+1}}$ that sum to 1 are nonnegative weights for the reference function. $\tilde{R}^{h,k+1}$ is the cumulative bonus that contains variance estimators and dominates the variances in reference estimations and advantage estimations. Accordingly, we can find that
> $$(Q_h^k - Q_h^\star)(s_h^k,a_h^k)\leq H\eta_0^{N_h^k} +\sum_{n=1}^{N_h^{k}}\eta_n^{N_h^{k}}(V_{h+1}^{k^n}-V_{h+1}^*)(s_{h+1}^{k^n})  $$
> $$+\sum_{n=1}^{N_h^{k}}\Big(\eta_n^{N_h^{k}}(V_{h+1}^*-V_{h+1}^{\textnormal{R},k^n})+ u_n^{N_h^{k}}V_{h+1}^{\textnormal{R},k^n}\Big)(s_{h+1}^{k^n})- (1-\eta_0^{N_h^k})\mathbb{P}\_{(s_h^k,a_h^k,h)} V_{h+1}^\star+R^{h,k}.$$
> [cont'd on part three]

---

> ### Author Response · Authors · 2024-11-19
> **Responses to Reviewer 8zvK (part three)**
>
> To establish the recursion on $h$ in the same way, when keeping the main terms unchanged and neglecting the term $H\eta_0^{N_h^k}$, the error term in our iteration becomes the weighted summation for
> $$ \sum_{n=1}^{N_h^{k}}\Big(\eta_n^{N_h^{k}}(V_{h+1}^{\star}-V_{h+1}^{\textnormal{R},k^n})+ u_n^{N_h^{k}}V_{h+1}^{\textnormal{R},k^n}\Big)(s_{h+1}^{k^n}) - (1-\eta_0^{N_h^k})\mathbb{P}\_{(s_h^k,a_h^k,h)} V_{h+1}^\star+R^{h,k}.$$
> It is much more complicated than $\tilde{O}(\sqrt{H^3/N_h^k})$ for the Hoeffding-type update.
>
> To handle this error, we propose a decomposition method following the reference-advantage structure. Naively, we can move towards advantage estimation errors (the first term), reference estimation errors (the second term), reference settling errors (the third term), the cumulative bonus (the fourth term), and a negative term (the last term), i.e.
> $$\sum_{n=1}^{N_h^{k}}\eta_n^{N_h^{k}}\left(\mathbb{P}\_{s_h^k,a_h^k,h}-\mathbb{1}\_{s_{h+1}^{k^n}} \right)(V_{h+1}^{\textnormal{R},K+1}-V_{h+1}^{\star})+ \sum_{n=1}^{N_h^{k}}u_n^{N_h^{k}}\left(\mathbb{1}\_{s_{h+1}^{k^n}} - \mathbb{P}\_{s_h^k,a_h^k,h}\right)V_{h+1}^{\textnormal{R},K+1}(s_{h+1}^{k^n})$$
> $$+\sum_{n=1}^{N_h^{k}}u_n^{N_h^{k}} (V_{h+1}^{\textnormal{R},k^n}-V_{h+1}^{\textnormal{R},K+1})(s_{h+1}^{k^n}) +R^{h,k}+ \sum_{n=1}^{N_h^{k}}\eta_n^{N_h^{k}}(V_{h+1}^{\textnormal{R},K+1}-V_{h+1}^{\textnormal{R},k^n})(s_{h+1}^{k^n})$$
> because the properties of the settled reference function $V_{h+1}^{\textnormal{R},K+1}$ is well-studied in [2, 3]. However, it will cause a non-martingale issue when we try to apply concentration inequalities as $V_{h+1}^{\textnormal{R},K+1}$ depends on the whole learning process. To solve this issue, we propose our **surrogate reference function** $\hat{V}\_{h}^{\textnormal{R},k}$ and decompose the error above as $\mathcal{G} \_1 := \sum_{n=1}^{N_h^k} \eta_n^{N_h^k} (\mathbb{P} \_{s_h^k,a_h^k,h}-\mathbb{1} \_{s_{h+1}^{k^n}})(\hat{V}\_{h+1}^{\textnormal{R},k^n} - V_{h+1}^\star),$ $\mathcal{G}\_2 := \sum_{n=1}^{N_h^k} u_n^{N_h^k} (\mathbb{1}\_{s_{h+1}^{k^n}} - \mathbb{P}\_{s_h^k,a_h^k,h})\hat{V}\_{h+1}^{\textnormal{R},k^n}$, $\mathcal{G}\_3 := \sum_{n=1}^{N_h^k} (u_n^{N_h^k} - \eta_n^{N_h^k}) \mathbb{P}\_{s_h^k,a_h^k,h}\hat{V}\_{h+1}^{\textnormal{R},k^n} + \sum_{n=1}^{N_h^k} u_n^{N_h^k}(V_{h+1}^{\textnormal{R},k^n} - \hat{V}\_{h+1}^{\textnormal{R},k^n})(s_{h+1}^{k^n})$, the bonus term $\mathcal{G}\_4 = R^{h,k}$, and a negative negligible term $\sum_{n=1}^{N_h^k} \eta_n^{N_h^k}(\hat{V}\_{h+1}^{\textnormal{R},k^n}-V_{h+1}^{\textnormal{R},k^n})(s_{h+1}^{k^n})$.  The first three terms correspond to advantage estimation error, reference estimation error, and reference settling error, respectively. Here, we creatively use the surrogate $\hat{V}_{h+1}^{\textnormal{R},k}$ as it is determined before the start of episode $k$. Thus, $\mathcal{G}_1,\mathcal{G}_2$ are martingale sums and can be controlled by concentration inequalities that are given in Equation (16), so the non-martingale challenge can be addressed. $\mathcal{G}_3$ corresponds to the reference settling error and can also be controlled given the settling conditions and properties of $\hat{V}_h^{\textnormal{R},k}(s)$. The bonus $\mathcal{G}_4$ is controlled using the same idea of bounding $\mathcal{G}_1,\mathcal{G}_2,\mathcal{G}_3$.
>
> Our decomposition above expands the technique of bounding the weighted sum of estimation errors to reference-advantage type estimations. In addition, we are the first to use the novel construction of the reference surrogates for reference-advantage decomposition in the literature, which makes a separate contribution to future work on off-policy methods and offline methods.

---

> ### Author Response · Authors · 2024-11-19
> **Responses to Reviewer 8zvK (part four)**
>
> **Question 2:** Comparison with other model-based work with similar goals.
>
> Thanks for this suggestion. Following your suggestion, we provide a comparison between our work and two model-based algorithms [4] and [5] from three different aspects:
>
> **Memory requirement:**
>
> Model-based algorithms such as [4] and [5] need to store estimates of transition kernels, so the memory requirement is $O(S^2AH)$, which is $S$ times larger than model-free algorithms.
>
> **Policy switching cost:**
>
> These two model-based algorithms do not benefit from a logarithmic policy-switching cost.
>
> **Regret upper bound:**
>
> [4] and [5] provide two different gap-dependent regret bounds.
>
> In [4], the regret upper bound is given by:
> $$O\left(\sum_{h=1}^H\sum_{s \in \mathcal{S}}\sum_{a \neq \pi_h^\star(s)} \frac{H \mathbb{Q}^*_{s,a}}{\Delta_h(s,a)}\log(T) + \frac{H|Z_{\textnormal{opt}}|\mathbb{Q}^*}{\Delta_{\textnormal{min}}}\log(T)\right),$$
> where $Z_{\textnormal{opt}} = \\{(s,a,h)|a = \pi_h^*(s)\\}$ and $\mathbb{Q}^*\_{s,a} = \max_{h}\\{\mathbb{V}\_{s,a,h}(V_{h+1}^\star)\\}$. Since $SH \leq |Z\_{\textnormal{opt}}| \leq SAH$, the dependency on the minimum sub-optimality gap is at least $O\left(\frac{\mathbb{Q}^* H^2S}{\Delta_{\textnormal{min}}}\log(T)\right)$. In MDPs where $\Delta_h(s,a) = \Theta(\Delta_{\textnormal{min}})$ for $\Theta(HSA)$ state-action-step triples (e.g. the example in Theorem 1.3 of [6]) or $|Z_{\textnormal{opt}}| = \Theta(SAH)$, the regret bound simplifies to
> $$O\left(\frac{\mathbb{Q}^*H^2SA}{\Delta_{\textnormal{min}}}\log(T)\right),$$
> which is better than our bound by only a factor of $H$ under their greater memory requirement.
>
> Using the same algorithm as in [4], [5] provides another regret upper bound:
> $$O\left(\sum_{h=1}^H\sum_{s \in \mathcal{S}} \sum_{\bar{\Delta}\_h(s,a) >0} \frac{\mathbb{Q}^*\_{s,a}}{\bar{\Delta}\_h(s,a)}\log(T) \right).$$
> Here $\bar{\Delta}\_h(s,a)$ is called the return gap (see the definition in Definition 3.1 of [5]). When the sub-optimality gap $\Delta_h(s,a) = 0$, the return gap $\bar{\Delta}\_h(s,a)$ can be as large as $\frac{H}{\Delta_{\textnormal{min}}}$. Compared to [4], while this return gap tightens the bound, it does not improve the dependency on $H$.
>
> After these comparisons, it is worth pointing out that, despite improving the regret bound of our work by a factor of $H$, model-based algorithms such as [4] and [5] require a significantly larger memory requirement by a factor of $S$ and do not benefit from a logarithmic policy switching cost. As a result, in many practical applications (e.g., Atari games), model-free algorithms are more helpful in dealing with high memory consumption.
>
> [1] Kunhe Yang, Lin Yang, and Simon Du. "Q-learning with logarithmic regret." AISTATS, 2021.
>
> [2] Zihan Zhang, Yuan Zhou, and Xiangyang Ji. "Almost optimal model-free reinforcement learning via reference-advantage decomposition." NeurIPS, 33 (2020): 15198-15207.
>
> [3] Gen Li, Laixi Shi, Yuxin Chen, and Yuejie Chi. "Breaking the sample complexity barrier to regret-optimal model-free reinforcement learning." NeurIPS, 34 (2021): 17762-17776.
>
> [4] Max Simchowitz, and Kevin G. Jamieson. "Non-asymptotic gap-dependent regret bounds for tabular MDPs." NeurIPS, 32 (2019).
>
> [5] Christoph Dann, Teodor V. Marinov, Mehryar Mohri, and Julian Zimmert. "Beyond value-function gaps: Improved instance-dependent regret bounds for episodic reinforcement learning." NeurIPS, 34 (2021): 1-12.
>
> [6] Haike Xu, Tengyu Ma, and Simon Du. "Fine-grained gap-dependent bounds for tabular MDPs via adaptive multi-step bootstrap." COLT, 2021.

---

> ### Author Response · Authors · 2024-11-25
> **Following up on the rebuttal**
>
> Thanks again for your insightful comments and valuable advice! We have uploaded the revised draft and replied to your suggested weaknesses and questions. If you have further questions or comments, we are happy to reply in the author-reviewer discussion period, which ends on Nov 26th at 11:59 pm, AoE. If our response resolves your concerns, we kindly ask you to consider raising the rating of our work. Thank you very much for your time and efforts!

---

> ### Comment · Reviewer_8zvK · 2024-11-26
>
> Thank you for your extensive responses and paper improvements. I have raised my score and recommend acceptance.

---

### Official Review · Reviewer_sZn1 · 2024-11-04

**Soundness:** 3
**Presentation:** 3
**Contribution:** 3
**Rating:** 8
**Confidence:** 2

**Summary:**

This paper establishes improved gap-dependent upper bounds on finite-horizon episodic Markov decision processes (MDPs). There already exists a gap-dependent upper bound of $\tilde O( \Delta_{\min}^{-1} H^6 SA)$. To provide improved guarantees, the paper analyzes two algorithms with variance-aware regret-analysis, UCB-Advantage due to Zhang et al. 2020 and Q-EarlySettled-Advantage due to Li et al. 2021. The paper proves that both algorithms admit regret upper bounds of $\tilde O( \Delta_{\min}^{-1} H^5 SA)$.

**Strengths:**

- Improved gap-dependent regret upper bounds for learning finite-horizon episodic MDPs are provided.
- The guarantees are obtained by analyzing some existing near-optimal algorithms for learning finite-horizon episodic MDPs.
- The regret analysis based on decomposing the errors into reference estimations, advantage estimations, and reference settling seems technically novel.

**Weaknesses:**

-

**Questions:**

Is it possible to demonstrate how close the provided regret upper bounds are to optimality? Are there gap-dependent regret lower bounds?

---

> ### Author Response · Authors · 2024-11-19
> **Responses to Reviewer sZn1**
>
> We thank the reviewer for the careful reading and thoughtful comments. We have addressed the reviewer's questions in detail below and revised the paper accordingly. The changes are marked in blue in the revised manuscript. We hope that the responses provided and the updates made to the paper satisfactorily address the reviewer’s concerns.
>
> **Question:** Gap-dependent regret lower bounds.
>
> Thanks for this insightful comment. To date, no optimal upper bounds have been established for gap-dependent tabular MDPs, nor has an information-theoretic lower bound (optimal lower bound) been identified for this problem. Existing lower bounds primarily serve to highlight the inevitability of certain terms in the corresponding regret upper bounds. However, their broader significance to other works remains relatively limited.
>
> Next, we discuss two existing non-asymptotic lower bounds for gap-dependent tabular MDPs.
>
>
>
> Theorem C.6 in [1] provides a lower bound $$O\left(\frac{HSA}{\Delta_{\textnormal{min}}}\log(K)\right)$$
> for a family of hard instances, which is introduced in Figure 2 of [1].
>
> Theorem 5.1 in [1] establishes another regret lower bound  $$O\left(\frac{|Z_{\textnormal{mul}}|}{\Delta_{\textnormal{min}}}\log(K)\right),$$ where $Z_{\textnormal{mul}} = \left\\{(h,s,a)|\Delta_h(s,a) = 0 \land |Z_{\textnormal{opt}}^h(s) | >1\right\\}$ and $Z_{\textnormal{opt}}^h(s) = \left\\{a|\Delta_h(s,a) = 0\right\\}$.
>
>
> Theorem 1.1 in [1] also gives a regret upper bound. When there are $\Omega(A)$ optimal actions for each state-step pair $(s,h)$ or $\Delta_h(s,a) = \Theta(\Delta_{\textnormal{min}})$ for $\Theta(HSA)$ state-action-step triples, the regret upper bound is given by:
> $$O \left( \frac{H^6SA}{\Delta_{\textnormal{min}}}\log(K)\right).$$
> which is also significantly worse than the above lower bounds. Moreover, the dependency on $H$ is worse than our results. Again, we emphasize that no current work reaches these lower bounds up to a gap-free term, which points to an important future research topic.
>
>
> [1] Haike Xu, Tengyu Ma, and Simon Du. "Fine-grained gap-dependent bounds for tabular MDPs via adaptive multi-step bootstrap." COLT, 2021.

---

> ### Author Response · Authors · 2024-11-25
> **Following up on the rebuttal**
>
> Thanks again for your insightful comments and valuable advice! We have uploaded the revised draft and replied to your suggested weaknesses and questions. If you have further questions or comments, we are happy to reply in the author-reviewer discussion period, which ends on Nov 26th at 11:59 pm, AoE. If our response resolves your concerns, we kindly ask you to consider raising the rating of our work. Thank you very much for your time and efforts!

---

### Author Response · Authors · 2024-11-19
**Response to everyone**

We sincerely thank the reviewers for their thorough reading and insightful feedback. Below, we summarize the key updates incorporated into our revised manuscript. Changes in the revised draft are marked in blue for clarity.

1. **Section 3.2:** To improve the presentation of our technical contribution on the surrogate reference function, we have added a description of the key steps in the Q-EarlySettled-Advantage algorithm prior to its definition. Furthermore, we now discuss the challenges of bounding the weighted sum in greater detail, as outlined in lines 338–348 of the revised manuscript and Appendix G.

2. **Appendix E:** We have added this new section in the appendix to provide a comprehensive discussion of related work, addressing several points raised by the reviewers.

3. **Appendix F:** We have added this new section in the appendix to present numerical experiments comparing the performance of UCB-Advantage and Q-EarlySettled-Advantage against two other model-free algorithms: UCB-Hoeffding and AMB, demonstrating the numerical performance and providing evidence supporting the theoretical results.

4. **Appendix G:** We have added this new section in the appendix to provide a detailed mathematical explanation of the surrogate function, elaborating on our novel ideas and highlighting the main differences from prior works.

---

### Meta-Review · Area_Chair_ZjBQ · 2024-12-20

**Metareview:**

This submission studies gap-dependent bounds and policy switching cost for Q learning algorithms which use variance estimation.

This paper gives novel gap-dependent variance-aware regret bounds, and provides an algorithm with a gap-dependent policy switching cost. These theoretical contributions could be of interest to the RL theory community. The reviewers also voted unanimously for acceptance.

**Additional Comments On Reviewer Discussion:**

The reviewers raised concerns regarding the clarity of the proof sketch part of the paper and improvements over prior work. However, the authors provided detailed responses which successfully addressed those concerns and resulted in improved scores.

---

### Decision · Program_Chairs · 2025-01-22

Accept (Spotlight)